



# Sea level rise in Europe: observations and projections

Coordinating lead authors: Angélique Melet[1], Roderik van de Wal[2,3]

Authors: Angel Amores[4,5], Arne Arns[6], Alisée A. Chaigneau[1,7], Irina Dinu[8], Ivan D. Haigh[9], Tim H. J. Hermans[2], Piero

5 Lionello[10], Marta Marcos[4,5], H. E. Markus Meier[11], Benoit Meyssignac[12], Matthew D. Palmer[13,14], Ronja Reese[15], Matthew J.

R. Simpson[16], Aimée B.A. Slangen[17]

[1] Mercator Ocean International, Toulouse, 31400, France
[2] Institute for Marine and Atmospheric Research Utrecht, Utrecht University, Utrecht, the Netherlands
[3] Department of Physical Geography, Utrecht University, Princetonlaan 8a, 3584 CB Utrecht, The Netherlands
10 [4] Mediterranean Institute for Advanced Studies (IMEDEA, UIB-CSIC), Esporles, 07190, Spain
Department of Physics, University of the Balearic Islands, Palma, 07122, Spain
[6] Faculty of Agricultural and Environmental Sciences, University of Rostock, Justus-von-Liebig-Weg 6, 18059 Rostock, Germany
[7] IHCantabria - Instituto de Hidraulica Ambiental de la Universidad de Cantabria, 39011, Santander, Spain
[8] Institute for Marine Geology and Geoecology (GeoEcoMar), Bucharest, 024053, Romania
[9] School of Ocean and Earth Science, University of Southampton, European Way, Southampton, SO14 3ZH, United Kingdom
[10] University of Salento, Department of Environmental and Biological Sciences and Technologies, Lecce, Italy
[11] Department of Physical Oceanography and Instrumentation, Leibniz Institute for Baltic Sea Research Warnemünde, Rostock, 18119, Germany
[12] LEGOS, University of Toulouse, CNES, CNRS, IRD, UT3, Toulouse, France
[13] Met Office, FitzRoy Road, Exeter, EX1 3 PB, United Kingdom
[14] University of Bristol, Bristol, BS8 1UH, United Kingdom
[15] Department of Geography and Environmental Sciences, Northumbria University, Newcastle upon Tyne, United Kingdom
[16] Geodetic Institute, Norwegian Mapping Authority, Hønefoss, 3511, Norway
[17] Department of Estuarine and Delta Systems, NIOZ Royal Netherlands Institute for Sea Research, Yerseke, 4401 NT, Netherlands

*Correspondence to*: Angélique Melet (amelet@mercator-ocean.fr), Roderik van de Wal (R.S.W.vandeWal@uu.nl)



**Abstract.**

Sea level rise (SLR) is a major concern for Europe, where 30 million people live in the historical 1-in-100-year event flood coastal plains. The latest IPCC assessment reports provide a literature review on past and projected SLR, and their key findings are synthesized here with a focus on Europe. The present paper complements IPCC reports and contributes to the Knowledge Hub on SLR European Assessment Report. Here, the state of knowledge of observed and 21st century projected SLR and
changes in extreme sea levels (ESLs) are documented with more regional information for European basins as scoped with stakeholders. In Europe, satellite altimetry shows that absolute sea level trends are on average slightly above the global mean rate, with only a few areas showing no change or a slight decrease such as central parts of the Mediterranean Sea. The spatial pattern of absolute SLR in European Seas is largely influenced by internal climate modes, especially the North Atlantic Oscillation, which varies on year-to-year to decadal timescales. In terms of relative sea level rise (RSLR), vertical land motions
due to human induced subsidence and glacial isostatic adjustment (GIA) are important for many coastal European regions, leading to lower or even negative RSLR in the Baltic Sea, and to large rates of RSLR for subsiding coastlines. Projected 21st century local SLR for Europe is broadly in-line with projections of GMSLR rise in most places. Some European coasts are projected to experience a relative SLR by 2100 below the projected GMSLR, such as the Norwegian coast, the southern Baltic Sea, the northern part of the UK and Ireland. A relative sea level fall is projected for the northern Baltic Sea. RSLR along other
EU coasts is projected to be slightly above the GMSLR, for instance the Atlantic coasts of Portugal, Spain, France, Belgium and the Netherlands. Higher-resolution regionalized projections are needed to better resolve dynamic sea level changes especially in semi-enclosed basins, such as the Mediterranean Sea, North Sea, Baltic Sea and Black Sea. In addition to ocean dynamics, GIA and Greenland ice mass loss and associated Earth gravity, rotation and deformation effects are important drivers of spatial variations of projected European RSLR. High-end estimates of SLR in Europe are particularly sensitive to
uncertainties arising from the estimates of the Antarctic ice mass loss. Regarding ESLs, the frequency of occurrence of the historical centennial event (HCE) level is projected to be amplified for most EU coasts, except along the northern Baltic Sea coasts where a decreasing probability is projected because of relative sea level fall induced by GIA. The largest HCE amplification factors are projected for the southern European Seas (Mediterranean and Iberian Peninsula coasts), while the smallest amplification factors are projected in macro-tidal regions exposed to storms and induced large surges such as the
south-eastern North Sea. Finally, an emphasis is given on processes that are especially important for specific regions, such as waves, tides in the north-eastern Atlantic; vertical land motion for the European Arctic and Baltic Sea; seiches, meteo-tsunamis and medicanes in the Mediterranean Sea; non-linear interactions between drivers of coastal sea level extremes in the shallow North Sea.





## 1   Introduction

Sea level rise (SLR) is a major concern for Europe, where more than 50 million people live in low-elevation (≤ 10m) coastal zones and 30 million in the 100-year event flood coastal plains.

Sea level (SL) changes at the coast result from processes acting at various space and time scales, from extreme events to long-term SL rise, with the superposition of global, regional, and local variations. SLR is a direct consequence of climate change,

which is due to the current energy imbalance of our planet at the top of its atmosphere induced by anthropogenic emissions of greenhouse gases (e.g. Forster et al., 2021). As our planet reemits less energy to space than it receives from the Sun, an excess of energy, mostly in the form of heat, accumulates in the climate system. About 91% of the excess heat stored in the climate system has been absorbed in the oceans (Cheng et al., 2017; Von Schuckmann et al., 2020), causing a thermal expansion of the ocean leading to global mean SLR (GMSLR). The remainder of the excess heat has been absorbed by the atmosphere, land

ice, sea ice and land surface. As land ice (glaciers, ice sheets) melts and is discharged to the ocean, water is added to the ocean, increasing its mass and volume and thereby rising SL. Changes in land water storage due to natural hydrological cycle and human interventions also lead to ocean mass and SL changes.

SLR has not been and will not be uniform over the ocean (Fox-Kemper et al., 2021). At regional scale, mean SLR can deviate substantially from GMSLR due to a number of processes, with three key drivers. First, ocean circulations redistribute the

seawater mass, heat and salinity, leading to regional dynamic SL changes. Changes in ocean circulations are mostly driven by surface wind stress, but also by air-sea heat and freshwater fluxes (Forget and Ponte, 2015; Meyssignac et al., 2017; Todd et al., 2020) and by intrinsic ocean variability (Llovel et al., 2018; Sérazin et al., 2015). Regional dynamic SL changes are mostly steric (ocean density changes), with a predominance of its thermosteric component. Second, geographical redistribution of mass over the Earth, including contemporary or past transfers between land and ocean such as glacier and icesheets mass loss

or land water storage changes, induce changes in Earth gravity and rotation as well as viscoelastic solid-Earth deformations (called GRD effects). GRD effects induce SL changes through changes in the geoid and vertical land motion (VLM, Tamisiea, 2011). Glacial isostatic adjustment (GIA, Peltier, 2004) causes contemporary relative SL change due to GRD effects through ongoing viscous changes in the solid Earth caused by past changes in land ice, mostly through the deglaciation following the Last Glacial Maximum (∼ 20,000 years ago). Third, regional changes of atmospheric pressure loading over the ocean (due to

changes in atmospheric circulations and moisture content) induce regional changes in the inverted barometer effect at scales longer than about a month (Wunsch and Stammer, 1997). The inverted barometer effect is a relatively minor driver of regional SL changes at seasonal and longer timescales.



At more coastal scales, relative SL changes can be due to VLM of natural (sediment compaction in deltas, Earth tectonics,
GIA and contemporary solid Earth deformation due to land ice mass loss) or anthropogenic origin (pumping of groundwater,
weight of the built environment etc.). In many coastal megacities, including European ones, VLM can induce relative SL trends
similar to or larger than trends induced by oceanic and climate factors causing absolute SL changes (e.g., Nicholls et al., 2021;
Wu et al., 2022). In addition, several other processes lead to substantial SL deviations from the open ocean and should be
considered when estimating local SL changes at the coast (Woodworth et al., 2019). Among these processes are tides,
atmospheric surges, wind wave setup and swash, seiches, coastal waves, and effects of river discharges. Coastal SL variability
spans a wide range of temporal and spatial scales (Hughes et al., 2019; Woodworth et al., 2019). Processes driving coastal SL
change can also interact (e.g., Idier et al., 2019) due to their effects and dependence on water depth for instance.

European regional seas (see Sea Level Rise in Europe: knowledge gaps identified through a participatory approach) and their
bordering EU coasts are presenting contrasting environments, from open ocean environment (north-eastern Atlantic, European
Arctic) to semi-enclosed (North Sea) and quasi-enclosed seas (Baltic Sea, Mediterranean Sea and Black Sea), microtidal
(Mediterranean, Baltic and Black Seas) to mesotidal (European Arctic) and macrotidal environments (north-eastern Atlantic,
North Sea), deep to shallow seas on the continental shelf (North Sea, Baltic Sea), regions exposed to large swells or storms
under the North Atlantic storm track, and regions experiencing different VLM. These contrasted ocean environments induce
different past and projected SL changes over European Seas. Here, the state of knowledge of observed and 21$^{st}$ century
projected SLR changes in extremes is documented for European basins as part of the Knowledge Hub on Sea Level Rise
Assessment Report (see Sea Level Rise in Europe: A Knowledge Hub at the ocean-climate nexus).

First, a synthesis of the key findings of recent IPCC assessment reports on past and future SLR is provided in Section 2.1, with
a European perspective in Section 2.2. The following sections 3-6 complement the IPCC assessment reports and provide
extensive regional information on European observed and projected SL changes, as requested by stakeholders (see Sea Level
Rise in Europe: knowledge gaps identified through a participatory approach). Observations of SLR in Europe from tide gauges
(Section 3.1) and satellite altimetry (Section 3.2) are discussed, together with available sea level tools and data portals (Box
1). As VLM due to human induced subsidence and GIA is important for many coastal European regions, observations of this
component of RSLR are discussed in section 3.3. Observed changes in extreme sea levels (ESLs, Section 3.4) and a selection
of iconic historical storms causing coastal flooding in Europe and their consequences are reported (Box 2). In Section 4, drivers
of SLR and ESLs are discussed, with a focus on Europe. Projected changes in European sea level are presented in Section 5,
with a focus on projected 21$^{st}$ century changes in mean sea level and extremes. A discussion on tipping points, irreversibility
and commitment of SLR is also provided. Finally, a regional focus per European regional sea (north-eastern Atlantic, North
Sea, Arctic Ocean, Baltic Sea, Mediterranean and Black Seas) with key developments per region is provided in Section 6.





## 2 Summary of previous assessments

 **2.1. Synthesis of recent IPCC assessment reports**

In this Section we present a synthesis of the key findings of the two most recent assessment reports that provided comprehensive information on past and future SLR: (1) the IPCC Special Report on the Ocean and Cryosphere in a Changing Climate (IPCC, 2019; Oppenheimer et al., 2019) ; and (2) the IPCC Sixth Assessment Report of Working Group I (Fox-Kemper et al., 2021; IPCC, 2021).  The text in this Section is based primarily on the AR6 WG1 and SROCC Summaries for Policy Makers (IPCC, 2021; IPCC., 2022), which have been endorsed by international government delegations during the IPCC approval sessions. Recent progress and additional regional information are provided in subsequent Sections.

During the 20[th] century, global mean SL has risen faster than during any preceding century in at least the last 3,000 years. Moreover, the rate of GMSL rise has increased by almost a factor of three since observational records began. The average rate was about 1.3 mm yr[-1] during 1901-1971, increasing to about 1.9 mm yr[-1] during 1971-2006 and further increasing to about 3.7 mm yr[-1] during 2006-2018. Our understanding of the physical mechanisms of these past changes has increased through demonstrated closure of the observed GMSL budget, in particular after 1970 (Oppenheimer et al., 2019); (Fox-Kemper et al., 2021). For example, the acceleration of GMSL rise in recent decades is driven primarily by a four-fold increase in the rate of ice sheet mass loss since the 1990s. For the period since 2006, ice mass input to the ocean from ice sheets and glaciers exceeds all other contributions to GMSL rise. It is now understood that anthropogenic forcing was the main driver of the observed GMSL rise since at least 1971 (Slangen et al., 2016). There is also scientific evidence for changes in the drivers of sea level extreme events. There is emerging observational evidence that the intensity of major tropical cyclones (category 3-5) has increased over the last four decades and there is high confidence that anthropogenic climate change has increased some cyclone-driven ESL events. Extreme wave heights have increased in the Southern Ocean and North Atlantic since the 1980s and loss of sea ice has been linked to increased wave heights in the Arctic Ocean since the 1990s ;Oppenheimer et al., 2019).

One of the main innovations in AR6 was the use of emulators with multi-model ensembles and observational constraints to develop sea level projections that were consistent with the assessment of climate sensitivity (Forster et al., 2021; Fox-Kemper et al., 2021; IPCC, 2021). Another important innovation was the explicit treatment of the potential for accelerated future SLR associated with deeply uncertain ice sheet instability processes through illustrative high-end storylines, intended to aid decision-making. While these high-end storylines yielded much higher multi-century SLR projections than seen in previous IPCC reports, the *likely* range (i.e., the central two-thirds of the distribution) of the projections has remained relatively stable since the publication of the IPCC Fifth Assessment Report (Church et al., 2013) despite major advancements in the models and methods used in AR6 (Slangen et al., 2023).





The latest IPCC *likely* range projections of GMSL yield values at 2100 of 0.32–0.62 m (low GHG emissions, SSP1-2.6) and
0.63–1.01 m (very high GHG emissions, SSP5-8.5), relative to the 1995-2014 average (Figure 1). Furthermore, GMSL rise
approaching 2 m by 2100 and 5 m by 2150 cannot be ruled out for a very high GHG emissions scenario, due to deep uncertainty
in ice-sheet processes. On longer timescales, GMSL rise will continue for centuries to millennia due to continued deep-ocean
warming and ice-sheet melt, as these elements of the Earth system slowly adjust to the anthropogenic warming. Over the next
2,000 years AR6 assessed that GMSL will rise by about 2-3 m if surface warming is limited to 1.5°C relative to pre-industrial.
This rise increases to about 2-6 m with a peak warming of 2°C and about 19-22 m with a peak warming of 5°C.

At regional scales, it is *virtually certain* (99-100% probability) that mean RSLR will continue throughout the 21st century,
except in a few regions with large vertical land uplift rates. By 2100 it is projected that ESL events that occurred once per
century in the recent past will occur at least annually at more than half of all tide gauge locations around the world due to local
mean SLR (Fox-Kemper et al., 2021). SLR will increase the frequency and severity of coastal flooding in low-lying areas and
coastal erosion along most sandy coasts. The combination of more frequent ESLs and increased extreme rainfall/riverflow
events associated with an intensified hydrological cycle will make flooding more probable in coastal cities and settlements by
the sea (IPCC, 2021).

Despite the inevitability of SLR in the coming centuries, the science also shows the benefit of reduced GHG emissions in terms
of avoiding the worst future risks and buying more time to adapt to the changes. By the end of the 21st century, scenarios with
very low and low GHG emissions would strongly limit the rate of increase in the frequency of ESL events relative to higher
GHG emissions scenarios. Excluding uncertain ice-sheet processes, the assessed ranges of projected GMSL rise at 2300 under
a low GHG emissions are substantially lower (0.6-1.0 m in SROCC; 0.3–2.9 m in AR6) than for the very high GHG emissions
scenario (2.2-5.3 m in SROCC; 1.7-6.8 m in AR6), implying that strong mitigation is needed to prevent large SLR in 2300.





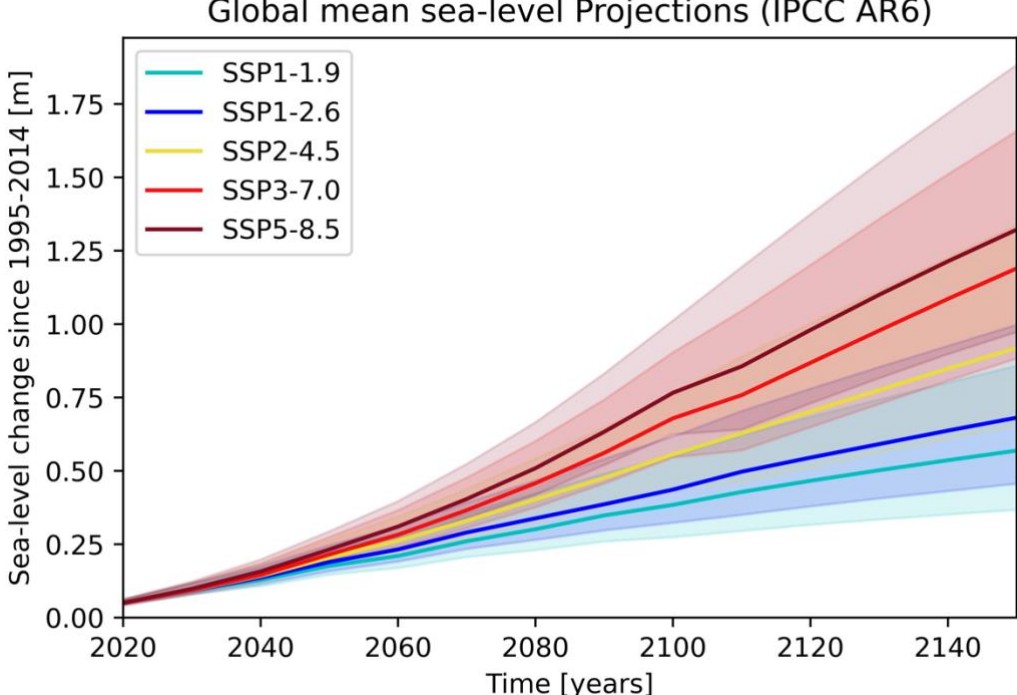

**Figure 1: Projected GMSLR from the 6<sup>th</sup> assessment report of the IPCC relative to 1995-2014. Median (50<sup>th</sup> percentile) projections**
**for all scenarios are indicated by the solid lines as shown in the figure legend. For each scenario, the shading shows the *likely* range**
**(17<sup>th</sup>-83<sup>rd</sup> percentiles) (Fox-Kemper et al., 2021).**

**2.2 The European Perspective**

Most coastal regions in Europe are currently experiencing a local SLR of a few mm per year, but there are large spatial
variations across the continent. A key driver of these spatial variations is GIA: the ongoing VLM associated with the solid
earth response to the last de-glaciation. The spatial pattern of this land motion is characterised by vertical land uplift in areas
covered by ice sheets during the last glacial period and land subsidence in other areas (Section 3.3). As a result, much of the
northern Scandinavian coastline is currently experiencing a local sea level fall, since the long-term rate of land uplift following
the last deglaciation exceeds the global warming driven contemporary SLR.


Projected local SLR for Europe is broadly in-line with projections of GMSL rise in most places (Section 5.1). GIA will continue
to be an important driver of spatial variations across the continent, with additional spatial differences also arising from the

effect of Greenland ice sheet mass loss on Earth's gravity field and also by local oceanographic processes (Fox-Kemper et al., 2021). There may also be highly localised VLM processes active either now or in the future, such as subsidence associated

with groundwater and hydrocarbon extraction or tectonic activity (Section 3.3). Risk-based decision-making should account for these additional non-climatic processes when assessing potential magnitudes or rates of future SLR.

The scientific consensus is that changes in future coastal flood hazard will be dominated by SLR, rather than changes in the drivers of sea level extremes such as waves, tides and surges  (e.g., Fox-Kemper et al., 2021a; Howard et al., 2019; Vousdoukas et al., 2018; see Sea Level Rise in Europe: impacts and consequences for more details). However, systematic changes in these

drivers could exacerbate local SLR and internal variability is expected to play a large role in shaping the evolution of wave and storm surge extremes on decadal timescales (Section 3.4, Section 5.3). In addition, there is a growing body of scientific evidence that suggests SLR could have substantive effects on local tidal characteristics (Haigh et al., 2020). Combined 21$^{st}$ century projections of SLR, tides, surges and waves for European coasts found the largest absolute increases in ESLs in the North Sea, followed by the Baltic Sea and Atlantic coasts of the UK and Ireland (Vousdoukas et al., 2017), but in the

Mediterranean the relative increase is larger implying a more urgent need to improve adaptation strategies. Changes in waves and storm surges were found to exacerbate SLR for most coasts with contributions of up to 40% of the change in sea level extremes. However, the response of waves and surges under climate change remains a key uncertainty (e.g., Howard et al., 2019). IPCC AR6 concluded that "relative SLR is *extremely likely* to continue around Europe (except in the northern Baltic Sea), contributing to increased coastal flooding in low-lying areas and shoreline retreat along most sandy coasts (*high*

*confidence*)" (Ranasinghe et al., 2021).

\*\*\*\*\*\*\*\*\*\*\* START OF BOX 1 \*\*\*\*\*\*\*\*\*\*\*

**Box 1: Common practices and available sea level tools and data portals**

As the impact of SL change is a local issue, it is important to communicate SL projections in a form that can be used by local

decision makers. In this Box, we will provide a non-exhaustive overview of online visualization tools and data portals which provide information on past and projected SL change.

The IPCC AR5 Chapter 13 (Church et al., 2013) made the SL projections available online, through the Integrated Climate Data Centre from the University of Hamburg (Table 1), but this was not actively communicated or referred to in the report, and the online tool was more science-focused than public-oriented. The focus on accessible regional information has increased





in the recent IPCC AR6 report, which produced an interactive atlas (https://interactive-atlas.ipcc.ch/Table 1), showing observations and projections of a wide range of climate variables for all IPCC working group 1 reference regions (Gutiérrez et al., 2021). For SL change, this atlas includes the SL projections for the different future climate scenarios. However, the atlas only shows SLR for three time periods for large regions. Therefore, in collaboration with NASA, the IPCC chapter 9 authors built an additional tool which specifically focuses on SL projections (Table 1). This tool allows the user to select and visualize

the projected changes for different time periods (by decade), scenarios, and contributions. It also provides projections at specific tide gauge locations using an interactive map. In addition, all the IPCC AR6 SL projections (global and gridded 1 by 1 degree regional) can be downloaded by the user from the NASA website and from a Zenodo archive (Table 1).

There are also several other online interactive SL tools. For instance, the INSeaPTION project (an ERA4CS European research consortium) has made a tool which includes IPCC AR5 and SROCC projections, but also allows for investigating different

scenarios, using sliders, and high-and-low-end scenarios (Table 1). The UK Met Office has made a 'sea level dashboard', where global mean projections (total and individual contributions) are connected to observations (Table 1). Focusing on local or national changes, there are also various online tools available. For instance, the Norwegian Mapping Authority has developed a tool which provides users with information on observed and forecasted water levels, predicted tides, extreme still water levels, VLM, and past and future sea level for Norway (Table 1). Users can find information on vertical datums (various tidal

datums and Norway's national height system NN2000) which are relevant for planning decisions, and on sea level impacts (more in Sea Level Rise in Europe: impacts and consequences, Section 5).

Several online data portals provide information on past and projected SL changes. The PSMSL, for instance, provides an overview of tide gauge measurements around the world (Table 1). Regarding coastal ESLs, return levels and simulated time series since 1979 from a global hydrodynamic model forced with atmospheric pressure, wind and tides are available in the

Climate Data Store (CDS) of the Copernicus Climate Change Service (C3S), which also hosts various other SL observation and projection-related datasets (Table 1). The Joint Research Centre hosts a number of datasets as part of the Large Scale Integrated Sea Level and Coastal Assessment Tool (LISCOAST), including historical and projected ESLs along the European coasts. A global database of daily maxima storm surges obtained with a data-driven model (Tadesse and Wahl, 2021) is also available at tide gauge sites using five different atmospheric reanalyses as forcing fields (Table 1). The European Marine

Observation and Data Network (EMODnet) also provides via its unified Portal, open and free access to integrated and harmonised data from tide gauges (including EuroGOOS platforms), together with specific data products for SLR, including sea level trends (relative and absolute) and relative sea level anomalies. Such societally-relevant data layers are also made publicly available via the European Atlas of the Seas, an EC Communication tool to support public awareness and ocean literacy.




European Copernicus Services also provide sea level data, with a free and open access policy (Melet et al., 2021). Altimeter sea level products are operationally produced and distributed by the Copernicus Marine Environment Monitoring Service (CMEMS) and by the Copernicus Climate Change Service (C3S) (Legeais et al., 2021), and used to produce Ocean Monitoring Indicators[1], such as observed mean SLR for the global ocean and regional EU seas, as well as regional sea level trends. In

addition, the CMEMS provides tide gauge data, ocean and wave forecasts and reanalyses (Irazoqui Apecechea et al., 2023).

**Table 1: Overview of publicly available data portals and online visualization tools of SL data. Please note that this is a non-exhaustive overview, providing entry points to data, with a focus on the IPCC and European-based organizations.**

| Data source/Organisation | Access Link | Contents |
|---|---|---|
| IPCC AR5 WG1 Chapter 13 (Church et al., 2013) | https://www.cen.uni-hamburg.de/en/icdc/data/ocean/ar5-slr.html | IPCC AR5 SL projections viewer & data portal (global) |
| IPCC AR6 WG1 report | https://interactive-atlas.ipcc.ch | IPCC AR6 climate observations and projections viewer & data portal (global) |
| IPCC AR6 WG1 Chapter 9 (Fox-Kemper et al., 2021) | https://sealevel.nasa.gov/ipcc-ar6-sea-level-projection-tool | IPCC AR6 SL projections viewer & data portal (global) |
| PSMSL | https://psmsl.org/ | SL observations (tide gauges) data portal (global) |
| INSeaPTION | http://www.inseaption.eu/index.php/news/23-web-map-of-sea-level-projections | SL projections viewer (global) |
| UK Met Office | https://climate.metoffice.cloud/sea_level.html | SL observations viewer (global) |

---

[1] https://marine.copernicus.eu/access-data/ocean-monitoring-indicators?f%5B0%5D=omi_family:438





| GSSR database (Tadesse and Wahl, 2021) | http://gssr.info/ | Observed daily maxima storm surges (global) |
|---|---|---|
| Copernicus Climate Data Store | https://cds.climate.copernicus.eu/#!/home | Various data sets on observed and projected sea levels and extreme sea levels (Europe) |
| Copernicus Marine | https://marine.copernicus.eu/ | Satellite altimetry, tide gauge records, ocean and wave reanalyses, ocean and wave forecasts (including for sea level). Vizualisation tool (global and Europe) |
| EMODnet | https://emodnet.ec.europa.eu/en | SL observations data portal (Europe) |
| European Atlas of the Seas | https://emodnet.ec.europa.eu/en/eu_atlas_of_the_seas | SL observations data portal (Europe) |
| Norwegian Mapping Authority | https://www.kartverket.no/en/at-sea/se-havniva/sea-level/se-havniva-i-kart--a-web-tool-for-mapping-storm-surge-and-sea-level-rise | Observed and projected sea level change viewer (Norway) |
| Joint Research Centre | Joint Research Centre Data Catalogue - Large_Scale_Integrated_Sea-level_and_Coastal_Assessment_Tool_-_European Commission (europa.eu) | Modelled historical and projected extreme sea levels |
| SONEL | Système d'Observation du Niveau des Eaux Littorales (SONEL) | Observed sea level trends from tide gauges and VLM trends derived from GNSS at tide gauge sites. |

*********** END OF BOX 1 ***********



## 3 Regional observations: past mean trends, extreme value intensification

### 3.1. Tide gauge record

Centennial changes in SL are largely based on tide-gauge observations. The European coastlines are home to many of the longest tide gauge records worldwide (Marcos et al., 2021; Raicich, 2020; P. L. Woodworth and Blackman, 2004; Wöppelmann et al., 2014; Wöppelmann and Marcos, 2016; Figure 2). Tide gauges measure sea level changes relative to the coastal point where they are installed. This implies that they observe the oceanic component of sea level (termed as absolute sea level) together with VLM driven by a variety of mechanisms (Section 3.3). To account for VLM, tide gauge measurements are often complemented with VLM measurements (Global Navigation Satellite System (GNSS)) to separate the ocean-related and solid Earth processes from sea level records (Wöppelmann and Marcos, 2016).

Tide gauges are installed and operated by national and subnational agencies and also by research institutions, each of which provide access to sea level records with a variety of formats, sampling frequencies and quality checks. User access is facilitated by original data providers, or data assembly centres and initiatives, including those in the framework of the Global Sea Level Observing System (GLOSS), the Copernicus Marine Service or EMODnet Physics, among others (Table 1). Low frequency global mean SL records at tide gauges at monthly and yearly frequencies are obtained by national providers and compiled and distributed by the GLOSS Permanent Service for Mean Sea Level (www.psmsl.org) (Holgate et al., 2013). A total of 595 tide gauge records are available along the European coasts on the PSMSL website, of which 55 span a period longer than 100 years (Figure 2). In addition to homogenised tide gauge data sets, the data base contains other historical records which provide valuable information on long-term SL changes, such as Amsterdam or Stockholm (see https://psmsl.org/data/longrecords/) (Figure 3a). For studies related to extreme events or storminess, high-frequency sea level observations are required. The Global Extreme Sea Level Analysis dataset (www.gesla.org), currently in its version 3, contains a global set of hourly and higher sampling tide gauge observations (Haigh et al., 2022; Figure 3b). High-frequency records are needed for ESLs and to capture high-frequency processes contributing to sea level changes at the coast such as seiches, meteo-tsunamis and infragravity waves Vilibić and Šepić, 2017).

In Europe, these observations can be obtained from the Copernicus Marine Service (https://marine.copernicus.eu/), from EMODnet Physics (https://emodnet.ec.europa.eu/en/physics) and from national and subnational agencies (see the GESLA website for more details on data providers). Different data portals may distribute repeated stations although with different metadata, convention names or ID, and distinct levels of processing. An intercomparison of available tide gauge portals is provided by SONEL (https://www.sonel.org/tgcat/). As an example of the database contents, there are a total of 1072 tide gauge stations of at least hourly sampling along the European coasts in the GESLA data base with a median length of 15 years and of which 48 span a period longer than 100 years, providing essential information (Figure 2).






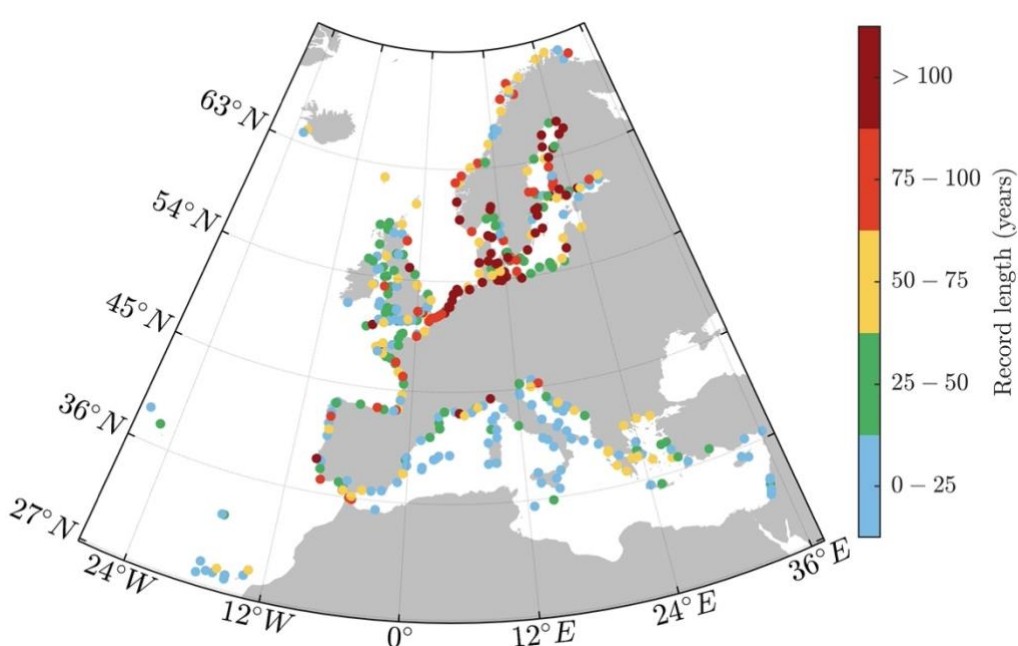

**Figure 2: Location of tide gauges in Europe from the PSMSL database with length of records in years (https://www.eea.europa.eu/legal/copyright). Copyright holder: European Environment Agency (EEA).**



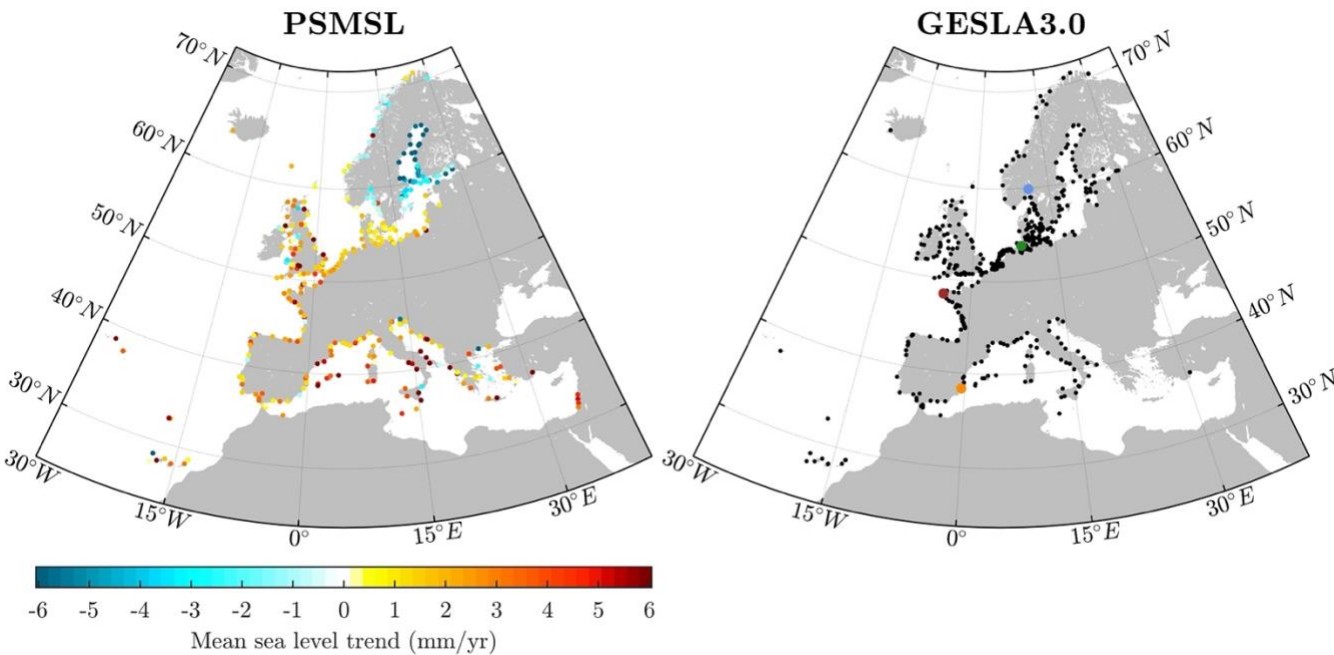

**Figure 3: (a) Relative sea level trends at PSMSL European tide gauges. Note that the tide gauge records are covering different periods (Figure 2). (b) Location of GESLA European tide gauges (top right). Colored dots in panel (b) indicate the location for which return level curves of storm surges are shown in Figure 4: blue for Oslo, green for Cuxhaven, purple for Brest, orange for Alicante.**





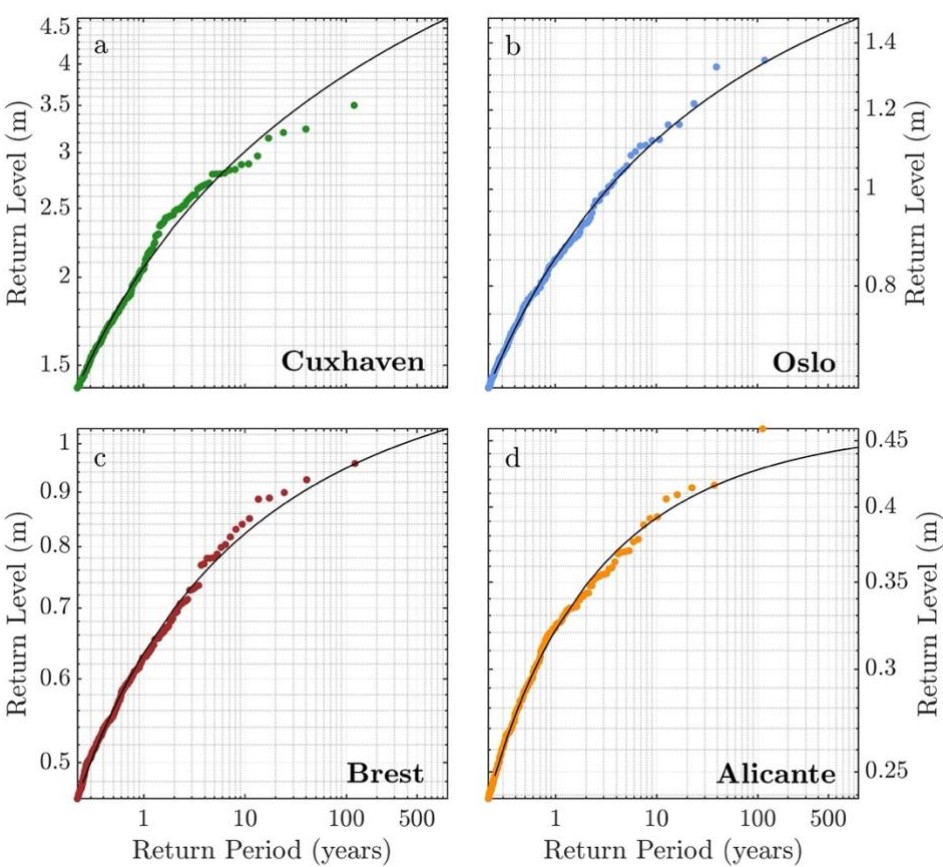

**Figure 4: Return level curves of storm surges at four selected GESLA tide gauges from different European basins shown in Figure**
**3b. Tide gauge records were detided (using Utide Matlab software) and extremes were selected as peaks over the 95th percentile of**
**each time series, with events separated by at least 3 days to ensure independence.  Return levels have then been calculated fitting a**
**Generalised Pareto Distribution to each record.**

### 3.2. Satellite record

While tide gauges provide point-wise, long-term relative sea levels (relative to the local land surface to which they are

grounded), altimetry measurements provide shorter, but spatially coherent and quasi-global measurements of absolute sea

levels (relative to a reference ellipsoid). Satellite altimetry measures sea level from space with a radar emitter to measure the

distance between the satellite and the sea surface and precise positioning instruments to measure the position of the spacecraft.

Satellite altimeters allow to measure the geocentric sea level, which is the sea level with respect to the centre of mass of the





Earth. Since 1993, sea level has been monitored routinely on a daily basis with a resolution of ¼°x ¼° from 82°S to 82°N (e.g.,

(Legeais et al., 2021). Although sea level dynamics is highly heterogeneous, the time and space samplings are enough to

resolve effectively the GMSL dynamics on a weekly basis (Fox-Kemper et al., 2021a; Henry et al., 2014; Scharffenberg and

Stammer, 2019).

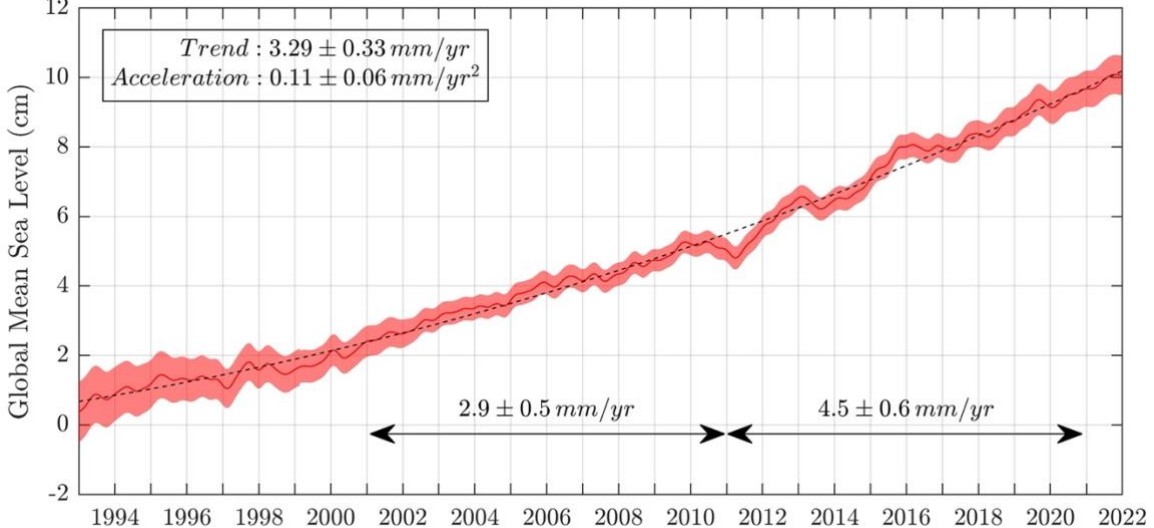

**Figure 5: Global mean sea level measured by satellite altimetry since 1993 (red curve), shaded area represents the uncertainty and the dotted line shows a trend line with an acceleration. Over 1993-2022, the GMSLR trend is 3.29±0.33 mm yr-1 and the GMSLR acceleration is 0.11±0.06 mm yr-2. Data source: Copernicus Marine Service (CMEMS) Ocean Monitoring Indicator based on the C3S altimetric sea level product. Credit: C3S/ECMWF/CMEMS.**

Since 1993 GMSL has risen by 3.3±0.3mm yr[-1] which represents a total increase in sea level of 10 cm (Figure 5). Over 1993-

2018, 46% of GMSLR is attributed to the ocean thermal expansion, 19% to melting mountain glaciers, 15% to land ice mass

loss from the Greenland ice sheet and 9% from the Antarctic ice sheet (Fox-Kemper et al., 2021). The remaining 11% is

attributed to changes in land water storage such as dam building, groundwater pumping and aquifer recharge/discharge

(Cazenave and Moreira, 2022; WCRP Global Sea Level Budget Group, 2018). The satellite altimetry sea level record also

shows an acceleration of 0.11±0.6 mm yr[-2] (Guérou et al., 2022). This acceleration is mostly due to an acceleration of ice mass

loss from Greenland, and to a lesser extent to an acceleration of the contribution from glacier melting and ocean warming

(Frederikse et al., 2020b). Studies have shown that present-day GMSLR cannot be explained by internal climate variability

and mostly results from anthropogenic forcing (Fasullo and Nerem, 2018; Marcos et al., 2015a, 2017; Richter et al., 2020;

Slangen et al., 2016). In Europe, sea level trends are slightly above the global mean rate, on average (Figure 6). Only a few

areas, such as central parts of the Mediterranean Sea, show no change or a slight decrease in absolute sea level. On interannual



time scales the global mean SL record shows significant variations which are mostly generated by El Nino Southern Oscillation

events during which the cloud pattern and the rain pattern are changed leading to changes in thermal expansion and land water

storage respectively (e.g., Cazenave et al., 2014; Hamlington et al., 2020).

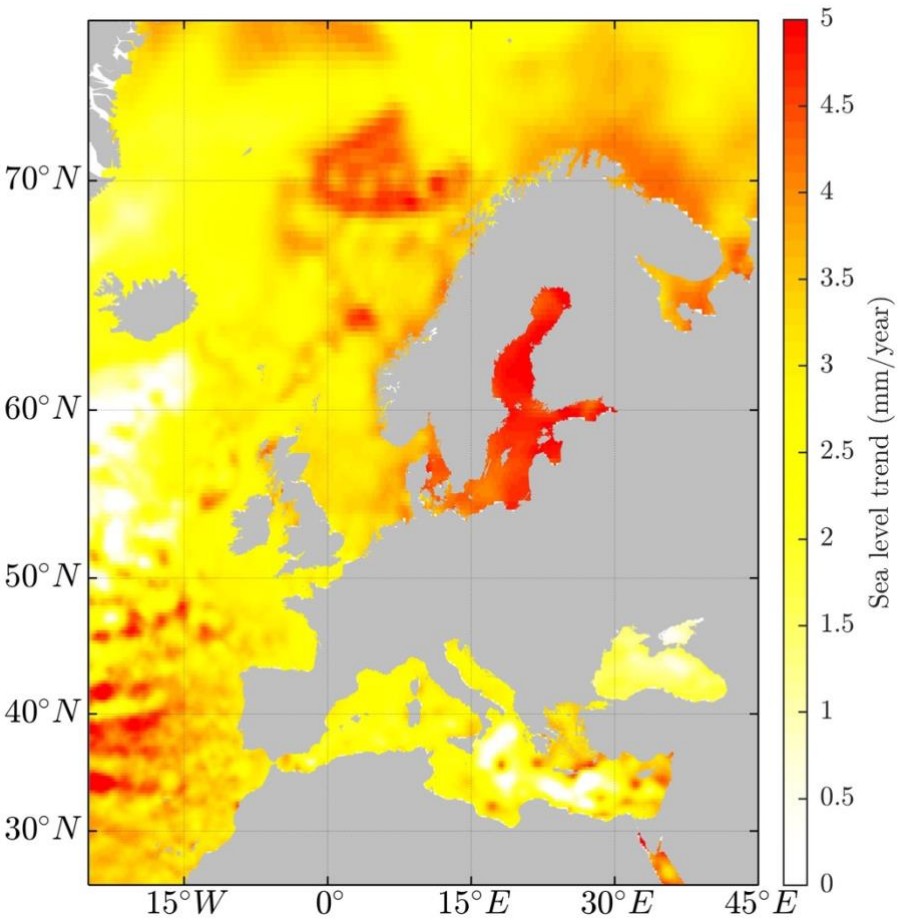

**Figure 6: Absolute sea level trends (mm/year) from January 1993 to July 2021. The data have not been adjusted for GIA nor for the**
**TOPEX-A instrumental drift. Data source: CMEMS Ocean Monitoring Indicator based on the C3S sea level product. Credit:**
**C3S/ECMWF/CMEMS. Absolute sea level does not account for VLM, which is described in Section 3.3. The trend of GMSLR**
**observed by altimetry over the same period, with no GIA correction and with the seasonal cycle removed, is 3.20 mm yr⁻¹ (Source:**
**AVISO).**





At regional scales, the ¼° x ¼° daily gridded satellite altimetry maps have an effective temporal resolution of around 34 days (spatially varying), and an effective spatial resolution ranging from 100 to 200 km in the north-eastern Atlantic, and from 90 to 160 km in the Mediterranean and Black Seas (Ballarotta et al., 2019). Satellite altimetry shows that the rate of sea level change is spatially highly heterogeneous. The dominant contribution to the regional sea level trend patterns is the non-uniform thermal expansion caused by the redistribution of heat within the ocean in response to the wind-forced ocean circulation, and

direct exchange of heat between the lower atmosphere and the upper ocean (Forget and Ponte, 2015; Meyssignac et al., 2017). The spatial trend patterns in sea level are not stationary, in particular in the North Atlantic, where positive trends shift to negative trends from the first to the second decade of the spatial altimetry sea level record, and the other way around (e.g., (Chafik et al., 2019). This is because spatial trend patterns in sea level remain so far mostly driven by the internal climate modes, such as the North Atlantic Oscillation (NAO) in the Atlantic (see Section 6.1).   Strong differences in SL trends at the

sub-basin scale are also recognized in the Mediterranean (Bonaduce et al., 2016; Mohamed et al., 2019; Skliris et al., 2018), in which variability and complexity arise from changes in the general circulation (Mauri et al., 2019; Meli et al., 2023; Menna et al., 2019, 2021, see Section 6.4.1). At interannual time scales, the sea level variability in the Atlantic Ocean is also driven by changes in the NAO climate mode (Boucharel et al., 2023; Roberts et al., 2016; Wakelin et al., 2003). This is also true for sea level extremes (see Sections 4.3 and 6.1).

Near the coast, the altimeter-based sea level variations and associated trends are more uncertain than the measurements retrieved in the open ocean (e.g., Birol et al., 2017; Cipollini et al., 2017; Vignudelli et al., 2019). This is due to local factors, such as the distortion of the altimeter radar echo by coastal features, the higher uncertainties of some altimeter corrections (e.g., ocean tides), other local processes that are not captured by satellites (e.g., how far waves wash up the shore), and the spatial resolution of the satellite data. Although more uncertain, recent estimate of the sea level at the coast (e.g., Birol et al.,

2021) show a general agreement between SLR on European coast and closest SLR in the open ocean in terms of trends and interannual variability, at least when getting as close as a few kilometers to the coast (The Climate Change Initiative Coastal Sea Level Team et al., 2020).

Satellite altimetry has also been used to study the annual, semiannual and interannual cycles in sea level (e.g., Fernández-Montblanc et al., 2020), including in relative sea level changes (Ray et al., 2021), which is of relevance for coastal flooding.

Along the coasts of Europe, the annual cycle of absolute sea level is characterized by annual maxima during the autumn (except for the Black Sea, where the annual maxima are reached in Spring). The annual cycle amplitude ranges from 40 cm to 100 cm with the largest amplitude found in the North Sea, Baltic Sea, along the Arctic coast of Norway and in the western Mediterranean Sea (Fernández-Montblanc et al., 2020; Ray et al., 2021). Based on altimetric data, it has been shown that the monthly mean SL (including SLR) contribution to ESLs at the coast is mostly larger than that of tides and of the same order





of magnitude as that of storm surges in microtidal areas (Black Sea, Baltic and Mediterranean Sea) (Fernández-Montblanc et al., 2020).

### 3.3. Vertical land motion including human induced subsidence

VLM is an important component of relative sea level change along Europe's coasts as measured by tide gauges. It encompasses all processes leading to a vertical change in the land surface such as GIA due to short- and long-term ice mass loss, tectonics,

volcanism, and subsidence owing to groundwater or hydrocarbon withdrawal or sediment compaction. These physical processes operate on different spatial and time scales and can be related to climate change, human activities, or natural processes. Several techniques can be used to measure VLM. Historically, repeat levelling has been the main technique and gives a relative measure of VLM. Repeat levelling has been extensively used in parts of Europe to help constrain VLM, for example, in Scandinavia and the Baltic countries where uplift rates are large (Vestøl et al., 2019). Levelling is also used to

measure differences in VLM between global navigation satellite system (GNSS) stations and tide gauges.

Permanent GNSS stations provide a continuous and very accurate (uncertainties smaller than 1 mm/year) measure of VLM in the terrestrial reference frame. GNSS thereby gives a high-quality pointwise measurement of VLM but lacks information in areas between stations, where station spacing is typically of several 10s of kilometres. There are several thousand GNSS stations in Europe which are operated in national or regional networks, and are owned by e.g., national agencies, research

institutions, and private companies (Figure 7a). Efforts to bring together GNSS data and products from across Europe are available from the EUREF permanent network (www.epncb.oma.be), and from the European Plate Observing System (www.epos-eu.org) research infrastructure. Several analyses have focused on combining European GNSS data (e.g., Kenyeres et al., 2019) with some of those interpolating VLM values for potential use along the coast (e.g., Hammond et al., 2021; Piña-Valdés et al., 2022). For users interested in tide gauges, the International GNSS Service has a program for analysing GNSS

data from stations near or co-located with tide gauges. These data are currently hosted at Système d'Observation du Niveau des Eaux Littorales (www.sonel.org)(GLOSS data portal for GNSS data at tide gauges). GNSS stations co-located at tide gauges are important for understanding the contribution of VLM to relative sea level (RSL) (Woodworth et al., 2017).

Finally, a more recent technique is interferometric synthetic aperture radar (InSAR), which uses satellite radar to measure VLM with millimetre accuracy. InSAR can image the pattern of VLM and has very good spatial coverage, allowing users to

detect local areas of land movement. For example, InSAR has been used in Venice to measure land subsidence (e.g., Teatini et al., 2012). Integrating InSAR and GNSS data can maximise the advantages of both techniques. InSAR data products are newly available from the European Ground Motion Service (www.egms.land.copernicus.eu), which uses Copernicus Sentinel-1 radar images.





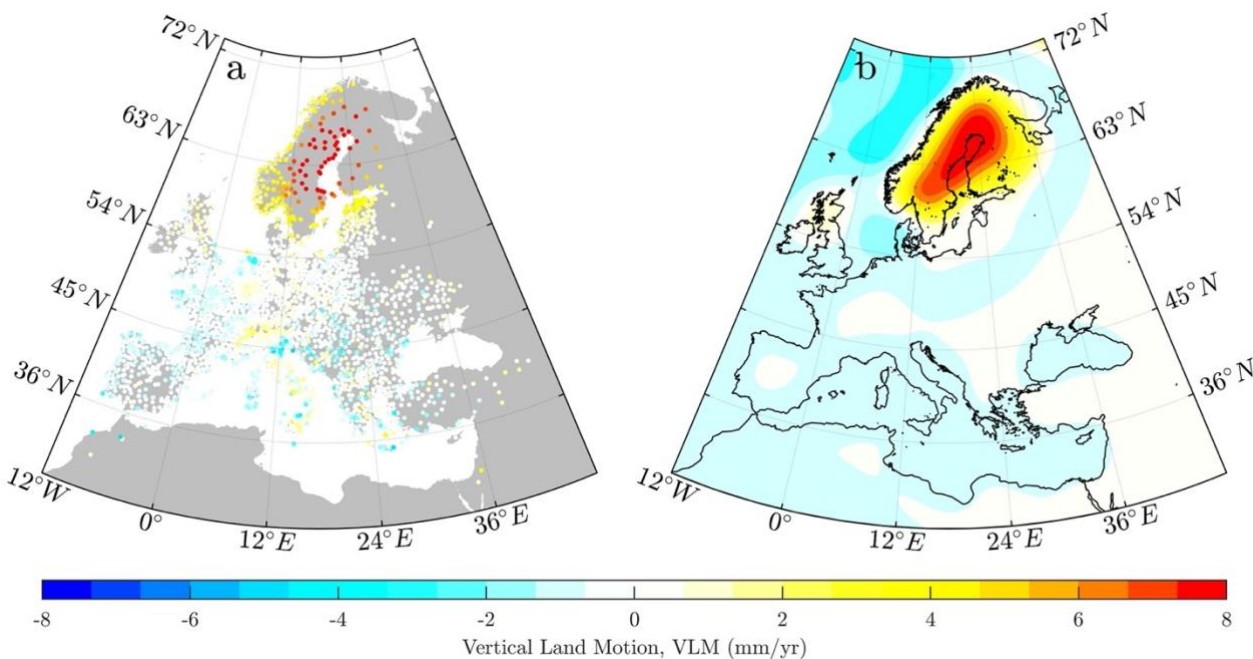

Vertical Land Motion, VLM (mm/yr)

**Figure 7: Left – preferred filtered and smoothed present day VLM field from Piña-Valdés et al. (2022) and based on data from ~4000 GNSS stations in Europe. Right – The present day VLM from the GIA inversion model from Caron et al. (2018). Units in mm/year.**

The broad pattern of land motion in Europe can be seen from GNSS measurements (Figure 7). Note that on local scales VLM can deviate significantly from this picture. In European cities like Antwerp, Rotterdam and Venice, for example, there are complex patterns of localized subsidence (see e.g., www.egms.land.copernicus.eu and (Wu et al., 2022)).

There are several distinct features in the broad VLM field: In northern Europe a dome pattern of uplift due to GIA is clearly visible related to the long-term contribution of unloading since the last ice age. On century time scales we assume this as a constant rate. GNSS observations show a maximum uplift of ~10 mm yr$^{-1}$ in northern Sweden and rates of subsidence exceeding 1 mm yr$^{-1}$ in northern central Europe. Rates of highest uplift correspond to where ice was thickest during the past glacial. GIA is the dominant driver of regional VLM in many parts of northern Europe (notably the North Sea, European Artic, and Baltic basins) (e.g., Kierulf et al., 2021; Milne et al., 2001; Teferle et al., 2009). In those regions the order of magnitude equals the climate driven increase in sea level. GIA also largely explains the broad pattern of differences in RSL rise in this region.





Land areas adjacent to the Atlantic basin (France and Spain) have generally low rates of VLM. In Southern Europe, the GNSS
VLM field shows uplift in the Alps. Around the Mediterranean and Black Sea basin there are (volcano)-tectonic deformations
in Italy, the Balkans, and Greece, causing a large variability in VLM that reflects in different RSL trends.

### 3.4. Past changes in coastal extreme sea levels

Observations of coastal ESLs rely on high-frequency tide gauge records. In Europe there is a relatively large number of high-
quality, long-term tide gauge records with hourly or higher sampling (Haigh et al., 2022, Section 3.1, Figures 2, 3) that has
been used to extensively characterize the magnitude and frequency of ESLs as well as their temporal variability (e.g., Marcos
and Woodworth, 2017; Figure 4).  Long tide gauge records demonstrate that mean SL change is a major driver of changes in
ESLs (Ferrarin et al., 2022; Weisse et al., 2014; Woodworth et al., 2011). However, variability in storm surges unrelated to
mean sea level has also been identified from observations at interannual and decadal timescales (Jänicke et al., 2021; Marcos
et al., 2015a; Mudersbach et al., 2013; Weisse et al., 2014). In addition, Calafat et al. (2022) has determined that long-term
trends in storm surges due to a combination of forced changes and internal variability along the European Atlantic coasts have
had a contribution comparable to that of mean SLR on changes of ESLs since 1960.  Changes in tides have also been evidenced.
Although generally small, contemporary past changes in tides were substantial in e.g., the German Bight (e.g., Haigh et al.,
2020). There is also evidence for changes in wave regimes over the past decades notably related to changes in the surface
winds in response to climate modes of variability and climate change (e.g., Dodet et al., 2019 for a review, see also Section 6
for European Seas), but past trends in wind-waves characteristics are associated with uncertainties due to the sensitivity of
processing techniques, inadequate spatial distribution of observations, and homogeneity issues in available records (Fox-
Kemper et al., 2021). Past changes in wind-wave regimes imply changes in wave setup and runup (e.g. Melet et al., 2018).

### 4    Drivers of sea level rise and extremes

### 4.1. The role of Antarctica and Greenland

Ice loss from the Antarctic and Greenland ice sheets contributes to SLR (e.g., The IMBIE team, 2018; The IMBIE Team,
2020). There is high agreement that for both Antarctica and Greenland, the rates of mass loss and relative contributions to SLR
have increased substantially since the 1990s (Otosaka et al., 2022; Figure 8a). As a consequence, the total mass loss of glaciers
and ice sheets has become the dominant term in the SL budget since 2006 (Fox-Kemper et al., 2021). Ice loss reduces the mass
of the ice sheets, thereby reducing their gravitational pull which causes a relative lowering of ambient sea levels and a relative
heightening of far-away sea levels. This means that ice loss in Antarctica raises sea levels in Europe proportionally more than
GMSLR while ice loss in Greenland raises sea levels proportionally less than GMSLR over Europe. This effect is more
pronounced in northern European coasts (Bamber and Riva, 2010; Grinsted et al., 2015) because that is closer to Greenland.
This process is usually referred to as the gravitational attraction. Currently, the regional sea level contribution for Greenland



and Antarctica are similar because the absolute contribution from Greenland is larger than the absolute contribution from Antarctica. If the contribution from Greenland increases in the future, this has not very large consequences for European coasts, but an increase caused by mass loss in Antarctica is amplified around the European coast.

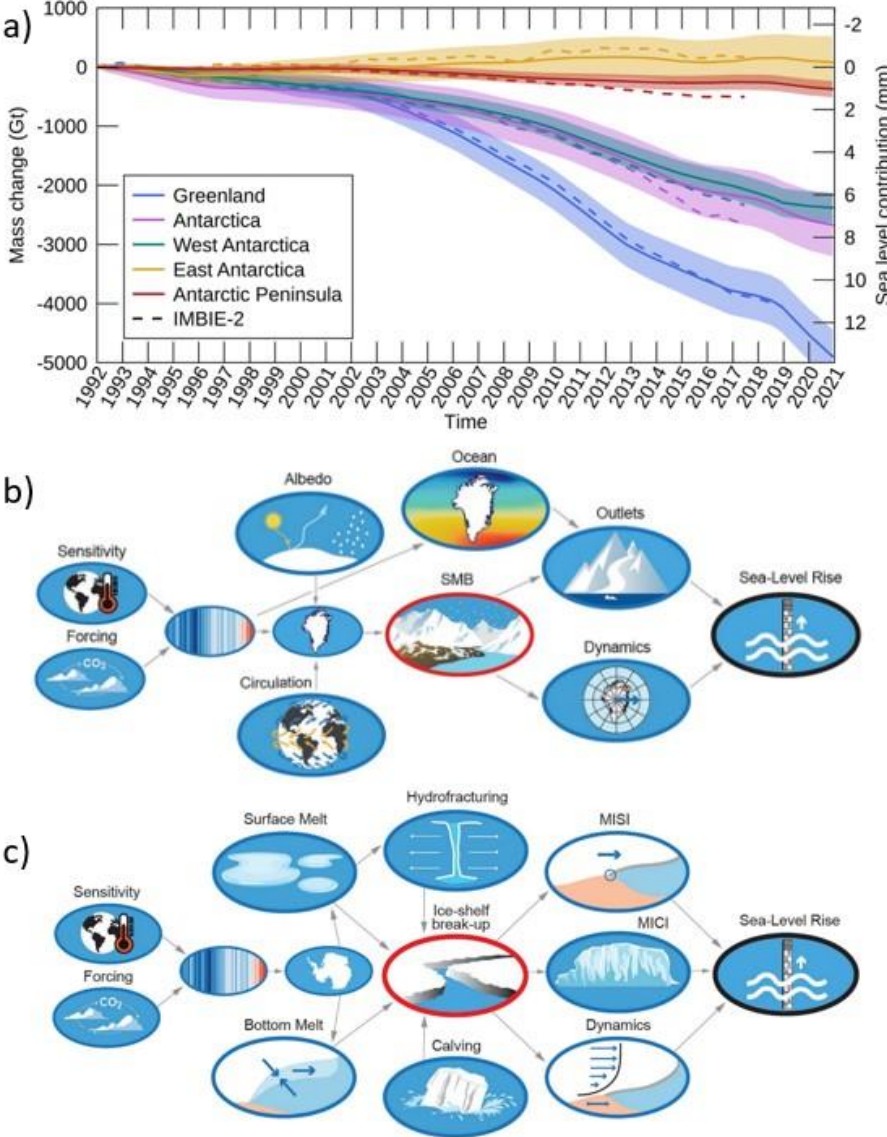



**Figure 8: (a) Greenland and Antarctic Ice Sheet mass changes. Credit: reproduced from Fig. 4 of Otosaka et al., 2022. (b) processes that influence Greenland's contribution to SLR and its future uncertainty. Credit: reproduced from Fig. 2 of Van De Wal et al., 2023, c) processes that influence Antarctica's contribution to SLR and its future uncertainty. Credit: reproduced from Fig. 3 of Van De Wal et al., 2023.**

The contribution of the Greenland ice sheet comes from dynamic changes at the margins (Smith et al., 2020), likely caused by
changes in the ocean (Holland et al., 2008; Straneo and Heimbach, 2013), as well as a reduction in the surface mass balance
due to atmospheric warming (Hanna et al., 2021) which causes increasing surface melt and runoff (Slater et al., 2021), see Fig.
2.8b. The latter was estimated to make about 60% of the mass loss (Van Den Broeke et al., 2016) and both processes are
expected to remain relevant over the coming decades (Choi et al., 2021). Projections of mass loss of the Greenland ice sheet
until 2100 show a clear greenhouse gas-emission dependency with higher levels of warming leading to higher SL contributions
(Goelzer et al., 2020). Uncertainties in atmospheric changes are directly reflected in Greenland projections: higher warming
in CMIP6 models in comparison to CMIP5 yield higher mass loss in Greenland due to surface melt (Payne et al., 2021).
Ocean-driven processes in Greenland projections are still highly parameterised (Slater et al., 2020). Uncertain is also the role
of the atmospheric dynamics. Increased surface melt was found during atmospheric blocking events (e.g.,Fettweis et al., 2013),
which are however not captured in CMIP5 models (Hanna et al., 2018) and CMIP6 models (Delhasse et al., 2021). As such,
van de Wal and others (Van De Wal et al., 2023) added a factor two for the possibility that the current CMIP models
underestimate the change in the atmospheric dynamics in their estimate of the high-end contribution to sea level change caused
by loss of ice in Greenland. Other sources of uncertainty in the contribution from Greenland to SLR are related to the sensitivity
of regional climate models used for downscaling GCM results and the calculation of the surface albedo. Finally, there is an
uncertainty related to the downscaling of GCM results with regional climate models. The chain of processes causing mass loss
in Greenland is outlined in Figure 8b.

Antarctica's current mass loss (Rignot et al., 2019; Smith et al., 2020) can be linked to the thinning and enhanced calving of
its surrounding ice shelves (Greene et al., 2022; Gudmundsson et al., 2019) driven by warmer ocean water temperatures, see
Figure 2.8c. In near future projections, a further mass loss due to ocean-driven melting is expected to be counteracted by
increased surface accumulation (Seroussi et al., 2020). Both processes were found to increase with increasing global forcing,
suppressing a scenario dependence of the Antarctic future SL contribution until 2100 (Edwards et al., 2021). The climate
forcing of this century will cause SLR on the longer term, and the contrast between lower and higher scenarios will emerge
increasingly clearly at longer time scales. Fox-Kemper et al. (2021) assess the Antarctic contribution to SLR to be –0.14 –
+0.78 m for the RCP2.6 (low emission scenario) versus –0.28 – +3.13 m for RCP8.5 (very high emission scenario) in 2300.
Critical for the timing of the accelerated mass loss of the Antarctic ice sheet is the timing of the collapse or weakening of ice
shelves. As long as the major ice shelves are in place, ice mass loss is limited, but the rate of mass loss can increase strongly
if ice shelves loose there buttressing force as enhanced ice discharge will then lead to an acceleration of SLR. For these reasons





the Antarctic projections constitute the largest source of uncertainty in SL projections (e.g., Fox-Kemper et al., 2021). The loss of ice shelves is controlled by atmospheric, oceanographic and glaciological conditions.

Currently global circulation models that are used to provide the ocean and atmosphere forcing for the ice sheet models in

projections do not include the processes on the continental shelf and in ice shelf cavities, which are ultimately determining the changes in sub-shelf melting below the ice shelves and thereby the dynamic mass loss of Antarctica. The sensitivity of melt rates to ocean temperature changes are highly uncertain but explain differences between LARMIP-2 (Levermann et al., 2020) and ISMIP6 projections (Reese et al., 2020; Seroussi et al., 2020). Recent projections with coupled atmosphere-ocean- and ice sheet models for Antarctica also show a wide range of results (Park et al., 2023; Siahaan et al., 2022). In the future, ice shelves

might also become increasingly vulnerable to atmosphere-driven melting (e.g., van Wessem et al., 2023). It may induce hydrofracturing of the ice shelves. For some shelves atmospheric process will be dominate and for others oceanographic controlled processes.

In addition, ice shelf collapse with subsequent self-sustaining ice cliff collapse has been suggested in DeConto and Pollard (2016). This instability is not yet convincingly demonstrated at present and the importance of this process is debated. A recent

paper discussed for example that the presence of ice mélange (a mix of icebergs and sea ice) could suppress this instability (Bassis et al., 2021; Schlemm et al., 2022). In the recent IPCC, it is treated as "deep uncertainty" (Kopp et al., 2023). The representation of ice shelf calving and damage in the ice constitutes a further uncertainty. Furthermore, uncertainty in processes related to basal sliding may also lead to large rates of ice mass loss (Sun et al., 2020). Hill et al. (2021) report that for the Filchner-Ronne Ice Shelf the range of possible parameters for atmospheric and oceanic changes yield a larger uncertainty than

numerical model parameters. An unrealistic upper limit on the ice sheet response due to complete loss of ice shelves is provided by Sun et al. (2020). Problematic in projections for the ice sheets is that the initial state of the ice sheet is poorly constrained (e.g., Aschwanden et al., 2021) and that the Antarctic Ice Sheet has the potential to cross critical thresholds, which cause irreversible ice loss (see Section 5.2) and amplify uncertainty in SL projections (Robel et al., 2019). For this reason, the low confidence (Fox-Kemper et al., 2021; Kopp et al., 2023) or high-end scenarios (Van De Wal et al., 2023) were developed.

**4.2. Sea level budget**

On decadal to multi-millennial time scales, global mean SL changes are essentially caused by changes in the Earth energy budget. Since the end of the 19[th] century, the increase of greenhouse gas concentrations from anthropogenic emissions modified the Earth energy budget such that the amount of outgoing radiation amount is less than the amount of incoming solar radiation, leading to global warming. Oceans absorbed 90% of global warming leading to seawater expansion and SLR. 3% of the excess

heat is absorbed by the cryosphere causing the melting of land ice, such as glaciers and ice sheets, which contributes to SLR.



Changes in terrestrial water storage, which are the changes in the amount of water stored on land, such as groundwater or water stored in lakes and rivers, also contribute to SLR. Part of the changes in terrestrial water storage are related to the energy budget and the climate variability through the changes in rain patterns that change the amount of water stored in areas such as lakes and rivers. Part of the changes in terrestrial water storage are related to direct anthropogenic activity (such as groundwater

depletion, dam building etc.) and thus are independent from the global energy budget.

Sea level budget analysis over the past century, based on development and application of new statistical methodologies for reconstructing GMSL (e.g., Dangendorf et al., 2019; Frederikse, Buchanan, et al., 2020) and its contributions (e.g., Bagnell and DeVries, 2020; Frederikse, Landerer, et al., 2020; Zanna et al., 2019), suggests that the primary factors contributing to

GMSLR over 1901-2018 is the mass loss of glaciers (41±15%), the thermal expansion of seawater due to global warming (38±10%) and the Greenland ice sheet mass loss (25±8%). The contribution of Antarctic ice sheets mass loss is relatively small (4±6%) over this period. The contribution of land water storage is largely uncertain, but it is likely to contribute to a sea level fall over 1901-2018 up to -8±20% (Fox-Kemper et al., 2021). Since the late 1960s, all contributions to SLR accelerated (Dangendorf et al., 2019). Over the period 1971-2018 the sea level budget is consistent with the global energy inventory of the

climate system which gives high confidence on the GMSL budget over this period (Fox-Kemper et al., 2021).

Over a more recent period, since 2006, GMSL has been monitored by satellite altimetry, thermal expansion of the ocean by Argo floats and the change in ocean mass by space gravimetry. Consequently, the sea level budget is significantly more precise and the closure more accurate. Over 2006-2018, the sea level budget is closed with an uncertainty of a few mm (Figure 9). Over 2006-2018, the primary factors contributing to SLR are the thermal expansion of seawater due to global warming (39%)

and the Greenland ice sheet mass loss (17%). The melting of glaciers represents 17% of GMSL rise over this period while the Antarctica contribution rose to 10%. Land water storage changes explain the remaining 17% (Fox-Kemper et al., 2021, Table 9.5). Percentages are based on central estimate contributions compared to the central estimate of the sum of contributions.

Since 2018, the GMSL budget derived from altimeter data has not closed anymore within conventional uncertainty thresholds (Barnoud et al., 2021, Figure 9). A cause for this non-closure is a drift in the wet tropospheric correction (which is the correction

for the path delay in the radar altimeter due to the water vapor content in the atmospheric column) of Jason-3 altimeter, and a drift in Argo salinity sensors. After correction of Jason-3 spurious drift and after using only thermosteric SLR (and not steric SLR) the non-closure is reduced by 40% but it remains larger than the uncertainty in the components of the sea level budget from 2019 on (Barnoud et al., 2023). More research is needed to understand the causes for the residual non-closure of the sea level budget over the past few years.






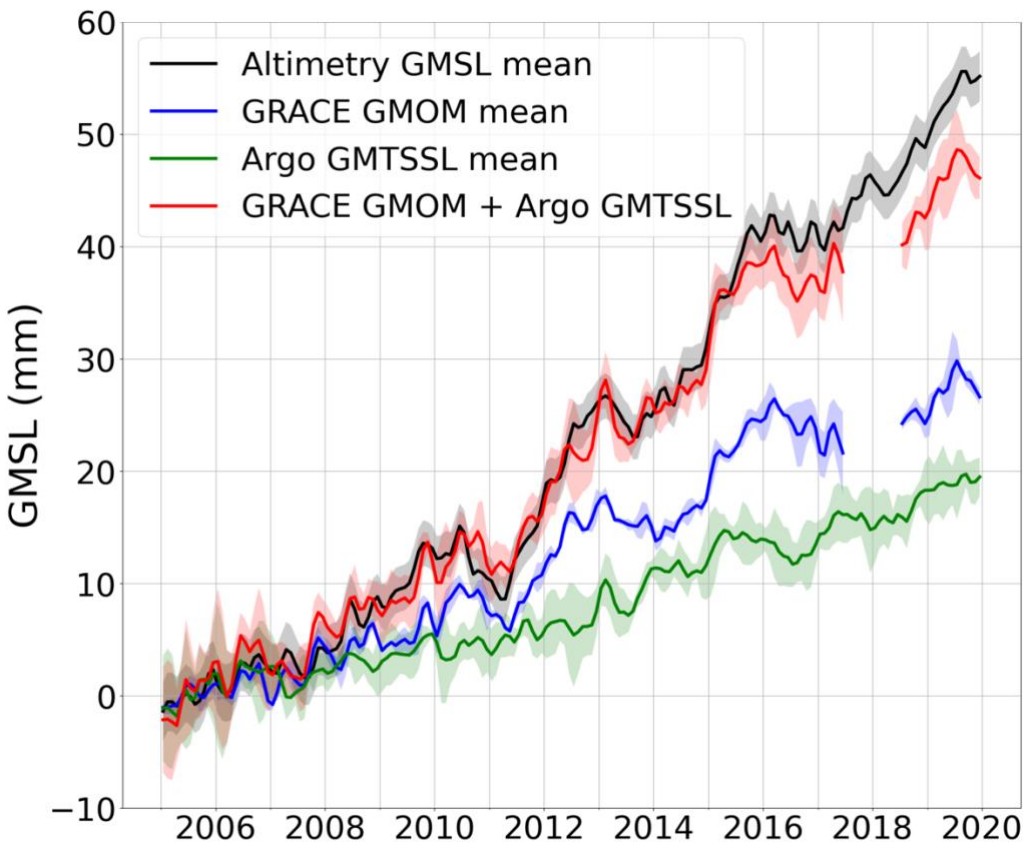

**Figure 9: GMSL budget from 2006 to 2021. GMSL estimated by satellite altimetry (black curve, data from the Copernicus Marine Environment Monitoring Service). Ocean mass change (sum of ice sheet mass loss, glaciers ice melt and land water storage changes) estimated from GRACE and GRACE-FO (blue curve, data taken from the JPL, CSR, GSFC mascon solutions). Ocean thermal**
**expansion estimated from Argo (Green curve, data taken from an ensemble of the NOAA, EN4, SCRIPPS and JAMSTEC Argo product). From Barnoud et al., 2021.**

At regional scale, closing the SL budget is more challenging due to the higher variance of the signal, although there have been some attempts to explore the closure from local (Royston et al., 2020) to large basin-wide scales (Purkey et al., 2014). Instead, Camargo et al. (2023) used unsupervised machine learning techniques to identify regions of coherent sea level variability.
Compared to previous studies, which looked at the regional budget on entire ocean basins, this study identified more and smaller domains. These domains reflect large-scale climate patterns, such as ENSO in the Pacific and NAO in the Atlantic, and highlight which ocean regions are connected through physical processes, such as propagating coastally trapped waves from Iberia to the north-western European shelf (Calafat et al., 2012, 2014; Hughes et al., 2019). While in gridded data (for





instance a 1x1 rectangular grid) the sea level budget cannot be closed everywhere, the budget can be closed in almost all

domains identified by self-organising maps when all contributions, including estimates for deep steric changes, are accounted

for and with a residual error of only 0.6 mm yr$^{-1}$ for the period 1993-2016. Regional SL budget has also been closed along

coastal regions of coherent variability (Dangendorf et al., 2021), also showing that most of the interannual changes are linked

to dynamic sea level variability. For example, in the North Sea, observations corrected for VLM display a linear trend of 2.01

(95% confidence intervals of 1.30-2.76) mm yr$^{-1}$ over 1960-2012, while the sum of the sterodynamic and barystatic

contributions is 2.09 (1.58-2.52) mm yr$^{-1}$ (Dangendorf et al., 2021).

### 4.3. Drivers of extreme sea levels

Coastal ESLs result from the combined action of mean sea level changes, astronomical tides and atmospheric pressure and

surface winds that generate storm surges and wind-waves. Higher-frequency processes such as coastal waves, meteo-tsunamis

and seiches (in semi-enclosed basins such as the Adriatic Sea) can also contribute to ESLs at the coast. Thus, sea level extremes

are short-term phenomena (time scale of min/hours to a day) triggered by atmospheric perturbations and tides, but they are

also modulated by long-term changes in mean sea level and by low-frequency variability in storminess (associated to changes

in frequency, tracks and/or severity of weather systems; Woodworth et al., 2019). Return levels for extreme storm surges are

shown in Figure 4 for selected European locations.

Changes in mean sea level affect ESLs in several ways: variation in water levels modify the baseline level upon which extremes

reach the coastline and at the same time, changes in mean sea level interact with other coastal sea level contributors like tides,

surges and waves (Idier et al., 2019b), for instance through velocity and friction effects over tides and storm surges due to

depth changes in coastal shallow waters.  Along many European coastlines, astronomical tides are an important component of

ESLs and in some regions tide-surge interactions take place, such as in the English Channel (Haigh et al., 2010; Idier et al.,

2012), the UK coastline (Horsburgh and Wilson, 2007) and the North Sea (Arns et al., 2020; Wolf, 1981). Changes in mean

sea level also affect tidal propagation in the same way as storm surges. As these processes take place at subregional spatial

scales, further research is needed to explore future changes and the resulting impacts on the coasts (see Section 5.3).

Changes in tides (Section 3.4) and storminess can also drive changes in ESLs at the coast. Besides storm surges, wind-waves

are also a driver of coastal hazards, especially when they co-occur with storm surge extremes. Wind-waves in the nearshore

contribute to coastal ESLs through transfer of momentum due to wave breaking (the so-called wave setup effect) and the wave

uprush on a beach or structure (the so-called wave run-up) (Dodet et al., 2019). The magnitude of the wave contribution is

locally variable and often difficult to assess from observations, as tide gauges are placed in sheltered areas to avoid instrumental

failures. The effect of wind-waves on sea level extremes over broad spatial scales has therefore been assessed mostly using



parametric approaches (Melet et al., 2018; Vousdoukas et al., 2017). Coupled hydrodynamic and wind-wave models are available, but they need a very high spatial resolution in coastal areas to properly represent wave setup, thus limiting their
applicability (Roland et al., 2009).

Despite the relatively good coverage of tide gauges along European coastlines (Section 3.1, Figures 2, 3), their location is sparse and sea level records are often incomplete, thus providing only a partial picture of the spatial and temporal footprints of coastal ESLs. One way to overcome the limitation of the observational network is to simulate sea level extremes using either numerical models or data-driven approaches. Hydrodynamic models, the most common to simulate storm surges, use the
shallow water equations to simulate the response of the ocean to atmospheric pressure and surface winds. Ocean general circulation models can also be used for this purpose (e.g., as in Copernicus Marine regional forecasting systems, Irazoqui Apecechea et al., 2023), as they explicitly resolve tides and storm surges, although at higher computational expenses. Model accuracy depends essentially on the available forcing fields and on the model setup, including the spatial resolution of the coastal bathymetry and the coastline. Computational needs and data availability of this type of models are currently one of the
main limitations to increase the spatial resolution. For example, available bathymetric data in the German Bight is coarse and inconsistent, and affected by morphodynamic changes at interannual time scales. Hydrodynamic model runs are available at global and European scales spanning several decades, thus allowing to explore seasonal variability and long-term trends in storm surges (e.g., Fernández-Montblanc et al., 2020; Muis et al., 2020).

Alternatively, data-driven approaches rely on establishing a statistical relationship between observed ESLs and a set of
predictors from atmospheric and/or oceanic variables. These data-driven approaches can be, in some places, more accurate than hydrodynamic models and require less computational resources (Tadesse et al., 2020). These alternative methods, however, are site dependent and may not be reliable to reproduce an event that is beyond the observational records. Quantitative information on coastal ESLs derived from data-driven approaches or models, in the form of simulated or reconstructed time series, are available online along all European coastlines (see Sections 3, 5.3, 6).

**5   Projections of sea level rise and extremes on global and regional scale**

**5.1. 21st century projections**

Projections of future SL change can be computed using global climate model information for the ocean density and dynamics, in combination with dedicated model simulations for the contributions from ice sheets (Section 5.1), glaciers, land water storage change and VLM (Section 3.3). The latest IPCC AR6 report provided 21st century SL projections for five different
emission scenarios (Fox-Kemper et al., 2021: SSP1-1.9 ('very low'), SSP1-2.6 ('low'), SSP2-4.5 ('intermediate'), SSP3-7.0 ('high') and SSP5-8.5 ('very high') (Figure 1)). These projections include all processes that could be assessed with at least



*medium confidence,* thereby excluding ice sheet processes associated with deep uncertainty as discussed in Section 4.1 (see also Section 5.2). In addition, low confidence projections (Fox-Kemper et al., 2021) and high-end projections (Van De Wal et al., 2023) were developed, reflecting the *deep uncertainty* associated with the contribution of the Antarctic and Greenland ice

sheets.

The *medium confidence* regional IPCC AR6 projections (Figure 10) show that some European coasts are projected to experience a RSLR by 2100 below the projected GMSLR, such as the Norwegian coast, the Baltic Sea, the Northern part of the UK and Ireland, and the northern coasts in the Mediterranean Basin. Other coasts also show projected RSLR above the global mean, for instance the Atlantic coasts of Portugal, Spain, France, Belgium and the Netherlands.  For semi-enclosed

basins, the projections can be improved by replacing the ocean density and dynamics component from the IPCC projections by high-resolution regional model results which capture the local dynamics and exchange with the ocean basins in much more details. More regional information on SLR projections is provided in Section 6 per European sea basin.

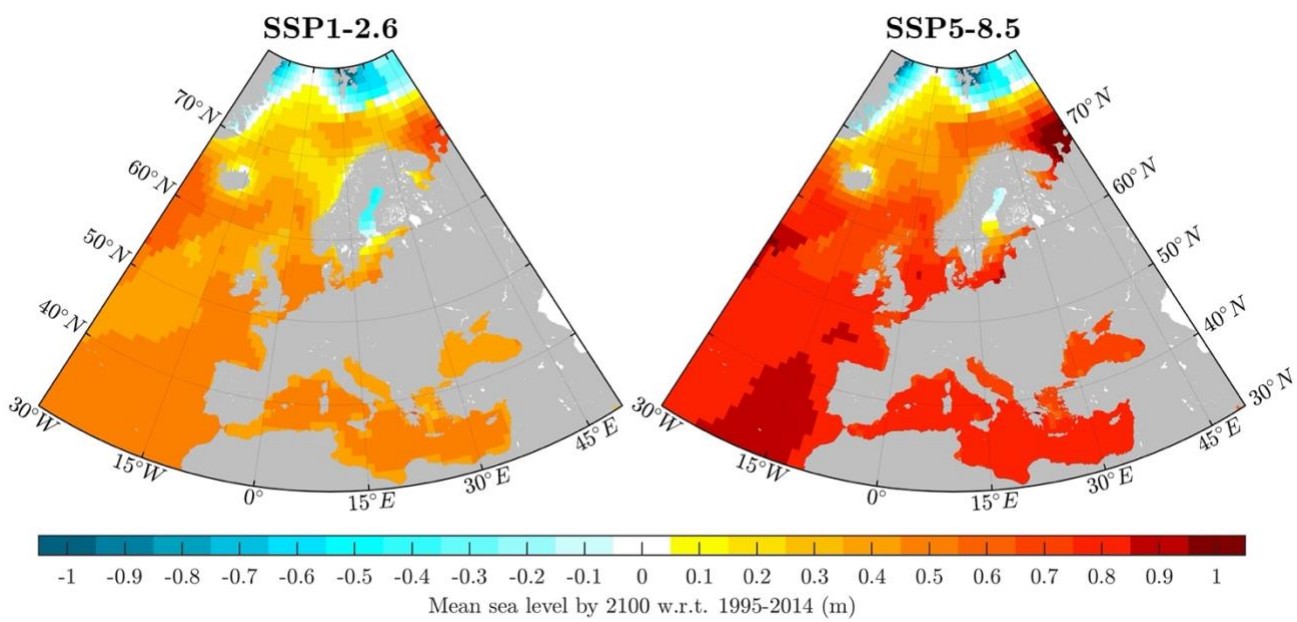




**Figure 10: Median relative sea level regional projections (*medium confidence*) from the IPCC AR6 report around Europe under a) SSP1-2.6 and b) SSP5-8.5 in 2100 with respect to 1995-2014 (m) (IPCC AR6 projection data available from https://doi.org/10.5281/zenodo.5914709).**

A new addition to the AR6 report, compared to previous IPCC reports, is the inclusion of SL projections stratified by warming
level (Fox-Kemper et al., 2021; their Section 9.6.3.4). As SLR is mostly a product of time-integrated warming, rather than
instantaneous warming (e.g., Bouttes et al., 2013; Hermans et al., 2021; Kuhlbrodt and Gregory, 2012; A. Melet and
Meyssignac, 2015), it is important to specify the timing of the peak warming. The AR6 projections (Table 2) are based on a
Global Mean Surface Air Temperature (GSAT) increase of 1.5, 2, 3, 4 and 5 degrees by 2081-2100, but do not specify the
route to this temperature increase. The differences in the pathways and their effect on the projected SLR are reflected in the
uncertainties of the temperature-level projections (Table 2).

**Table 2: GMSL projections (m) for exceedance of five global warming levels, defined by sorting GSAT in 2081-2100 wrt. 1850-1900. Median values and likely range in 2050 and 2100 relative to a 1995-2014 baseline (Fox-Kemper et al., 2021). Data for the temperature pathways are available at https://zenodo.org/record/6382554#.ZFjOx3bP3ao.**

| GMSLR (m) | 1.5 deg | 2.0 deg | 3.0 deg | 4.0 deg | 5.0 deg |
|---|---|---|---|---|---|
| Total (2050) | 0.18 (0.16-0.24) | 0.20 (0.17-0.26) | 0.21 (0.18-0.27) | 0.22 (0.19-0.28) | 0.25 (0.22-0.31) |
| Total (2100) | 0.44 (0.34-0.59) | 0.51 (0.40-0.69) | 0.61 (0.50-0.81) | 0.70 (0.58-0.92) | 0.81 (0.69-1.05) |

For decision making around SLR, it may be useful to ask 'when' a certain SLR threshold will be crossed (Slangen et al., 2022),
as this provides an indication of the time left to prepare adaptive and protective measures for that specific threshold. Figure 11
regional SLR projections, indicating the first decade in which the median projected SL change has crossed a certain threshold
(0.5, 0.75, 1.0 m above the 1995-2014 baseline) under two emissions scenarios. A lower emissions scenario (Figure 11, left
column) typically leads to slower SLR, which in turn leads to a later crossing of thresholds, whereas high emissions scenarios
(Figure 11, right column) have a faster SLR and therefore earlier crossing of thresholds. For thresholds crossed before 2050
there is little dependence on the emission scenario used.







**Figure 11: The first year of the decade (between 2020 and 2150) when the median regional SL projections around Europe have crossed a threshold since 1995-2014 of 0.5 m (a,b), 0.75 m (c,d) and 1.0 m (e,f), under SSP1-2.6 emissions (left column) and SSP5-8.5 (right column). Dark blue indicates no crossing before 2150. Based on** *medium confidence* **IPCC AR6 SL projections (Fox-Kemper et al., 2021; IPCC AR6 projection data available from https://doi.org/10.5281/zenodo.5914709).**





## 5.2. Tipping points, irreversibility and commitment

Greenland and the West and East Antarctic ice sheets are considered tipping elements in the climate system (Armstrong McKay
et al., 2022; Lenton et al., 2008). In this case, tipping is understood as crossing a critical threshold beyond which ice loss
becomes irreversible on human time scales, i.e, the relevant climate forcing (regional oceanic and atmospheric conditions)
would need to be reduced substantially below the pre-tipping value to halt or reverse the retreat of the ice sheet.

The tipping behaviour of the Greenland ice sheet is linked to the melt-elevation feedback, where the ice-sheet surface lowering
brings the ice surface into regions of higher surface air temperatures which causes more melting and thereby further surface
lowering (Levermann and Winkelmann, 2016; Weertman, 1961). The Greenland ice sheet was confirmed to exert a tipping
behaviour in Robinson et al. (2012), however in other model simulations, e.g., of a coupled ice and atmosphere general
circulation model (Gregory et al., 2020), only the northern part of the ice sheet, corresponding to 2 m of SL equivalent, was
found to behave irreversible. In some cases, examining statistical properties indicate whether the system is close to a tipping
point. Boers and Rypdal (2021) suggest that based on surface melt reconstructions the central-western Greenland ice sheet is
close to a critical transition.  Importantly, the timescale of tipping depends on the strength of the forcing scenario. A nearly
complete disappearance of the Greenland ice sheet might still take millennia if the threshold is marginally crossed (Robinson
et al., 2012), which would imply that rates of SLR are still modest.

A widely accepted mechanism for tipping in Antarctica is the Marine Ice Sheet Instability (MISI) (Schoof, 2007; Weertman,
1974), where the ice sheet retreats rapidly in marine parts of the ice sheet. Since stability conditions for MISI are more
complicated than only a retrograde slope for realistic Antarctic conditions (Gudmundsson, 2013; Haseloff and Sergienko,
2018; Pegler, 2018; Sergienko and Wingham, 2022), numerical modelling is required to identify these tipping points. Studies
suggest tipping behaviour for the glaciers (e.g., Thwaites and Pine Island) in the Amundsen Sea (Favier et al., 2014; Rosier et
al., 2021) and West Antarctica (3 m of SL equivalent, SLE, Feldmann and Levermann, 2015). The East Antarctic ice sheet
contains marine basins that can also show tipping, such as the Wilkes basin (19 m SLE) (Mengel and Levermann, 2014). Also,
Aurora basin (3.5 m SLE) which is a large marine base area, has been suggested to have the potential of tipping. Whether MISI
is already ongoing is still a matter of debate (Favier et al., 2014; Joughin et al., 2014; Rignot et al., 2014). Garbe et al. (2020)
report that retreat of West Antarctic grounding lines could be initiated by around $1 - 2$ °C of global warming above pre-
industrial. Golledge et al. (2021) also find that in a simulation coming from the last interglacial, the West Antarctic Ice Sheet
starts retreating after 1500 years with constant current climate conditions.  So, the general idea is that West-Antarctica is
unstable for high forcing scenarios (Oppenheimer et al., 2019), but our insights are not detailed enough to indicate where the





threshold is in detail. Importantly, the timescales of tipping also depend on the strength of the forcing scenario, parameter choices and physical choices made in model set up. Feldmann and Levermann (2015), Golledge et al. (2019) find that a collapse of the Antarctic ice sheet takes millennia for temperature values close to the threshold, while the unrealistic ice shelf removal in ABUMIP (Sun et al., 2020) causes the West Antarctic Ice Sheet to collapse within a few centuries.

An alternative tipping mechanism has been suggested by DeConto and Pollard (2016) based on ice shelf disintegration followed by Marine Ice Cliff-Instability (MICI) yielding even more rapid rates of mass loss. The importance, timescales and mechanism of this process are debated (e.g., Bassis et al., 2021) and it is for this reason classified as "deep uncertainty" in the latest IPCC report (Fox-Kemper et al., 2021). The importance of both MISI and MICI strongly depends on the extent to which the ice shelves retain their buttressing force to keep the ice sheet in place. Timing of collapse or thinning of the major ice

shelves is not foreseen in the 21st century, but DeConto et al. (2021) suggest that increased mass loss due to shelf collapse starts to play a role around 2100 with consequences for enhanced SLR in the early 21st century.

Crossing tipping points would mean an irreversible commitment to SLR unless a rapid temperature decrease is materialized (Bochow et al., 2023). SL commitment is, however, not only due to crossing of tipping points, but also because ice sheets respond on long timescales and climate forcing might be hard to reverse. The contribution of the Greenland ice sheet in 2100

that has already been committed through past climate change has been estimated to be around 3.3 cm SLR (Nias et al., 2023). Climate change during this century will commit ice loss over the coming centuries to millennia even without further climate change.

Furthermore, there is a long-term SLR commitment from the ocean, through the key role it plays for uptaking heat from the atmosphere and the consequently induced thermal expansion (e.g., Bouttes et al., 2013). The efficiency of ocean heat uptake

(OHU, the temporal rate of change of the ocean heat content) depends on how quickly heat gained at the ocean surface is transported to depth. The faster heat is mixed to the deep ocean, the less the surface air temperature warms as more excess heat is taken up by the ocean and the lower the transient climate response (Krasting et al., 2018; Marshall and Zanna, 2014). If emissions of GHG were to stop, the radiative forcing of GHG that were previously released in the atmosphere would remain quasi-constant with a slow decay over centuries (e.g., Ehlert et al., 2017; Zickfeld et al., 2017). As a result, the global mean

surface air temperature would remain quasi-constant. The upper ocean temperature, which exchanges heat with the atmosphere, tracks the radiative forcing and would thus equilibrate. On the other hand, the deep ocean, which is coupled to the upper ocean through mixing, would continue to warm and to export heat to deeper layers (e.g., Bouttes et al., 2013; Dalan et al., 2005; Ehlert et al., 2017; Melet et al., 2022). Although the ocean heat uptake would decline over time, the large thermal inertia of the deep ocean and the long timescales of its adjustment would result in a net warming of the ocean and related steric SLR that





is largely irreversible for at least a millennium after emissions stop (e.g., Zickfeld et al., 2017).  Only on very long time scales

the deep ocean may release this energy again.

## 5.3. Projected changes in extremes

Projections of future changes in ESLs generally either only include the effect of an increase in the mean SLR on the baseline

height of extremes, assuming that the distribution of ESLs is stationary, or also include non-stationarity in extremes due to

changes in storm surges, tides and/or waves based on numerical modeling (Section 4.3).

### 5.3.1.    Projected changes in extremes

Projections of future changes in ESLs due to SLR are often reported as the amplification of the probability of a certain ESL

(Buchanan et al., 2016; Fox-Kemper et al., 2021; Frederikse, Landerer, et al., 2020; Hermans et al., 2023; Jevrejeva et al.,

2023; Lambert et al., 2020; Oppenheimer et al., 2019; Rasmussen et al., 2018; Tebaldi et al., 2021; Wahl et al., 2017) or as the

height by which coastal defenses need to be raised to restore the historical flood probability (allowances;  Hunter, 2012; Hunter

et al., 2013; Slangen et al., 2017; Woodworth et al., 2021). For instance, the IPCC AR6 (Fox-Kemper et al., 2021) projected

that the sea level associated with the historical centennial event (HCE), which is the event that historically had a 1% chance of

occurring each year (once per century on average), will be exceeded at least annually at 19-31% of 634 tide gauges worldwide

in 2050, and at 60-82% in 2100. In Europe, the largest amplification factors of the frequency of ESLs are projected for the

south (Mediterranean and Iberian Peninsula coasts), whereas in the northeast of the United Kingdom and in the southeastern

North Sea, amplifications are generally smaller because the current variability of ESLs is larger (Figure 12). Amplifications

of the HCE are below one, implying a decreasing probability of the HCE, in the northern Baltic Sea because of the land uplift

anticipated for that region associated to GIA (Sections 3.3 and 6.5). The spatial pattern in Figure 12 is a robust feature across

different studies (e.g., Fox-Kemper et al., 2021; Frederikse, Landerer, et al., 2020; Oppenheimer et al., 2019).






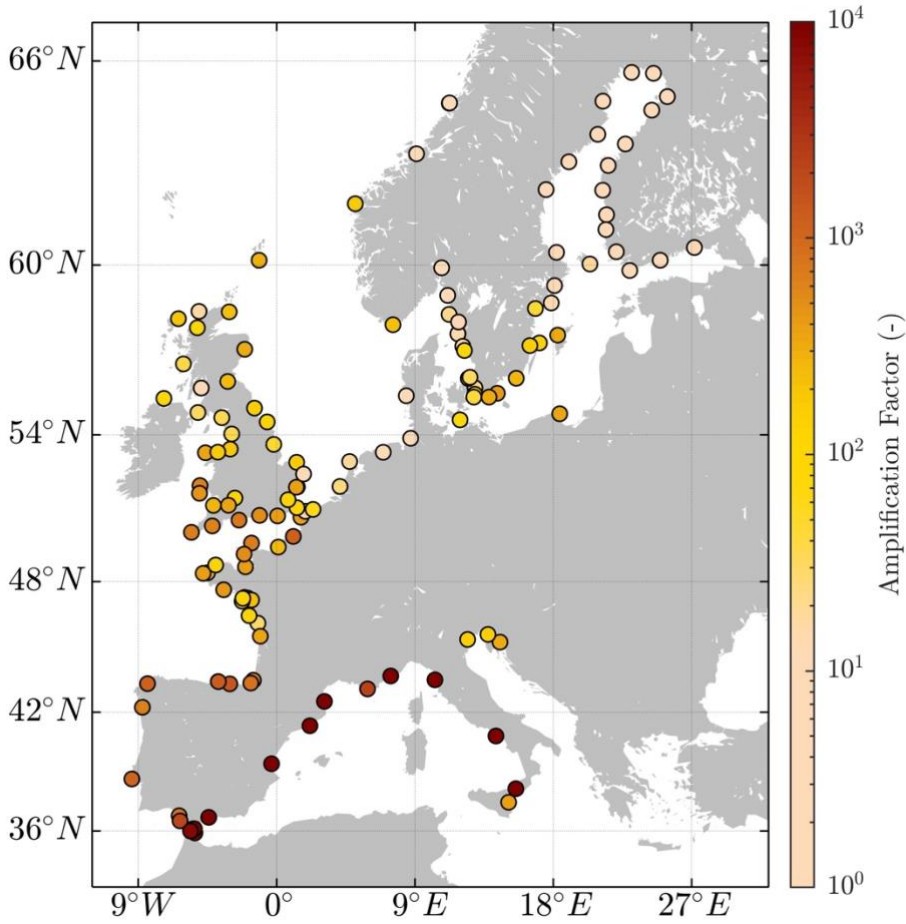

**Figure 12: Amplification factors of the historical centennial sea level event in 2100 projected by the IPCC AR6, for Europe, under the SSP2-4.5 middle-of-the-road emission scenario (obtained from Fig. 9.32 of Fox-Kemper et al., 2021).**

Projected amplification factors in most of the studies mentioned above are derived by combining inferences of the historical ESL distribution with projected relative SLR, incorporating the uncertainty in both and assuming that the historical extremes distribution remains the same (so-called static or mean SL offset method). Projections of amplification factors are therefore sensitive to the type of extreme value distribution used, and to the threshold above which events are defined as extreme (Buchanan et al., 2016; Wahl et al., 2017). A Generalized Pareto Distribution with the 99[th] percentile as a threshold was identified to be the preferred approach to assess ESLs at a global scale (Wahl et al., 2017). Acknowledging that the same threshold is not appropriate at all locations, two recent studies implemented an automatic threshold selection (Hermans et al., 2023; Lambert et al., 2020), which substantially affected their results in specific locations. To characterize the events below





the threshold, different approximations have been used such as a Gumbel distribution between mean higher high water and the extremes threshold (Buchanan et al., 2016; Fox-Kemper et al., 2021; Rasmussen et al., 2018), or a simple extrapolation (Hermans et al., 2023; Sweet et al., 2022). Rasmussen et al. (2022) applied an extreme value mixture model instead, but the extent to which declustering the data below the threshold is appropriate is unclear. Furthermore, since wave-sheltered tide

gauge measurements are typically used to infer the extreme sea level distributions, the effect of waves is not (fully) incorporated in the type of projections in this Section. Incorporating waves generally increases the historical range of extremes at a given location, which leads to smaller projected amplification factors (Lambert et al., 2020).

Most studies have projected the amplification of the HCE. However, information on changes in the probability of a single extreme sea level can be of limited salience locally (Rasmussen et al., 2022). For instance, the design height of local protective

infrastructure may differ from the height of the HCE, and large amplifications of the HCE do not necessarily affect a large fraction of the local population (Rasmussen et al., 2022). Projections of the population affected by changes in extremes (e.g., Haasnoot et al., 2021; Kirezci et al., 2020, 2023; Rasmussen et al., 2022) or projections of the amplification factors of specifically those ESLs that local coastal protection is designed to withstand (Hermans et al., 2023), help to add context to projections of amplification factors which facilitates translating hazards into impacts (Rasmussen et al., 2022; Sea Level Rise

in Europe: impacts and consequences). Policy-relevant information may also be provided by projecting *when* certain critical increases in the probability of ESLs may be reached instead of how much that probability will increase in 2100 (Rasmussen et al., 2022), akin to the timing of mean SLR milestones (Cooley et al., 2022; Fox-Kemper et al., 2021; Haasnoot et al., 2019; Slangen et al., 2022). Recent projections of the timing of amplification factors due to SLR indicate that the probability of ESLs that coastal flood defenses are designed to withstand will increase substantially within the time it may take to implement large

adaptation measures, also in Europe (Hermans et al., 2023).

### 5.3.2.    Projections of dynamic changes in extremes

To account for changes in the distribution of extremes, numerical models can be used to simulate changes in storm surges, tides and waves due to changes in atmospheric conditions and water depth (e.g., Figure 13). Barotropic hydrodynamic models (Section 4.3) have been used to simulate storm surges, tides and their future changes, either only as a function of atmospheric

changes simulated by regional or global climate models (Jevrejeva et al., 2023; Palmer et al., 2018; Vousdoukas et al., 2017, 2018), or also due to projected mean SLR, imposed in the model as a change in water depth (Muis et al., 2020, 2022). High-resolution baroclinic ocean models, which can simulate both changes in mean SLR and in storm surges, tides and their non-linear interactions, can provide more consistent simulations of dynamic changes in extremes. As these models are computationally more expensive than hydrodynamic models, they are often limited to a specific region (e.g., Northern Atlantic

and North Sea in Chaigneau et al., 2022; Chinese Seas in Kim et al., 2021 and Jin et al., 2021). As explained in Section 4.3,



wave contributions to ESLs and their projections can be evaluated using parameterizations based on numerical wave models outputs (Dodet et al., 2019; Kirezci et al., 2020; Lambert et al., 2020; Melet et al., 2018) but these parameterizations are limited as they are restricted to specific coastal environments, rely on the specification of a local beach slope, and are calibrated with relatively sparse historical field data (Lambert et al., 2020, 2021; Melet et al., 2020).


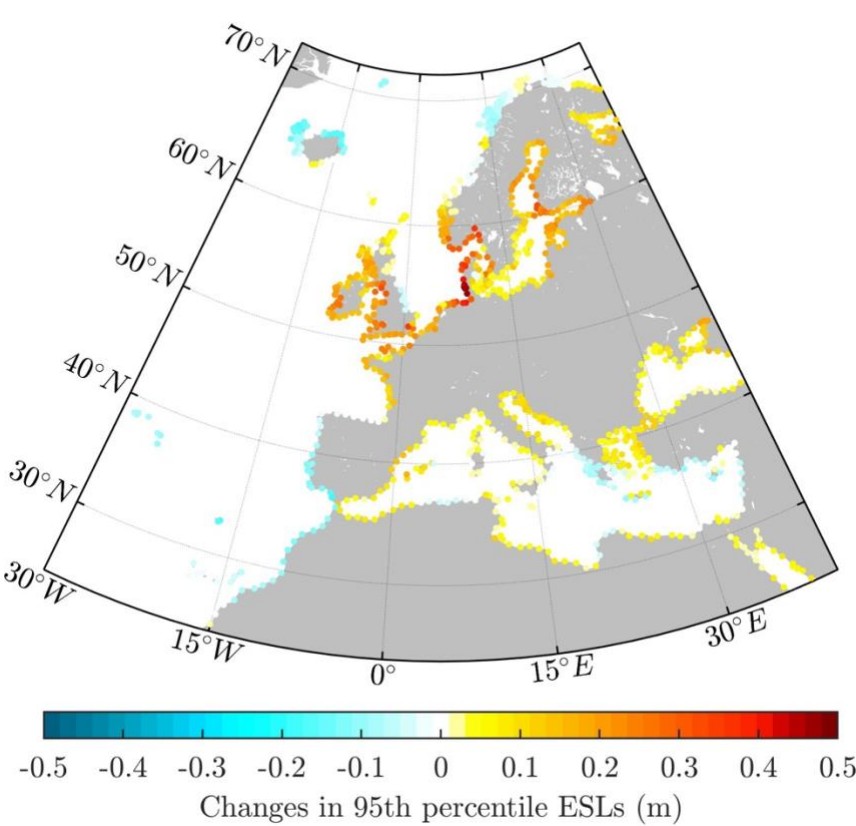

**Figure 13: Projected changes in the height of extreme sea levels associated with storm surges and waves only, with a return period of 0.01 yr$^{-1}$ (centennial event) (95$^{th}$ percentile) by 2100 relative to 1980-2014 along the European coastline (adapted from Fig. 3 of**
**Jevrejeva et al., 2023).**



Using the models described above, substantial dynamic changes in each contribution to ESLs have been projected for the European coasts, especially under the SSP5-8.5 scenario. The results are presented here for the SSP5-8.5 scenario, since this is the scenario that shows the largest projected changes and that has been the focus in most dynamic approaches in the past

years. Forcing a hydrodynamic model with atmospheric simulations from HighResMIP (Haarsma et al., 2016) models, Muis et al., (2022) projected a decrease in storm surges of up to 15% in southern Europe by mid-21st century. Around the UK, Palmer et al., (2018) and Howard et al., (2019) concluded no projected changes in storm surges due to the spread of the forcing global climate models. For the same region a strong decrease of around -10% in mean and extreme wave heights and periods (Aarnes et al., 2017; Lobeto et al., 2021b; Mentaschi et al., 2017; Meucci et al., 2020; Morim et al., 2018, 2021) resulting in a

decrease in wave setup and runup (Melet et al., 2020) is also expected by the end of the century. In the southern North Sea, Jevrejeva et al., (2023) showed an increase of +50 cm in extreme storm surges and waves under a low-probability high-impact scenario (Figure 13). In addition, non-linear interactions between sea level, surges, waves and tides, for instance through changes in water depth, can impact ESLs and their future changes in Europe (Idier et al., 2019b). For example, tidal ranges may change by up to +/-10 cm for a uniform SLR of 2 m in Europe (Haigh et al., 2020; Idier et al., 2017; Pickering et al.,

2017). Extreme significant wave heights are projected to be significantly larger (up to +40%) at the end of the century under the SSP5-8.5 scenario due to the consideration of mean SLR and tides (Arns et al., 2017; Chaigneau et al., 2023) with implications on wave setup and runup and thus on projections of ESLs. In addition, recent studies have shown, on a global scale and more specifically for Europe, that historical trends in storm surges (Calafat et al., 2022; Reinert et al., 2021; Roustan et al., 2022; Tadesse et al., 2022) and tides (Pineau-Guillou et al., 2021) have been comparable in magnitude to the historical

mean SLR trend.

The historical and projected dynamic changes in extremes and their non-linear interactions in Europe suggest that studies using a static approach may miss an important aspect of changes in ESLs (e.g., Boumis et al., 2023). However, studies using dynamic approaches concluded that generally mean SLR is the dominating driver of the projected changes in ESLs (Howard et al., 2019; Jevrejeva et al., 2023; Muis et al., 2020; Vousdoukas et al., 2018). For instance, Jevrejeva et al., (2023) concluded that

projected changes associated with storm surges and waves contribute less than 10% to the total increase in ESLs by 2100 in Europe and elsewhere. Nevertheless, these studies typically do not include projected changes in all the coastal sea level components (tides, storm surges, waves) nor their non-linear interactions and may therefore underestimate the importance of dynamic changes in extremes. Moreover, most studies projecting dynamic changes in extremes are based on small ensembles of model simulations, often for a single emissions scenario, due to the high computational cost of high-resolution

hydrodynamic or 3D ocean and wave models and the limited availability of appropriate forcing data (Jevrejeva et al., 2023; Muis et al., 2020, 2022; Vousdoukas et al., 2017, 2018). The projections may therefore not be robust due to structural differences between the different driving climate models and internal climate variability, as also suggested by Calafat et al.,

 

2022). Furthermore, the driving global climate models often have a relatively low atmospheric resolution, so cannot very well resolve historical and future cyclones.

In summary, while several studies have concluded that mean SLR is the dominant driver of changes in ESLs at most locations, including in Europe, further research is required to better quantify dynamic changes in extremes.

## 6   Key developments per region
### 6.1. Atlantic Ocean
### 6.1.1.   General context

The north-eastern Atlantic Ocean basin bordering western Europe (Portugal, Spain, France, the UK, Ireland) is characterized by strong bathymetric gradients, with a deep ocean basin, a continental shelf that is narrow along the Iberian Peninsula and that widens northward up to Ireland. This region includes parts of the North Atlantic subtropical and subpolar gyres, separated by the North Atlantic Current. A slope current flows northward along the continental slope separating the deep ocean from the continental shelf. Strong summer upwellings of deeper, colder water occur along the coasts of Portugal. On the continental

shelf ocean dynamics are characterized by shorter timescales and spatial scales, and by sea level variations of greater amplitude.

Tides on the north-eastern Atlantic continental shelf are amongst the most energetic ones worldwide, with the semidiurnal M2 component dominating. The coasts of Portugal, Spain, the Bay of Biscay, Ireland and northern UK experience a lower macrotidal regime (3.5 to 5.0 m tidal range). An upper macrotidal regime along the coasts of the English Channel, French

Brittany and southern UK reaches from 5.0 to 10 m of amplitude (e.g., Flemming, 2005).

The North Atlantic mid-latitude storm track induces large waves, swells and storm surges due to surface winds and low atmospheric pressure that directly impact western Europe. Extreme storm surges along the northeast Atlantic coasts are therefore directly related to the track location and intensity of extra-tropical cyclones. In addition, swells generated by North Atlantic extratropical storms are reaching the west EU coasts (e.g., Amores and Marcos, 2020; Bricheno and Wolf,

2018). Under the present climate, the world's highest 50-yr return period significant wave heights are found in the north-eastern Atlantic (Morim et al., 2023).

### 6.1.2.   Past sea level changes

Sea level changes along the coastline of the north-eastern Atlantic have been monitored through a rather dense network of tide gauges for decades, and up to centuries at specific locations (e.g., at Brest, France, Figure 14, or Newlyn, UK). Sea level rose





at a pace of 1.68 +/- 0.24 mm yr$^{-1}$ at Brest over 1900-2020, accelerated to 3.06 +/-0.72 mm yr$^{-1}$ over 1971-2020, and to 3.73 +/- 0.63 mm yr$^{-1}$ over 1991-2020 (PSMSL). Since 1993 and the advent of precise satellite altimetry to monitor sea level changes from space, SLR over the coasts of western EU in the north-eastern Atlantic has not largely deviated from the global mean, with most places exhibiting rates ranging between 2 and 4 mm yr$^{-1}$ (Section 3.2).

Regional SLR patterns in this region are mostly explained by ocean dynamics and by gravitational induced patterns related to
mass loss of the Greenland ice sheet and to mountain glaciers mass loss.

The regional pattern of SLR in this region can differ from one decade to another (Section 3.2), due to the large influence of the North Atlantic Oscillation (NAO). The NAO is the most prominent and recurrent pattern of large-scale atmospheric circulation variability over the mid- and high-latitudes of the northern hemisphere (e.g., Hurrell et al., 2003). Its strength and phase can be characterized by the difference in surface atmospheric pressure between the Icelandic low-pressure system and
the Azores high-pressure system. In addition to its influence on the regional pattern of sea level trends, the NAO also influences the year to year (or interannual) variability as well as sea level extremes in the north-eastern Atlantic as the variation in pressure patterns influences the strength and location of the jet stream and the path of storms across the North Atlantic. At interannual to decadal timescales, coastal sea level (as recorded by tide gauges) are highly correlated to the NAO (Calafat et al., 2012).

Regarding extremes, storm surges along the coasts of western Europe are related to extra-tropical storms under the storm track
and hitting the coasts. During positive NAO phases, the North Atlantic westerlies and storm tracks are shifted northwards. This results in increased (resp. decreased) storminess, storm surges and precipitations over northern (resp. southern) Europe (e.g., Hurrell and Deser, 2010). The maximum amplitude of surges increases from the coasts of Portugal and Spain to France and the UK in the Atlantic. The 50-yr return period level of surges characteristics of the past decades is close to 0.5 m along the coast of Portugal (Cid et al., 2016) and reaches between 1 (e.g., at Brest, France) and 2 m (e.g., at Liverpool, UK) along
the Atlantic coasts of northern Europe (Marcos and Woodworth, 2017). The median number of extreme skew surges per year also tend to be larger around the coasts of the UK and of the English Channel than of the Bay of Biscay or the Iberian Peninsula (Marcos and Woodworth, 2017). The median duration of extreme skew surge events is less than 5h in most places along the north-eastern Atlantic coasts (Marcos and Woodworth, 2017).

In many places, changes in MSL have been the dominant driver of changes in ESLs since at least 1960 (Section 4.3). As such,
both MSL changes and ESLs are modulated by the NAO in the north-eastern Atlantic at interannual timescales. The extreme value distribution of skew surges has been shown to evolve over time along the Atlantic European coasts, even when mean sea level changes are discarded (Marcos and Woodworth, 2017). According to a review of storminess over north-western Europe by (Feser et al., 2015), trends in storminess vary with the analyzed time period (see also Section 5.3).





The storm surge component of ESLs observed at tide gauge stations tends to have decreased amplitude along the north-eastern
Atlantic coast (Marcos and Woodworth, 2017). In Europe, it has recently been reported that changes in storm surges activity,
related to the NAO, has contributed just as much as MSLR to the overall change in ESLs in Europe since 1960 (Calafat et al.,
2022). The probability of extreme storm surges since 1960 has been suggested to have increased north of 52°N and decreased
south of 52°N (especially so along the coasts of Brittany and the English Channel). This is due to the compounding effect
(north of 52°N) or cancelling effect (south of 52°N) of trends in both the storm surge extremes and of regional MSL, which
make comparable contributions to the overall change in ESLs in Europe (Calafat et al., 2022).

Along the European Atlantic coast, the timing of the storm surge season is highly correlated with the NAO and the timing of
the storm atmospheric events. Extreme storm surges tend to occur earlier in the year in the south (Portugal/Spain) than in the
north (English Channel, UK). A consistent spatio-temporal shift of the timing of the storm surge season over the 2$^{nd}$ half of
the 20th century has recently been reported (Roustan et al., 2022). The storm surge season has tended to occur earlier along
the Atlantic coasts of Europe south of 50°N, at an average pace of around 5 days/decade (e.g., a 25-day shift over 1950-2000).

### 6.1.3.   21$^{st}$ century projections

Projections indicate that 21$^{st}$ century SLR along the coasts of the north-east Atlantic is expected to be close to GMSLR south
of 55°N and lower for northern UK and Ireland (Figure 10), notably due to VLM (Figures 7, 14). For instance, under the high-
emission, low-mitigation SSP5-8.5 scenario total mean SL is projected to increase by 0.77 m on global mean, 0.85 m in Cadiz
(SP), 0.73 m in Brest (FR) and 0.56 m in Tobermory (UK) in 2100 compared to 1995-2014 (Fox-Kemper et al., 2021; Garner
et al., 2021; see also Table 3).

Sterodynamic SLR, which includes global mean thermal expansion of the warming ocean, steric and dynamic sea level changes
induced by ocean circulations (Gregory et al., 2019) remains the dominant contributor to total SLR along the EU Atlantic
coast. In terms of ocean circulation changes, Chaigneau et al., (2022) suggested a projected intensification of the Portugal
Current and of the Canary Current, as well as a decline of the poleward slope current flowing from the Iberian Peninsula to
Ireland. In addition to Chaigneau et al., (2022), Hermans et al., (2020) and Gomis et al. (2016) also produced regionally
downscaled projections of sea level changes over parts of the north-eastern Atlantic.





The historical centennial climate extreme event amplitude (incl. storm surges and wave setup) is estimated to range from 1.5
m in the Gulf of Cadiz, increasing northward along the Atlantic EU coast to up to 3.0-3.5 m in the western UK coast
(Vousdoukas et al., 2017). 21$^{st}$ century projections indicate a decrease of this contribution of extreme total sea levels along the
Atlantic coast of the Iberian Peninsula, and an increase northward, with values ranging between +/- 0.3 m by 2100 under a
high-emission scenario (Vousdoukas et al., 2017). Along the coast of Portugal and in the Gulf of Cadiz, the projected reduction
in surge and wave extremes correspond to an offset of relative SLR by 20-30%.

Future changes in North Atlantic storms position and intensity will induce changes in mean and extreme wave conditions along
the western coasts of Europe. Changes in significant wave height, period, and energy flux in turn contribute to changes in
coastal flooding through overflowing or overtopping and in coastal erosion (Chapter 3).

Global and regional projections of the wave climate indicate a robust decrease in annual and seasonal mean significant wave
height, together with a decrease in the mean wave period over the north-eastern Atlantic (e.g., Bricheno and Wolf, 2018;
Lobeto et al., 2021; Morim et al., 2018). This leads to a decreased wave setup contribution to 20-yr mean SLR at the coast by
the end of the 21$^{st}$ century in this region. Along the Atlantic coast of the Iberian Peninsula, projected lower wave setup
contributes substantially to the regional departure of 20-yr mean coastal sea level changes from GMSLR (Melet et al.,
2020). Changes in wave direction are also relevant for wave impacts at the coast, and yet understudied. Indeed, impacts of
waves on the coast depend on the wave direction relative to the orientation of the shoreline. For instance, wave setup is largest
when wave direction is shore normal. A robust clockwise change in mean wave direction is projected for the Atlantic Iberian
coast (e.g., Lobeto et al., 2022; Morim et al., 2019). Extreme significant wave heights are also consistently projected to
decrease over the north-eastern Atlantic, with the largest decrease found along the Iberian coasts (Aarnes et al., 2017;
Chaigneau et al., 2023; Meucci et al., 2020; Morim et al., 2018, 2021). In an analysis of 14 stations distributed worldwide,
Lobeto et al. (2021) indicates that the stations located along the Atlantic coasts of Europe are the ones exhibiting the strongest
projected decrease in wave energy by the end of the 21$^{st}$ century under a high-emission scenario.

Non-linear interactions between the different components of extreme coastal water levels can be substantial in the north-eastern
Atlantic (e.g., Idier et al., 2019). Tides are sensitive to SLR as increased water depths will alter tidal dynamics (e.g., Haigh et
al., 2020; Idier et al., 2017; Section 4.3). The English Channel and the Irish Sea are ones of the world regions where tides
would change the most substantially in response to SLR (Haigh et al., 2020) and induced shifts in amphidromic points (Idier
et al., 2017; Pickering et al., 2017). Changes in M2 amplitude would be spatially heterogenous and might be up to 10% of the
MSLR within the next century (e.g., Palmer et al., 2018; Pickering et al., 2017; Schindelegger et al., 2018).



Wave-sea level interactions can lead to a substantial increase in significant wave heights and water levels in macro-tidal areas
of the north-eastern Atlantic during extreme events (e.g,; (Calvino et al., 2023; Chaigneau et al., 2023; Staneva et al., 2017).
In terms of coastal impacts, accounting for wave-water level interactions can increase the centennial wave setup event by
+10% at some locations and the wave energy flux by up to +40% in 2100 under a high-emission, low-mitigation scenario
(Chaigneau et al., 2023). Sea-state induced processes also modulate ESLs in the north-eastern Atlantic (Bonaduce et al., 2023).







**Figure 14. Top: Yearly mean relative sea level from the Brest tide gauge (starting in 1885) and yearly reconstructed basin average mean absolute sea level (so that GIA effects are not included) from Dangendorf et al. (2019), together with basin average projected multi-model ensemble mean relative sea level and 17th-83rd percentile uncertainties under SSP2-4.5 and SSP5-8.5 obtained from AR6 IPCC. Lower left: Linear trends of vertical land motion inferred from point-wise observations (from GNSS and from altimetry**





**and tide gauges comparisons) reconstructed using Bayesian principal component analysis and interpolated along the coastlines.**
**Lower right: 50-year return levels of extreme still water levels representative of the recent past computed using a historical regional ocean model forced by a climate model (Chaigneau et al., 2022).**

## 6.2. North Sea

### 6.2.1.     General context

The North Sea is a shallow continental shelf sea bordering France, Belgium, The Netherlands, Denmark, Norway and The United Kingdom. Due to the prevailing westerly winds over the North Sea, the ocean circulation in the North Sea is predominantly cyclonic (Sündermann and Pohlmann, 2011). The North Sea receives relatively saline and warm water from the North Atlantic Ocean from the south, through the English Channel, and from the north, through the Orkneys-Shetland section, the Shetland shelf area and the Norwegian Trench. In the east it is also connected to the Baltic Sea, from which it

receives relatively cool and fresh water. Water exits the North Sea mainly along the Norwegian coast.

Astronomical tides significantly influence the dynamics of the North Sea (Sündermann and Pohlmann, 2011) and contribute to the height of extreme water levels. The semidiurnal tides of the North Sea are driven by co-oscillation with Northern Atlantic tides and travel counterclockwise through the North Sea. As the tidal wave propagates from the deep ocean towards the shallower shelf, it is deformed by shallow water and frictional effects, resulting in overtides (having multiple periods of the

fundamental constituents) and compound tides (as linear combinations of multiple constituents). Largest tidal ranges are observed along UK east coast (Pugh, 2004), reaching spring tidal ranges of up to 3.60 m at Aberdeen and 6.20 m at Immingham (Horsburgh and Wilson, 2007). Mean tidal ranges amount to 3.40 m in the UK, 1.98 m along the Dutch west coast, 2.33 along the northern Dutch coast, and 2.82 m along the German coast (Jänicke et al., 2021). The northern and central North Sea are stratified from early summer to early autumn, but the southern North Sea has no thermocline throughout the year due to strong

tidal mixing (Sündermann and Pohlmann, 2011). Large non-linear interactions between the tidal and non-tidal components of water level have been recognised and studied for a long time, particularly in the southern North Sea. For example, Doodson, (1929) noticed a tendency for surge maxima in the Thames Estuary in the UK to occur most frequently on the rising tide; this phenomenon has been studied in depth by many authors (e.g., Horsburgh and Wilson, 2007; Prandle and Wolf, 1978; Proudman, 1955, 1957; Williams et al., 2016; Wolf, 1981). Large historic changes in tides have been observed in the North

Sea (Haigh et al., 2020; Jänicke et al., 2021; Woodworth et al., 1991).





The North Sea has a long history of severe coastal flooding, which accelerated the development in coastal flood risk management such as the 1953 flood which killed more than 2,000 people around the coastlines of the southern North Sea (Baxter, 2005; Gerritsen, 2005; McRobie et al., 2005) and the 1962 flood in the German Bight, in which more than 300 people lost their lives (Von Storch and Woth, 2008). Today, settlements along the North Sea coast are much better protected against the impacts of ESLs, relying on ongoing improvements in flood warnings and defences (van den Hurk et al., 2022).

### 6.2.2.    Past sea level changes

Based on tide gauge records, sea level averaged over the North Sea rose at a rate of 1.38 mm yr$^{-1}$ during 1958-2014 (Frederikse et al., 2016). The observed total SLR rate over this period matches well the sum of the rates of observed individual components, with the sterodynamic component contributing the most (both to the observed trend and temporal variability) (Dangendorf et al., 2021; Frederikse et al., 2016). The relative SL change measured at tide gauges in the North Sea is also influenced by VLM due to glacial isostatic adjustment, present-day ice mass loss and other processes such as tectonic activity and naturally or anthropogenically driven subsidence. Near regions that were covered by ice sheets during the last glacial maximum, such as Scandinavia and the UK, GIA causes relatively large land uplift contributing to a relative SL fall, whereas further away, in the southeastern North Sea, the land is gradually subsiding due to a collapse of the forebulge, contributing to relative SLR (Frederikse et al., 2016; Peltier et al., 2015). Based on satellite altimetry, linear SL trends estimated for 1993-2014 vary spatially over the North Sea from 1.3 to 3.9 mm yr$^{-1}$, with the highest rates found in the southeastern North Sea (Sterlini et al., 2017). Averaged over the wider Northwestern European Shelf, the SL trend seen by satellites during 1993-2022 was 3.1 mm yr$^{-1}$ (Copernicus Marine Service, Ocean Monitoring Indicator, Box 1). However, interannual to decadal SL variability has a large impact on the estimated SLR trends in the North Sea when evaluated over periods of only a few decades, especially in the southeastern North Sea where the variability is largest (Calafat and Chambers, 2013; Dangendorf et al., 2014; Gerkema and Duran-Matute, 2017; Tinker et al., 2020). Consequently, temporal SL variability is projected to continue to be the dominant source of uncertainty of SL change in the North Sea for the coming decades (Palmer et al., 2018).

Observational and model studies have shown that seasonal (Frederikse and Gerkema, 2018) and interannual to decadal SL variability (Dangendorf et al., 2014; Frederikse et al., 2016; Hermans et al., 2020; Tinker et al., 2020) in the North Sea is primarily caused by the variability in local wind and SL pressure, and to a lesser extent also by variability in buoyancy fluxes (Hermans et al., 2020). After removing part of the SL variability driven by local wind and SL pressure variability from tide gauge records, recent studies have found statistically significant accelerations of SLR in the southeastern North Sea (Dutch coast) (Keizer et al., 2022; Steffelbauer et al., 2022). At timescales of years to decades, a spatially coherent SL variability can be found along the eastern boundary of the North Atlantic Ocean, extending from the Canary Islands all the way up to the Norwegian Sea, which is thought to also affect the North Sea (Dangendorf et al., 2014, 2021; Frederikse et al., 2016). This



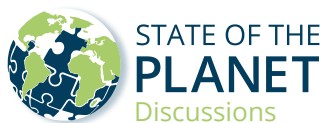

signal is thought to be caused by remote along-shore winds and the subsequent northward propagation of coastally trapped waves (Calafat et al., 2012, 2013; Dangendorf et al., 2014; Frederikse et al., 2016; Hermans et al., 2020; Hughes et al., 2019), but may also be caused by open ocean steric anomalies that follow from decadal variability in the strength of the Subpolar
North Atlantic Gyre (Chafik et al., 2019).

Trends and variability in mean sea level raise the baseline height of ESLs (Section 5.3). Furthermore, storm surges, waves and tides, which constitute ESLs in the North Sea, have also been observed to change. For instance, Calafat et al. (2022) concluded that historical trends in the height of storm surges are similar in magnitude to trends in mean sea level. They found positive trends in storm surges mainly along the north-western North Sea coastline (north-eastern UK), and negative trends along the
southern and south-eastern North Sea coasts. The changes along the English North Sea coast were mainly attributed to internal climate variability and partly to forced change associated to the strengthening and eastward extension of the North Atlantic storm track that is also projected for the 21st century (Calafat et al., 2022). Besides changes in storminess, the historically increasing water depth (due to SLR) has been shown to affect storm surges, wave height and tides non-linearly in the German Bight, with the largest changes found in the Wadden Sea due to spatially variable changes in tidal constituents (Arns et al.,
995  2015).

Changes in North Sea tides have long been identified due to non-astronomical factors in estuaries (e.g., Amin, 1983; Keller, 1901), harbors (e.g., Doodsen, 1924; Marmer, 1935; Schureman, 1934) and on the coast of the North Sea (e.g., Jänicke et al., 2021; A. Jensen, 1984; J. Jensen and Mudersbach, 2007; Mudersbach et al., 2013; P. Woodworth et al., 2017).

For the North Sea, studies have identified different contributions, covering SLR induced change in shallow water effects (Arns
et al., 2015; Jänicke et al., 2021) as well as local effects from larger construction measures but a comprehensive and generalized explanation is still missing.

### 6.2.3.  21st century projections

Recent SL projections (Fox-Kemper et al., 2021; Palmer et al., 2018; KlimaatScenario's 2023) suggest that 21st century SLR in the North Sea will be close to or slightly higher than GMSLR at southern North Sea coasts, whereas at the more northern
North Sea coasts, projected SLR is lower than the global mean (Section 5.1, Table 3). For example, for the emissions scenario SSP5-8.5, the IPCC AR6 projects for 2100 a SLR of 76-85 cm at the south-eastern UK, Belgian, Dutch and German coasts, whereas at the northern UK and southern Norwegian coastlines, the projected rise is typically below 70 cm (Fox-Kemper et al., 2021; Garner et al., 2021). This gradient is predominantly caused by GIA (Section 3.3) and the gravitational imprint of the melt of the Greenland Ice Sheet (Fox-Kemper et al., 2021; Palmer et al., 2018). In contrast, the projected sterodynamic





SLR is spatially relatively uniform over the North Sea and slightly higher than elsewhere on the Northwestern European continental shelf (Fox-Kemper et al., 2021a; Hermans et al., 2022, their Supplementary Figure 1).

The sterodynamic SLR in the North Sea is typically projected using simulations of GCMs, several of which have a too coarse resolution to capture important bathymetric and topographic features influencing the North Sea circulation such as the Norwegian Trench and the English Channel. Downscaling the simulations of GCMs with a high-resolution regional ocean

model can have large effects on the projected ocean dynamic SL change for the North Sea depending on the GCM (Hermans et al., 2020), but downscaling has not been applied to large ensembles of GCMs yet. Besides changes in annual mean ocean dynamic sea level, CMIP6 GCMs also simulate changes in the amplitude and phase of the seasonal SL cycle (Hermans et al., 2022; Widlansky et al., 2020). The projected changes are largest in the south-eastern and eastern parts of the North Sea and may have implications for intertidal ecosystems.

Several studies have projected changes in the frequency of ESLs in the North Sea due to future SLR using the static approach described in Section 5.3.1. Compared to other regions, the projected frequency amplification factors of the historical centennial event and other return heights are small in most of the North Sea (see Figure 12), because the current variability of extremes is large (Fox-Kemper et al., 2021; Frederikse, Buchanan, et al., 2020; Hermans et al., 2023; Oppenheimer, 2019). These studies did not consider changes in ESLs due to dynamic changes such as changes in storminess or the effect of an increasing water

depth (due to SLR) on surges, tides and waves.

The impact of changes in water depth induced by SLR on surges, tides and waves is more important in the North Sea than elsewhere in Europe since the North Sea is a shallow sea, especially near the southern coasts. Haigh et al. (2020) and Idier et al. (2017) both demonstrated a +10 cm and +10% increase in the semi-diurnal component of the tide along the south-eastern North Sea coast, respectively, for a hypothetical increase of +2 m and + 80 cm. Arns et al. (2017) used fine-scale (1 km)

numerical modelling in the German Bight to highlight that the long-term SLR would generate waves of greater amplitude (around +48-56% depending on the scenario). Chaigneau et al. (2023) showed with regional climate modelling (6 km resolution) that future mean significant wave heights could become up to +8% higher in the southern North Sea than at present, if sea level would rise by 80 cm. These important future changes will also impact the interactions between processes (e.g., tide-surge interaction, (Arns et al., 2020; Bonaduce et al., 2020; Staneva et al., 2021) and lead to further changes in ESLs in

the North Sea.



The potential contribution of changes in atmospheric storminess to changes in ESLs in the North Sea is uncertain and strongly depends on the (large-scale) atmospheric forcing used to project such changes (Howard et al., 2019; Palmer et al., 2018; Vousdoukas et al., 2016). For instance, based on a small ensemble of high-resolution regional model simulations forced with downscaled atmospheric changes from CMIP5 models, Palmer et al. (2018) (UKCP18) and Howard et al. (2019) find that storm surges around the UK may change by –1 to 1 mm yr$^{-1}$ depending on the model, but that the ensemble mean change is close to 0. Under a high emissions scenario, Vousdoukas et al. (2016) project increases in the height of storm surge events with return periods of 5 to 100 years of several percents but report that the disagreement between models is large elsewhere in the region. In conclusion, the amount by which changes in storminess affect ESLs in the North Sea is uncertain, but studies agree that these changes are small compared to the effect of mean SLR itself. In Lobeto et al., (2021a), Chaigneau et al. (2022) and Aarnes et al. (2017), mean and extreme wave characteristics slightly decrease in the north of the North Sea under the SSP5-8.5 scenario. The amplitude of storm surges does not appear to be significantly modified by the mid-century in (Muis et al., 2022) under a very high emission scenario. In contrast, Jevrejeva et al. (2023) showed an increase of +50 cm in extreme storm surges and waves under a low-probability high-impact scenario in the southern North Sea.






**Figure 15: Top: Yearly mean relative sea level from the Cuxhaven tide gauge (starting in 1843) and yearly reconstructed basin average mean absolute sea level (so that GIA effects are not included) from Dangendorf et al. (2019), together with basin average projected multi-model ensemble mean relative sea level and 17th-83rd percentile uncertainties under SSP2-4.5 and SSP5-8.5 obtained from AR6 IPCC. Lower left: Linear trends of vertical land motion inferred from point-wise observations (from GNSS and from altimetry and tide gauges comparisons) reconstructed using Bayesian principal component analysis and interpolated along the coastlines. Lower right: 50-year return levels of extreme still water levels representative of the recent past from GTSM dataset (Yan et al., 2020).**






### 6.3. European Arctic

The European Arctic basin is defined here as the area covering the Nordic Seas; that is the Norwegian Sea, Icelandic Sea, and Greenland Sea. European countries considered in this report and within the European Arctic basin are Iceland and the middle to northern coast of Norway, including Svalbard.

**6.3.1.    Vertical land motion**

An important component of SL change in the European Arctic is VLM. The broad pattern of VLM in the region can generally be ascribed to past ice mass loss and GIA (e.g., Kierulf et al., 2021; Milne et al., 2001; Vestøl et al., 2019). Note that GIA also causes gravity and Earth rotation effects on sea level, which are around 5-10% of the VLM signal. A regional semi-empirical model of VLM and gravity changes (Vestøl et al., 2019) from the Nordic Geodetic Commission (NKG) has been applied in
several sea level studies and can be found at Lantmäteriet (the Swedish mapping cadastral and land registration authority).

There are important contributions to VLM from ongoing ice mass loss on Iceland (Compton et al., 2015) and Svalbard (e.g., (Kierulf et al., 2022) driving high rates of local elastic land uplift and variability. GIA on Iceland, where there is a low viscosity Earth structure which deforms on short timescales, is thought to be dominated by ice mass changes over the past ~100 years (Auriac et al., 2013). VLM on Iceland is further complicated by significant tectonic and volcanic movements. Recent studies
have also shown that ice mass loss in the Arctic and from Greenland produces widespread non-negligible elastic VLM in the European Arctic (e.g., Coulson et al., 2021; Frederikse et al., 2016; Kierulf et al., 2021; Richter et al., 2012). These show that during years of high mass loss from Greenland rates of uplift in Scandinavia reach ~0.7 mm yr$^{-1}$.

InSAR data products for Norway, useful for detecting local VLM changes, are available from Norway's national geological survey (https://insar.ngu.no).

**6.3.2.    Past sea level changes**

Measuring sea level in the European Arctic is challenging due to (1) its remote location and lack of land masses, limiting the number of tide gauges in this region, and (2) hampered measurements from satellites by e.g., sea ice and limited satellite coverage at high latitudes. In a recent analysis of the Arctic Ocean SL record from altimetry, Rose et al. (2019) found a rate of 3.19 mm/year (3.10–3.37 95% confidence interval) between 1991 and 2018 for the sector covering the European Arctic.





A number of studies in the region have looked at coastal sea level variability and trends with particular focus on Norway (e.g., Breili, 2022; Breili et al., 2017; Frederikse et al., 2016; Henry et al., 2012; Mangini et al., 2022; Richter et al., 2012). Interannual sea level variability can be largely explained by atmospheric forcing on wind and the inverse barometer effect. Decadal variability appears to largely reflect steric changes that have been linked to a remote forcing; wind-driven coastally trapped waves which can travel over long distances and reach up to the Arctic (e.g., Calafat et al., 2013; Dangendorf et al.,

2014; Frederikse et al., 2016). Studies have shown that the long-term trends and regional sea level budgets can be explained by mass, steric, and VLM changes (e.g., Frederikse et al., 2016; Richter et al., 2012). Ocean temperature and salinity measurement data used in some of these analyses are accessible from the Institute of Marine Research (https://www.hi.no/en).

The official extreme water levels used in planning along the Norwegian coast are calculated using the average conditional exceedance rate (ACER) method (Skjong et al., 2013). Data on these extreme water levels are available from the Norwegian

Mapping Authority (https://www.kartverket.no/til-sjos/se-havniva). Data on several different storm surge and wave products are available from the Thredds server at the Norwegian Metrological Institute (www.met.no). This includes a detailed wave hindcast that covers parts of the European Arctic (Breivik et al., 2022).

### 6.3.3.   21st century projections

Projections for the European Arctic indicate the region will experience a SL change somewhat below the global average rise

due to (Gravitational, Rotational and Deformational GRD effects, e.g., Simpson et al., 2017; Table 3). These effects also largely explain spatial differences in projected SL change.

Apart from GIA several components of projected SL changes are relevant for the European Arctic. (1) Owing to its relatively close proximity to Arctic glaciers and the Greenland ice sheet, GRD effects cause a negative or less than average SLR in the region. Compared to other basins the European Arctic is particularly sensitive to the pattern of ice melt on Greenland (e.g.,

(Mitrovica et al., 2018) inducing a below average regional SLR. (2) At the same time projections generally indicate that steric dynamic SLR in the Arctic will be larger than the global average. Here the halosteric term is positive and dominates due to ocean freshening (e.g., Pardaens et al., 2011). We note that the large projected steric dynamic SLR in this region also has a large model spread.

As discussed in Section 5.3, there is considerable uncertainty attached to projections of changes to storm surges and waves.

However, these changes tend to be smaller than the projected mean SL change (see e.g., Howard et al., 2019). Projections of future wave climate in the period 2070-2100 generally indicate a lower mean significant wave height in the northeast Atlantic



(e.g., Aarnes et al., 2017). The RCP8.5 scenario yields the strongest reduction in wave height. The exception to this is the northwestern part of the Norwegian Sea and the Barents Sea, where receding ice cover gives longer fetch and higher waves.

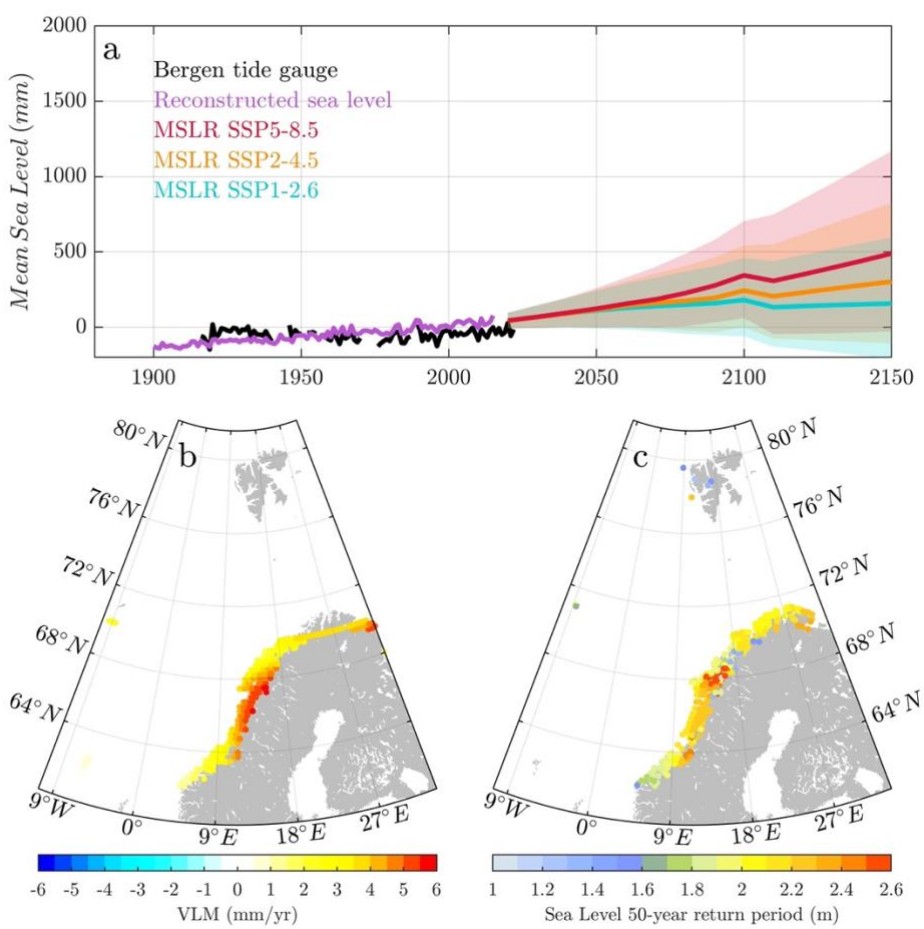


**Figure 16: Top: Yearly mean relative sea level from the Bergen tide gauge (starting in 1915) and yearly reconstructed basin average mean absolute sea level (so that GIA effects are not included) from Dangendorf et al. (2019), together with basin average projected multi-model ensemble mean relative sea level and 17th-83rd percentile uncertainties under SSP2-4.5 and SSP5-8.5 obtained from AR6 IPCC. Lower left: Linear trends of vertical land motion inferred from point-wise observations (from GNSS and from altimetry and tide gauges comparisons) reconstructed using Bayesian principal component analysis and interpolated along the coastlines. Lower right: 50-year return levels of extreme still water levels representative of the recent past computed using the average conditional exceedance rate (ACER) method (Skjong et al., 2013) and accessed online 9th June 2023 (https://www.kartverket.no/til-sjos/se-havniva).**



### 6.4. Mediterranean Sea and Black Sea

#### 6.4.1.    General context

The Mediterranean Sea is a semi-enclosed basin connected to the Atlantic Ocean, to which it exports around 1 Sv (1 Sverdrup = $10^6$ m$^3$ s$^{-1}$) of Mediterranean waters through the narrow Strait of Gibraltar. The mass component is considered the dominant contributor to the mean SL trend in the Mediterranean Sea (Calafat et al., 2010; Pinardi et al., 2014), while the steric component accounts for approximately 20% of the total variance (Calafat et al., 2012). At the sub-basin scale, however, there are large differences, and the steric component can explain a substantial part of the total SL variance, such as in the Aegean, Southern Central Mediterranean and Levantine Basin (Mohamed and Skiliris, 2022). Mean sea level variability at long time scales (interannual to decadal) averaged over the basin has been shown to be consistent with the nearby Atlantic (Calafat et al., 2012). At the regional scale, however, sea level changes within the basin deviate from the mean value, due to ocean circulation, heat redistribution and atmospheric-ocean momentum fluxes. Storm surges are particularly relevant due to the microtidal nature of the basin and are generated both by incoming atmospheric perturbations from the North Atlantic and by regional cyclogenesis, which occasionally generates tropical-like cyclones in the basin (see Section 6.4.7). The Mediterranean Sea is also a hotspot for atmospherically-induced high-frequency sea level oscillations known as meteorological tsunamis (see Section 6.4.8), that affect various locations in the basin.

The Black Sea is an enclosed basin connected to the Mediterranean Sea through the Marmara Sea and the Turkish Straits: The Bosphorus and Dardanelles Straits. The Mediterranean and Black Seas are microtidal basins. The Black Sea receives freshwater from the Danube, Dnieper and Don rivers especially. The salinity of the Black Sea (~18 psu at the surface) is much lower than that of the Mediterranean Sea (~38 psu at the surface). In the Black Sea, most of the steric SLR appears to be related to salinity reduction (implying a SLR), rather than to an increase in temperature (Tsimplis et al., 2004).

#### 6.4.2.    Past sea level changes

In the Mediterranean and Black Seas there is a geographical bias in coastal sea level monitoring with most tide gauge stations located along the northern coasts of the basin and the Black Sea (see Pérez Gómez et al., 2022 for a recent summary of all stations and operators). Although most tide gauges have been deployed since the 1980s, some records date back to the 19$^{th}$ century. This is the case of Marseille and Genova, that indicate a centennial mean sea level trend of 1.3-1.4 mm yr$^{-1}$ since the late 19$^{th}$ century (Figure 17). Linear mean sea level trends from satellite altimetry since 1993, with a GIA correction applied, display positive values among most of the Mediterranean and Black Sea basins (Figure 6) with an average rate of 2.5 mm yr$^{-1}$ over 1993-2022 for the Mediterranean Sea and 1.4 mm yr$^{-1}$ for the Black Sea (European Union-Copernicus Marine Service, 2019b, a). Sea level trends as observed by altimetry over 1993-2020 are lower than the global mean (and the European Seas



mean) in the eastern Mediterranean Sea and Black Sea (Prandi et al., 2021, Figure 6). In addition, a slight deceleration of SLR has also been observed in the eastern Mediterranean Sea and more substantially in the Black Sea (Prandi et al., 2021).

The combination of in-situ and remote measurements allows reconstructing sea level changes over the basin for long periods of time. Mediterranean basin average mean sea level based on the data from Dangendorf et al.  (2019) results in a linear trend of 1.5 mm yr$^{-1}$ since 1900, consistent with long-term tide gauge records. Temporal variability at multidecadal to inter-annual time scales is evidenced by tide gauge records (e.g., Marcos and Tsimplis, 2008). At decadal and multi-decadal time scales, the basin average sea level rates range between -5 mm yr$^{-1}$ and +7 mm yr$^{-1}$ and respond, to a large extent, to variations in the

nearby north-eastern Atlantic Ocean. Part of this variability is coherent along all the European coasts and is driven by longshore winds propagating northwards along the European continental shelves (Calafat et al., 2012; Hughes et al., 2019). At interannual time scales, nearby records are very coherent. At these time scales, mean sea level is largely correlated to large-scale atmospheric circulation patterns, particularly the North Atlantic Oscillation that has been shown to force Mediterranean sea level through different mechanisms (Martínez-Asensio et al., 2014; S. Masina et al., 2022; Volkov and Landerer,

2015). Temporal and spatial variability also results from satellite altimetry data, where non-linearity in sea level trend occurs due to oceanographic processes at the sub-basin scale, being also reflected at the basin scale. Main changes in SL trend occur around 1997, 2006, 2010, and 2016, driven by variations in thermohaline circulation and mass redistribution in the Ionian Sea and other sub-basins (Meli et al., 2023). Subannual sea level fluctuations in the Aegean Sea, in the Mediterranean basin, and the Black Sea are correlated, with the Black Sea lagging behind the Aegean Sea (Volkov and Landerer, 2015). The time lag

between the Aegean/Marmara sea level and the Black sea level increases from approximately 10 days for monthly averages, to nearly 40 days for 9-months averages. The response of the Black Sea level is due to barotropic flow anomalies through the Bosphorus Strait, constrained mainly by friction and the strait geometry (Volkov et al., 2016). Black Sea elevation changes are also forced by sea level pressure, wind stress along the Bosphorus, and the net freshwater flux into the Black Sea.

### 6.4.3.    Vertical land motion

Coastal VLM is a significant contributor to changes in relative sea level in the Mediterranean Sea (Wöppelmann and Marcos, 2012). GIA-related subsidence (Section 3.3) is small in comparison to northern European regions and is estimated to be, on average, 0.5 mm yr$^{-1}$ over the last millennia, although spatially varying (Vacchi et al., 2018). In-situ VLM observations from GNSS and from the combination of altimetry and tide gauges are concentrated over the European coast and around the southern Black Sea, with very few information in northern Africa. Linear trends obtained from these observations are mapped in Figure

17. Data have first been corrected for offsets and outliers (Oelsmann et al., 2022), then used to determine their spatio-temporal variability using a Bayesian Principal Component Analysis and, finally, spatially interpolated along the coastal regions. The results (Oelsmann et al., 2022) display regional variability of VLM in the Mediterranean basin with a median value of -0.4





mm yr$^{-1}$ and highlight areas with differential VLM, as is the case of Venice. However, local variability in VLM is much larger, due to active neo-tectonics and volcano-tectonics affecting large part of the Mediterranean coasts.

### 6.4.4.    21$^{st}$ century projections

Mean sea level projections of the Mediterranean Sea were explored by Sannino et al., (2022) under the RCP8.5 climate scenario, using a high-resolution ocean model capable of resolving the water exchanges through the Strait of Gibraltar. The increase of model resolution together with improved sea level information at the Atlantic lateral boundary and the adequate treatment of the complex, hydraulically driven dynamics across the Gibraltar Strait resulted in an improved description of the

subregional SL patterns. They concluded that the resulting basin-average mean sea level change was within the uncertainties of the multi-model ensemble of global coarser resolution models from CMIP5 (excluding models without an open connection between the basin and the Atlantic Ocean). This study is in line with Adloff et al. (2018) who pointed at the mean sea level in the nearby Atlantic Ocean as a major driver of projected mean sea level changes in the Mediterranean. Therefore, projected regional mean sea level time series averaged over the Mediterranean Sea are nowadays obtained from multi-model ensemble

from CMIP6 (Figure 17, top panel, Table 3). It is worth mentioning that available climate models have a relatively coarse spatial resolution over the oceans, of around 1 degree in latitude and longitude, that misrepresent water exchanges through the Strait of Gibraltar, which are a major component of sea level changes in this semi-enclosed basin. Thus, caution must be taken using the sterodynamic contribution from such models in the Mediterranean Sea. Projected mean sea level values reach, under the SSP5-8.5 climate change scenario, 0.79 m (0.64-1.06 m likely ranges 17-83%) by 2100 and 1.22 m (0.91-1.78 m) by 2150

with respect to the period 1995-2014 (Ali et al., 2022). Under SSP2-4.5, projected mean sea level by 2100 is 0.57 m (0.44-0.79m). Few studies assessed projected sea level changes in the enclosed Black Sea. According to Görmüs and Ayat (2019), relative SLR for the Black Sea would be within ±20% of GMSLR.

### 6.4.5.    Extreme water levels

Coastal ESLs generated by storm surges can be assessed using high-frequency tide gauge records or model hindcasts. Largest

values of storm surges are observed at tide gauges located in the northern Adriatic Sea (Marcos et al., 2009)  and along the Tunisian and Aegean coasts (Cid et al., 2016). Wind-waves, when co-occurring with storm surges, exacerbate the coastal hazard (Lionello et al., 2017). The 50-year return levels of coastal sea level extremes obtained with a 72-year run of a hydrodynamic model coupled with wind-waves (Toomey et al., 2022) are mapped in Figure 17 (bottom panel), showing a consistent picture with observations. Values exceeding 1 m are found in the northern Adriatic and the Gulf of Lions and along

the Tunisian and Libyan coasts. Along the rest of the coasts, 50-year return levels are smaller than 50 cm. Besides changes linked to mean sea level variations, storm surges also display long-term to interannual variability unrelated to mean sea level and associated to changes in storminess. Decadal variations, as those observed in the tide gauge records from Trieste (Raicich,





2003) and Marseille (Marcos et al., 2015b), are geographically consistent and related to large-scale atmospheric patterns (Lionello et al., 2021b; Marcos et al., 2015b). The same applies to changes in storm surges at interannual timescales (Masina

and Lamberti, 2013). Projections of storm surges based on hydrodynamic runs forced with climate models show small and mostly negative changes in southern Europe during the 21$^{st}$ century (Conte and Lionello, 2013; Muis et al., 2020; Vousdoukas et al., 2017). Considering the small changes of marine storminess in climate projections, mean SLR will be the dominant driver of increasing coastal ESLs also in the future, but overall decrease in meteorological surges and storm wave severity is expected in the Adriatic Sea (Benetazzo et al., 2022; Lionello et al., 2021b).

**6.4.6.    Wave climate**

The wave climate of the Mediterranean Sea is characterized by two well defined seasons (winter and summer, with spring and autumn having mixed characteristics). In winter, mean and extreme waves are highest in the western Mediterranean, mostly caused by the dominant north-westerly mistral wind. In summer, waves are lower, with a mean wave maximum in the Levantine basin, caused by the Etesian winds and extreme wave maximum in the western basin (Barbariol et al., 2021; Lionello

and Sanna, 2005). Along the coastal regions, largest waves are found in areas with longer fetch distance, such as the Balearic Islands, west of Sardinia and northern Algerian coasts, with 100-year return levels exceeding 4 m (Toomey et al., 2022). In contrast, values smaller than 1 m are typical of continental coasts protected by small islands, as the Dalmatian coasts and parts of the Aegean Sea (Toomey et al., 2022).

Multidecadal trends from wave gauges have been computed only in the north Adriatic Sea (Pomaro et al., 2017), while in other

locations time series are too short (e.g., Amarouche et al., 2022). Multidecadal trends based on satellite data are still associated with large uncertainties (e.g., Dodet et al., 2020). Therefore, analyses of trends have commonly been based on hindcasts with no overall consensus on trends, possibly associated with the selected period. Trends of the mean wave height are negative or non-significant during the second part of the 20th century (Lionello and Sanna, 2005; Musić and Nicković, 2008; Ratsimandresy et al., 2008), and become positive, particularly in winter, in the western Mediterranean since the 1980s

(Amarouche and Akpinar, 2021; Barbariol et al., 2021). Anyway, projections tend to agree that mean significant wave height will decrease as a consequence of anthropogenic climate change (Lionello et al., 2008; Casas-Prat and Sierra, 2013; De Leo et al., 2020).

**6.4.7.    Medicanes: past and future projections**

Medicanes are mesoscale maritime extratropical cyclones developing over the Mediterranean, whose structure resembles

tropical cyclones. Analysis of their past trends has not been possible until now, but evidence is for a future decrease of their frequency and an increase of intensity, as a consequence of future sea surface temperature increase (González-Alemán et al.,



2019; Koseki et al., 2021; Romero and Emanuel, 2013, 2017). Projected changes in medicane-induced coastal hazards do not exceed 20% of present-day values in terms of storm surges and wind-waves, although there is poor agreement among models' projections (Toomey et al., 2022).

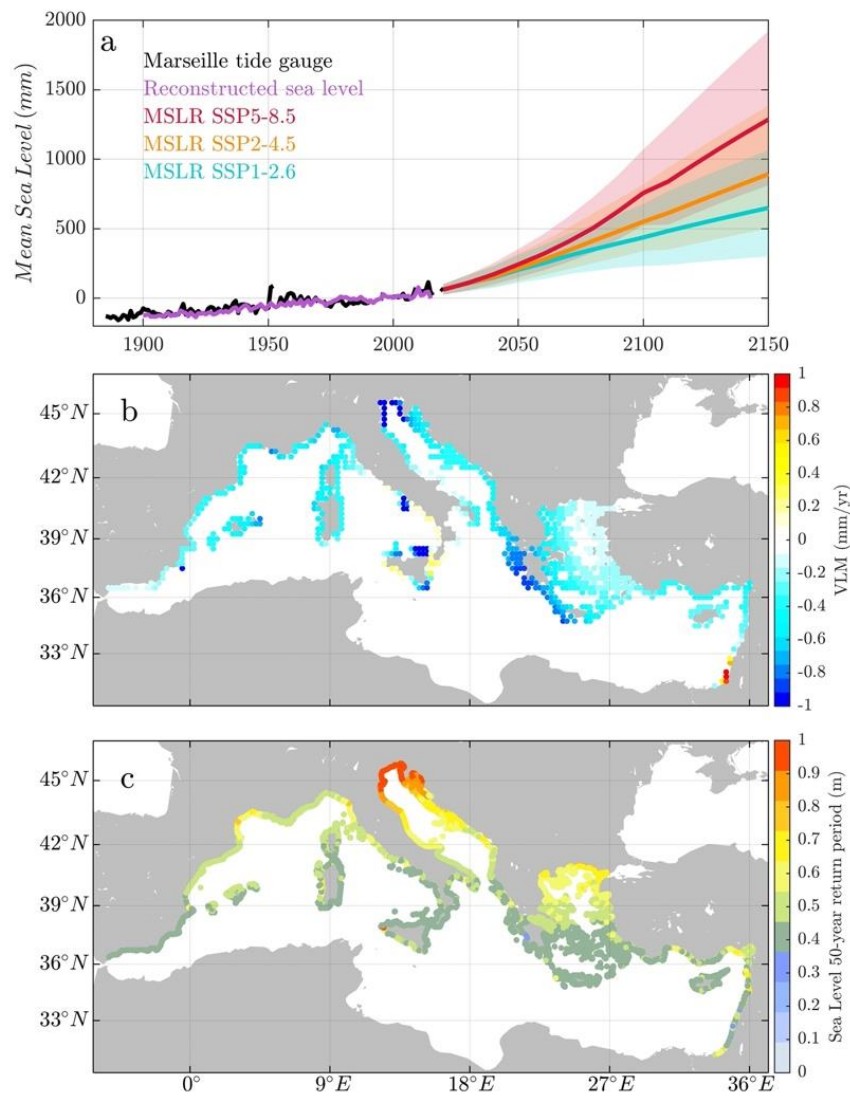


**Figure 17: Top: Yearly mean relative sea level from the Marseille tide gauge (starting in 1885) and yearly reconstructed basin average mean absolute sea level (so that GIA effects are not included) from Dangendorf et al. (2019), together with basin average projected multi-model ensemble mean relative sea level and 17th-83rd percentile uncertainties under SSP2-4.5 and SSP5-8.5 obtained**



**from AR6 IPCC. Middle: Linear trends of vertical land motion inferred from point-wise observations (from GNSS and from altimetry and tide gauges comparisons) reconstructed using Bayesian principal component analysis and interpolated along the coastlines. Note that North African coasts have not been represented due to lack of data. Bottom: 50-year return levels of coastal extreme sea levels computed using a 72-year ocean simulation of coupled hydrodynamic and wave model.**

### 6.4.8.    Meteotsunamis: past present and future

Meteotsunamis are atmospherically induced high-frequency (< 2h) oceanic waves generated by travelling atmospheric perturbations (Monserrat et al., 2006). There are different mechanisms by which an atmospheric disturbance can generate a meteotsunami wave in the open sea, such as Proudman resonance (Proudman, 1929), Greenspan resonance (Greenspan, 1956), frontline passages and even atmospheric Lamb waves (Villalonga et al., 2023). Analogously to seismic generated tsunamis, meteotsunami waves can travel long distances across the ocean, being amplified when they reach the coastline under specific bathymetric and morphological conditions. The Mediterranean Sea is a hotspot for meteotsunami events. These have been observed at various locations within the basin and sometimes have reached heights of several meters along the coast of Croatia (Orlić, 2015), in the Balearic Islands (Rabinovich and Monserrat, 1998; Vilibić et al., 2021), Algeria (Okal, 2021) and the Black Sea (Šepić et al., 2018). In addition, meteotsunamis can also significantly contribute to ESLs generated by other mechanisms (Ruić et al., 2023; Vilibić and Šepić, 2017). For example, recently a meteotsunami has been identified as a contribution to an extreme sea level event in Venice (Ferrarin et al., 2023).

Forecasting meteotsunami events is challenging due to the high-computational load required to simulate all high-resolution processes involved. Some examples have recently been implemented in the Balearic Islands (Mourre et al., 2021; Romero et al., 2019). Alternatively, other proxy-based methods use the relationship between observed high-frequency sea level oscillations and synoptic atmospheric patterns, which is validated using reported meteotsunami events and atmospheric reanalyses (Vilibić et al., 2018; Zemunik et al., 2022). As it is plausible that the effects of climate change will affect atmospheric circulation and synoptic patterns, it will also imply an effect on the frequency and intensity of meteotsunamis (Vilibić et al., 2018). Therefore, these proxy-based methods have also been applied to explore projected changes in meteotsunamis (Denamiel et al., 2023; Vilibić et al., 2018). An analysis of selected events suggests that under extreme global warming scenarios the intensity of meteotsunamis could increase under the higher emission climate scenario (Denamiel et al., 2022).

### 6.5. Baltic Sea

### 6.5.1.    General context

The semi-enclosed and shallow Baltic Sea (mean depth < 54 m, see Seifert and Kayser, 1995) is located in northern Europe in the highly variable transition zone between the maritime North Atlantic region (warm and wet) and the continental Siberian climates (cold and dry). During winter, about 50% of the climate variability is explained by the North Atlantic Oscillation





(Hurrell, 1995; Weisse et al., 2021; see also Chen and Omstedt, 2005). As the Baltic Sea is connected to the adjacent North Sea only through the narrow and shallow Danish straits, sea level oscillations on time scales shorter than 1 month are characterized by oscillations of a quasi-closed system. Pronounced seiches have been observed but all in all, they are energetically insignificant (Neumann, 1941; Wubber and Krauss, 1979). In addition, the amplitude of the diurnal and semi-diurnal tides is small within the Baltic Sea in big contrast to the North Sea (Maagard and Krauss, 1966).

On time scales longer than 1 month, the mean sea level in the Baltic Sea approximately follows the sea level in Kattegat, outside the Baltic Sea, but with larger variance at the northern and eastern most bays (Samuelsson and Stigebrandt, 1996).

It is expected that SLR in the Baltic Sea approximately follows the projected GMSLR (or slightly less) due to the melting of ice sheets and glaciers and the expansion of the warming water (Hieronymus and Kalén, 2020; Meier, et al., 2022; Pellikka et al., 2020; Weisse et al., 2021). However, in particular in the northern sub-basins of the Baltic Sea, GIA (Section 3.3) plays an

important role (Ekman, 1996). Land uplift with a maximum of about 10 mm per year close to the Swedish city Luleå and slight subsidence along the southern Baltic Sea coasts were found (Vestøl et al., 2019) (Figure 7). Due to the seasonality of the wind fields over the Baltic Sea region, sea levels in winter are generally highest, especially in mild winters with high North Atlantic Oscillation index. During periods of strong westerly winds, the Baltic Sea temporarily fills with additional water from the North Sea, also leading to higher storm surges. Storm surges are a threat to low-lying Baltic Sea coastlines (Dieterich et al.,

2019; Meier et al., 2004; Wolski et al., 2014).

### 6.5.2.   Past mean sea level changes and extremes

During the 20[th] century the GMSL and thus also the absolute mean SL in the Baltic Sea rose by about 1-2 mm yr$^{-1}$ (Madsen et al., 2019; Meier, Dieterich, et al., 2022; Oppenheimer et al., 2019; Stramska and Chudziak, 2013; Weisse et al., 2021, section 3.1). In Stockholm, for example, absolute sea level rose by about 20 cm between 1886 and 2009 (Hammarklint, 2009). Over

the last 2-3 decades, sea level rose at rates of 3-4 mm yr$^{-1}$ (Oppenheimer et al., 2019; Weisse et al., 2021; Section 2.2). However, such rates are spatially non-uniform and include impacts of multidecadal variations in wind fields (Passaro et al., 2021). The current acceleration of SLR in the Baltic Sea is small and could only be detected through spatial averaging of observations (Hünicke and Zorita, 2016). However, the amplitude of the seasonal cycle significantly increased during the 20[th] century (Hünicke and Zorita, 2008). The land uplift in the northern Baltic Sea as a result of GIA is still faster than absolute SLR, so

the sea level there is currently falling relative to the land (Groh et al., 2017; Hill et al., 2010; Hünicke et al., 2015; A. Richter et al., 2012; Vestøl et al., 2019; Weisse et al., 2021).

For the 20[th] century, no long-term rising trend was found for ESLs in the Baltic Sea compared to mean changes (Madsen et al., 2019; Meier, Dieterich, et al., 2022; Ribeiro et al., 2014; Wolski et al., 2014). Sea level extremes in the Baltic Sea are





caused by pronounced atmospheric cyclones that sometimes interact with seiches on daily time scales and with volume changes on weekly time scales. As long-term changes in wind fields (frequency, intensity and position of cyclones) on time scales longer than 100 years have not been detected and as changes in other drivers such as tides or non-linear interactions are small, sea level extremes therefore have not significantly changed relative to the mean sea level. This conclusion is supported by a paleoclimate model study for the adjacent North Sea that shows no difference between the impact of warmer and colder climate periods on ESLs (Lang and Mikolajewicz, 2019). Studies that nevertheless report an increase in sea level extremes such as Ribeiro et al. (2014) might be affected by the influence of the pronounced multidecadal variability of the wind fields (Marcos et al., 2015b; Marcos and Woodworth, 2017; Wahl and Chambers, 2016).

### 6.5.3. 21st century projections

As the Baltic Sea is almost completely landlocked and has a complex, highly variable coastline and topography with many individual sub-basins, internal sills and underwater channels, global climate models such as in CMIP6 cannot sufficiently resolve the physics and processes of the Baltic Sea in general and water level oscillations in particular. Therefore, projections for this basin require high-resolution regional climate models, which are driven by global models, for example, by using the statistical and dynamical downscaling approaches (Gröger et al., 2021). An overview about the most recent projections is given by Weisse et al. (2021) and Meier et al. (2022).

Under medium and high emission scenarios, global mean and, thus, Baltic Sea SL will continue to rise during the 21st century (Bamber et al., 2019; Oppenheimer et al., 2019, Table 3). For the Baltic Sea, the contemporary GRD-induced SLR (Gregory et al., 2019) from the melting Antarctic ice sheet will be more pronounced than that from the melting Greenland ice sheet (Grinsted et al., 2015; Hieronymus and Kalén, 2020). Following Pellikka et al. (2018), Pellikka et al. (2020) regionalized nine GMSL projections based on different methods (process-based, semi-empirical) and different emission scenarios (RCP2.6, 4.5, 6.0, 8.5) and found that the sea level in the Baltic Sea will approximately rise by about 87% of the global mean rate.

Future changes in sea level extremes in the Baltic Sea depend on future changes in mean sea level and large-scale atmospheric circulation in combination with changing wind patterns. Model projections do not agree on changes in atmospheric circulation, and therefore their relevance for extreme future sea levels remains unclear (Christensen et al., 2022; Meier, Kniebusch, et al., 2022; Räisänen, 2017). For the Baltic Sea, changes in mean sea level are expected to have a greater impact on future extreme values than changes in atmospheric circulation (Gräwe and Burchard, 2012). Sea level fluctuations are dampened by the sea-ice cover during winter when the ocean surface is shielded from the wind stress. Therefore, it can be concluded with a relatively high degree of confidence that future sea ice loss caused by warming will result in higher sea level extremes in the northern Baltic Sea in those regions that have previously been ice-covered and that will be ice free in future (Meier, Dieterich, et al.,



2022). This would lead to an increase in significant wave height, coastal erosion and resuspension of sediment (Girjatowicz, 2004; Leppäranta, 2013; Orviku et al., 2011). Available projections of ESLs on the European coasts have so far considered all

influencing factors by linear superposition, i.e., absolute mean SLR and land uplift, tides (negligible in the Baltic Sea), storm surges and waves (e.g., Vousdoukas et al., 2016, 2017). The results of some studies, such as Vousdoukas et al. (2016, 2017), suggested that ESLs will rise more than mean sea levels due to small changes in the large-scale atmospheric circulation, such as a northward shift of Northern Hemisphere storm tracks and westerly winds and an increase in the North Atlantic and Arctic Oscillations (e.g., Intergovernmental Panel On Climate Change, 2014). However, these changes in the large-scale atmospheric

circulation over the Baltic Sea region are not consistently depicted in the CMIP5 and CMIP6 global climate models, so that these extreme sea level projections have only little confidence.

For further details, the reader is referred to the Baltic Earth Assessment Reports (e.g., Meier et al., 2023; Christensen et al., 2022; Meier, Dieterich, et al., 2022; Meier, Kniebusch, et al., 2022; Rutgersson et al., 2022; Weisse et al., 2021).






**Figure 18: Top: Yearly mean relative sea level from the Stockholm tide gauge (starting in 1889) and yearly reconstructed basin average mean absolute sea level (so that GIA effects are not included) from Dangendorf et al. (2019), together with basin average projected multi-model ensemble mean relative sea level and 17th-83rd percentile uncertainties under SSP2-4.5 and SSP5-8.5 obtained from AR6 IPCC. Bottom left: Linear trends of vertical land motion inferred from point-wise observations (from GNSS and from altimetry and tide gauges comparisons) reconstructed using Bayesian principal component analysis and interpolated along the coastlines. Bottom right: 50-year return levels of coastal extreme sea levels from the GTSM dataset (Yan et al., 2020).**




**Table 3: Rates of RSLR (in mm yr⁻¹) per European regional seas for 19502-2014 (based on Dangendorf et al., 2019), and 2080-2100 under the SSP1-2.6 low emission, high mitigation scenario, SSP2-4.5 middle of the road scenario, and SSP5-8.5 very high emission, low mitigation scenario. Corresponding time-series are shown in Figure 14 for the north-eastern Atlantic Ocean, Figure 15 for the North Sea, Figure 16 for the European Arctic, Figure 17 for the Mediterranean Sea and Black Sea and Figure 18 for the Baltic Sea. Note that for the European Arctic and the Baltic Sea, rates are also presented over 1950-2014 without GIA contribution (i.e., GIA corrected), as provided by Dangendorf et al. (2019). Reported uncertainties for the 1950-2014 rates correspond to the standard error of the time series only. For 2080-2100 rates, the rate of the median relative rise is reported, together with the trends of the RSLR 17th-83th percentiles, in brackets.**

| mm yr⁻¹ | Reconstruction (1950-2014) | SSP1 2.6 (2080-2100) | SSP2 4.5 (2080-2100) | SSP5 8.5 (2080-2100) |
|---|---|---|---|---|
| Arctic | 1.5±0.1 | 1.6 [-0.7 - 4.5] | 3.4 [1.1-6.8] | 5.9 [2.7 – 10.9] |
| Arctic (no GIA) | 1.4±0.1 | | | |
| Baltic | -1.1±0.4 | 0.6 [-1.5 – 3.2] | 4.5 [3.1-7.1] | 9.2 [5.0-14.7] |
| Baltic (no GIA) | 1.8±0.4 | | | |
| Mediterranean | 1.2±0.1 | 4.3 [1.8 – 7.2] | 6.8 [4.4-10.4] | 12.6 [9.7-17.2] |
| North Atlantic | 1.2±0.1 | 4.4 [1.9 – 7.3] | 7.3 [5.1-10.7] | 12.3 [9.5-17.1] |
| North Sea | 1.5±0.1 | 3.7 [1.6 – 6.3] | 6.7 [5.2-9.5] | 11.8 [8.7-16.5] |



\*\*\*\*\*\*\*\*\*\*\* START OF BOX 2\*\*\*\*\*\*\*\*\*\*\*

**Box 2: A selection of historical storms causing EU coastal flooding and their consequences**

Many severe marine flooding events have affected European coastlines throughout history (Ferrarin et al., 2022; Haigh et al., 2015, 2017; Paprotny et al., 2018). For example, large numbers of people (perhaps as many as 10,000 to 100,000 people per event) may have been killed around the coastline of the North Sea during events in 1099, 1206, 1287, 1421, 1446, 1507 and 1717 (Gönnert et al., 2001). The 'Big Flood' of 31 January–1 February 1953 killed 1,836 in the Netherlands, 28 in Belgium, 307 in England and 19 in Scotland, and damage costs were over € 2 billion in today's prices (Gerritsen, 2005; McRobie et al., 2005). This event, together with the 16–17 February 1962 flood in Germany, were the driving force for major improvements in sea defenses (e.g., The Delta-program in the Netherlands) and led to the establishment of storm surge forecasting and warning services (Gerritsen, 2005; Gilbert and Horner, 1986). On January 3, 2018, storm Eleanor crossed the North Sea and caused large storm surges along the coasts of the Netherlands. Based on the water level forecasts, five barriers of the Delta Works were closed. In particular, the automated closure of the Maeslantkering, one of the largest mobile storm surge barriers worldwide, was tested during Eleanor by adjusting the water level critical threshold, leading to the 2$^{nd}$ closure of the storm surge barrier since its completion in 1997. On the other side of the North Sea, the Thames Barrier was also raised to protect London from flooding.

During the winter of 2013/14, the UK, France and Spain experienced an unusual sequence of storms and some of the most significant coastal floods in the last 60 years (Garrote et al., 2018; Spencer et al., 2015; Toimil et al., 2017).

Venice and the Northern Adriatic Sea have long suffered the impact of rising sea levels experiencing several coastal floods, with the most intense events occurring in 1966, 1979, 2018 and 2019 (Lionello et al., 2021b). It is worth nothing that four of the eight largest flooding events in Venice since 1872 happened in 2018 and 2019 (Lionello et al., 2021b), suggesting a possible change in frequency. Below, a focus is given on the Venice case, and on two storms: Xynthia and Gloria.

**Venice: Nov 1966, Nov 2019 (Mediterranean Sea):**

Since the mid 20th century, the frequency of floods of the historical centre of Venice has been progressively increasing (Lionello et al., 2021a). Two extreme water levels, namely the floods of 4 November in 1966 (De Zolt et al., 2006), and 12 November 2019 (Ferrarin et al., 2021) have dramatically posed the issue of the security of the monumental heritage and of the economic activities. The November 2019 extreme water level was analysed in detail by Giesen et al. (2021) and the CMEMS could forecast the anomaly three days in advance. This has motivated the construction of the MoSE defence system, which was first operated to prevent the flooding of the city in 2020 (Lionello et al., 2021a). MoSE closes temporarily the inlets of the





Venice Lagoon preventing the sea level extremes to reach the city centre. MoSE relies on an accurate sea level forecast (see Umgiesser et al., 2021 for a review), which failed in the case of 12 November 2019 (Ferrarin et al., 2021) and is based on the concept that the frequency and duration of closures are limited. This principle might become unrealistic in the second part of the 21st century, where long closures will have negative impact on the lagoon ecosystems and the ship traffic.

The highest floods are produced by the south-easterly wind blowing above the shallow northern Adriatic Sea and associated with the passage of a mid-latitude cyclones above northern Italy (Lionello et al., 2021b). On 12 November 2019 an unprecedented substantial contribution of a small mesoscale cyclone was among the multiple causes of the extreme event (Ferrarin et al., 2021). The increased frequency of floods is produced by the increase of the relative mean sea level (Lionello et al., 2021a), at a rate of 2.5 mm/year in the past 150 year, resulting from approximately equal contributions of vertical land movements and mean SLR (Zanchettin et al., 2021).

The likely range of North Adriatic relative level projections at the end of the 21st century goes from 32 cm (lower limit of the RCP2.6 low emission scenario) to 110 cm (upper limit of the RCP8.5 high emission scenario), and it might reach 1.8 m in a high-end scenario (Zanchettin et al., 2021). However, divergence among scenarios occurs after 2050, time at which all values are in the range 20-40 cm (Zanchettin et al., 2021). It is estimated that preventing the flood of the city centre would require the closure of the inlets for 2-3 weeks, 2 and 6 months per year in correspondence of RSLR of 30, 50 and 75 cm, respectively (see
Lionello, Barriopedro, et al., 2021 and references therein).

**Storm Xynthia (north-eastern Atlantic):**

The Xynthia storm hit the Atlantic coast of France, especially Vendée and Charente-Maritime, during the night of Feb 27-28 in 2010 (Figure 19). Xynthia caused 41-flood related deaths (Vinet et al., 2012), 79 injured and 500,000 affected people. Dikes were overtopped and damages were estimated to a total of € 2.5 billion with 4,800 houses flooded, 120 km of coast eroded,
failure and damages to flood defences occurred along a coastline of 200 km, and 50,000 ha of land areas flooded (e.g., Kolen et al., 2013).

Although the storm characteristics (atmospheric pressure, winds) were less exceptional than previous storms such as storm Martin in December 1999 or storm Klaus in 2009, it resulted in exceptional coastal floods as the peak of the storm surge (reaching 1.53 m at La Rochelle) was reached during spring high tides (+3.0 m with a coefficient of 102 at La Rochelle), and
with high waves (7.5 m of maximum significant wave height). Tide gauges recorded water levels reaching +4.51 m NGF (official levelling in France) at La Rochelle (8.01 m wrt hydrological zero). Such water levels are well above the centennial





level for Vendée and Charente-Maritime, estimated at +4.0 m NGF (Simon, 2008) and are estimated to correspond to a 200-250-yr return period.

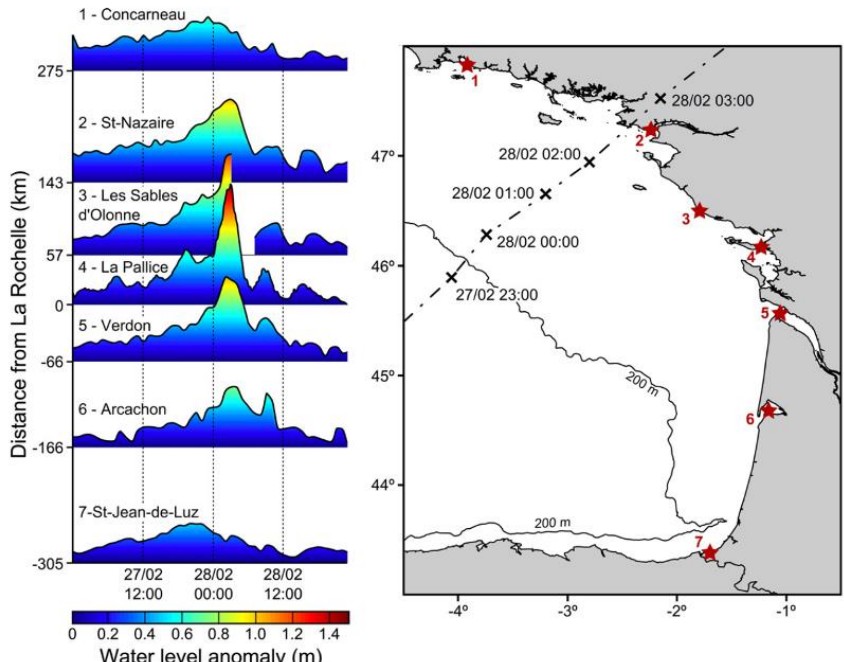

**Figure 19. Storm surge during Xynthia along the French coast of the Bay of Biscay, showing a maximum at La Pallice station (station 4). Extracted from Bertin et al., (2012).**

Xynthia was a tipping point for adaptation to coastal flood and associated risk management for France, due to its high impact. Following Xynthia, different measures were implemented. A national coastal flood early warning system was developed by national agencies (SHOM and Météo-France) and the prevention fund for major natural hazards, known as the Barnier fund, was extended to marine flooding. As such, since April 2010, owners of houses that were severely damaged or are threatened due to their location in high-risk of coastal flooding areas have been allowed to sell their property to the State. 1176 properties were sold to the State for a total of 330 million euros. Dikes were repaired for an amount of € 300 million and more than 300 local priority coastal risk prevention plans were defined.

**Storm Gloria (Mediterranean Sea):**





Storm Gloria was formed by a low-pressure system of Atlantic origin that intensified over the Western Mediterranean, starting on 19 January 2020 and lasting until 26 January. It affected the eastern coasts of Spain and the Balearic Islands, with intense and sustained winds that led to record-breaking wind-waves (Figure 20) and heavy precipitation (Amores et al., 2020; de Alfonso et al., 2021; Pérez-Gómez et al., 2021; Toomey, 2022). It caused severe damages along the coasts of the Spanish

mainland and the east of the Balearic Islands, including a total of 13 fatalities, flooding and strong erosion with economic losses of several millions of euros and damages to the power supply networks.

In-situ wave observations from deep water buoys provided measurements of significant wave height over 8 m, exceeding all historical records and corresponding to return periods of several centuries when only previous measurements are accounted for (de Alfonso et al., 2021). Likewise, in-situ sea level observations from tide gauges along the eastern Spanish coasts

measured storm surges over 50 cm (Amores et al., 2020; Pérez-Gómez et al., 2021). In particular, in the southern Gulf of Valencia, a hydrodynamic-wave coupled model simulation quantified the effect of wave setup as large as 40% of the total storm surge observed, which was close to 70 cm (Figure 20), due to sustained strong winds (Amores et al., 2020).





Figure 20: Storm Gloria (a) total sea surface elevation (SSE) along a coastal stripe affected by the storm in the western Mediterranean Sea, and contributions to SSE: (b) wave setup, (c) atmospheric pressure and (d) wind setup contributions to storm surge. In panels (b-c-d), the absolute (relative) contributions are indicated by the profile on the left (right). Units for contributions



to SSE in absolute terms are in cm. Units for contributions to SSE in relative terms are in percentages. Note that the colour scales for the wind contribution have higher limits. From Amores et al., (2020).


*********** END OF BOX 2 ***********

**7 Conclusions**

This paper provides an assessment of regional to local historic and future sea level changes in Europe, both for the long-term trends and for extremes. It complements existing global and European assessments by providing additional geographical and
contextual details, as scoped with stakeholders during dedicated regional workshops and the Sea Level Rise Conference 2022 (see Sea Level Rise in Europe: knowledge gaps identified through a participatory approach). European regional seas present contrasted environments, from the microtidal and enclosed Mediterranean and Black Seas, to the open ocean in the northeastern Atlantic with large tides and exposition to extra-tropical storms, to the uplifting northern Baltic Sea and European Arctic. The main drivers of RSLR and of ESLs thus vary along European coastlines. Key processes and drivers or specificities of each
European regional seas with regard to sea level changes are reviewed.

In terms of SLR, absolute sea level trends since 1993 have been on average slightly above the global mean rate, with only a few areas showing no change or a slight decrease. VLM, notably due to GIA and human activities, can lead to substantial regional to local deviations between absolute and relative sea level changes, especially over the uplifting northern Baltic and hotspots of coastal subsidence.

Projected mean RSLR is the largest in the northeastern Atlantic, North Sea and Mediterranean and Black Sea, and the lowest in the European Arctic and Baltic Sea. The Baltic Sea exhibits strong spatial gradients of projected RSLR, with SLR close to the global mean in the southern basin, and relative sea level fall in the northern Baltic due to GIA.

ESLs will occur more frequently along most European coasts during the 21st century. Amplification factors of the frequency at which ESLs will occur during the 21st century broadly show a meridional gradient, mostly related to the spatial amplitudes
of tides and of storm-induced sea level variability. The largest amplification factors are projected for southern Europe, especially so in the microtidal Mediterranean Sea. The lowest (but positive) amplification factors are projected for northern Europe, in macro-tidal regions exposed to storms and induced large surges such as the south-eastern North Sea. ESLs are projected to occur less frequently in the northern Baltic Sea, due to a relative mean sea level fall.

Several knowledge gaps are identified. An important one concerns ESLs, including the contribution from wind-waves, dynamic changes in tides, surges and wave setup and runup, non-linear interactions between these drivers of ESLs, and marine and fluvial/pluvial extreme compound events. Regionally downscaled projections or more local information of relative mean and ESL changes are needed with characterized uncertainties. A major uncertainty for SLR remains attached to ice sheets instabilities and overall contributions. Finally, more robust projections beyond 2100 are needed. Finally, the interpretation of
regional SLR variations for local perceptions and decision making is also an area needing improvement.

**Authors contributions**

AM and RvdW coordinated the paper. AM led the writing of the abstract, conclusions and Section 1; MDP led Section 2; MM, MS, BM and AM led Section 3; RR, RvdW, BM, MM and AM led Section 4; AS, RR, RvdW, AM, TH, AC led Section 5; AM, AA, TH, MS, MM, MaM led Section 6. AS led Box 1, AM and MM led Box 2. AnA, MM, MP, AS, MS, TH created the
figures. All authors participated in the iterations and revision of the paper.

**Acknowledgments**

This paper is part of the Assessment Report of the Knowledge Hub on Sea Level Rise, a joint effort between JPI Climate and JPI Oceans operating as a networking platform facilitating the interaction between research and policy on sea level rise. We sincerely thank all the participants in regional scoping workshops and in the Sea Level Rise Conference 2022 held in Venice
in October 2022, for contributing their knowledge, experience and information scoping this paper.

RvdW, TH and AS were supported by PROTECT, which has received funding from the European Union's Horizon 2020 research and innovation programme under grant agreement No 869304. This is PROTECT contribution xx. AM and BM were supported by CoCliCo, which has received funding from the European Union's Horizon 2020 research and innovation programme under grant agreement No 101003598. RR has been funded by the TiPACCs project, which receives funding from
the European Union's Horizon 2020 research and innovation programme under grant agreement no. 820575.

**Competing interests**

The authors declare that they have no conflict of interest.

**Data availability**



Data used in this paper are available from IPCC AR6 (projection data available from https://doi.org/10.5281/zenodo.5914709),
from GESLA, PSMSL, Copernicus Marine Service, and the Copernicus Climate Data Store. Data from Dangendorf et al.
(2019) are available from the corresponding author on request. Data from Jevrejeva et al. (2023) are available from their
Supplementary Files.  Statistics on present-day uplift, geoid, gravity and Stokes coefficient rates derived from Caron et al.
(2018) are available at https://vesl.jpl.nasa.gov/solid-earth/gia.




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
