# Peer review of "Sea level rise in Europe: observations and projections"

_State of the Planet, 2023_

## Community Comment (CC2)

**Comments on sp-2023-36 by Sergiu Dov Rosen***

* Retired Senior Scientist, Haifa, Israel. Previously MedGLOSS Coordinator, member of GLOSS Group of Experts, chair/co-chair of Working Group 3 of ICG/NEAMTWS on Sea Level Data Collection and Exchange, incl. Offshore Tsunami Detection and Instruments.

Estimeed authors,

First I wish to congratulate the authors for the excellent paper produced, covering the full range of processes leading to SLR and its future assessment for European seas.

My comments are mainly related to the processes contributing to SLR and ESL in the Levantine basin of the Mediterranean Sea, particularly in the Nile littoral cell, covering the South Eastern Mediterranean coasts from Egypt Nile delta to Israel. They are listed below and then a short list of references.

1. The paper mentioned that one of the factors affecting the SL is the ocean circulation and that regional models of higher resolution are needed to improve the assessments (page 2, lines 46-47). However, in the South Eastern Mediterranean, an important contribution seems that it has been overlooked as it was not mentioned in the paper. I refer to the discharge of Red Sea water via the Suez Canal in the Mediterranean Sea, which, since the beginning of the 1990's increased significantly, due to the Canal deepening and widening a few times, as already indicated by Rosen (2014). The Red Sea waters flowing to the Mediterranean, pass on their way through the Bitter Lakes region, where they collect salt and heat, entering the Mediterranean much saltier and warm than both the Red Sea and the Mediterranean Sea, sinking toward the bottom of the Mediterranean. My rough estimate, based on published cross sections and flow speeds ( Eid F.M., et al., 1997;  Alam El-Din K.A., et al., 1999; Elzeir M.A., and T. Hibino, 1999;  Suez Canal Authority) is of a discharge of about 100 to 120 Km$^3$/year. Thus, a significant amount of salt has been added to the Mediterranean water volume. Published studies indeed confirm the presence of increased salinity waters off the SE Levantine basin. Furthermore, the warmer waters add to the heating of the lower waters of the region thus contributing to the thermosteric contribution of the Mediterranean. Unfortunately, so far model studies of the SLR in the Mediterranean did not include this contribution which lacks their assessments. This however has been observed from the analyses at the GLOSS station 80 Hadera which indicated a faster SLR of 5.9 mm/year for the period 04/1992-003/2014 (and about 5 mm/year for the period 04/1992-03/2023) relative the global average SLR rate for the same period of 3.5 mm/year. Furthermore, another flow change apparently has not been incorporated in the models. I refer to the Nile river discharge to the Mediterranean, which until 1965, when the High Aswan dam was finished, was of millions of m$^3$/year, reduced to about 5,000 m$^3$/year ever since. However, some part of the Nile water of unknown volume, used for agricultural use in the Nile delta coast, has been seeping to the Mediterranean, as indicated by satellite monitoring.

2. In the introduction of the paper (page 3, lines 61-62) flood coastal plains are mentioned. In my opinion a clarification would be appropriate, indicating that in the paper the flood plain is defined as that due only to the penetration of sea level .However, the true flood plain is of higher elevation, induced by reduced river flow discharge into the sea, caused by the reduced flow gradient due to elevated sea water levels at the river mouths during river floods.

3. In section 2.1 (page 6, lines 124-125) the article refers as stated, mainly to the two IPCC latest reports, which were published a few years ago. However, it seems to have disregarded important new information published in November 2023, indicating potential much higher and faster sea level rise from melting ice caps (ICCI, 2023).

4.  a.  Both in the IPCC AR6 report and in your article (page 6, line 150), future SLR are referred relative to the average SL during the decade 1995-2014. The reasoning to refer sea level rise to a decade (1995-2014) has not been clearly motivated.  Indeed one possible explanation I found is due to similar global average temperatures as during the Last Interglacial stage, about 125,000 years ago (global average sea surface temperature was about 0.5° ± 0.3°C (0.9° ± 0.5°F) above the preindustrial level [that is, comparable to the average over 1995–2014, when global mean temperature was about 0.8°C (1.4°F) above the preindustrial levels] (USGCRP, 2017). Personally I am not convinced that such reasoning is adequate for selecting the average of that period for reference since the accuracy of the temperatures during the historic times is somewhat questionable. I would rather prefer reference to a certain year or series of calm years without strong NO and El Ninio/La Ninia activities.

    b.  Furthermore, the use of Gregorian calendar years is wrong in reviewer's opinnion, as division by calendar years (January to December) from climatic cycle point of view, has the potential to place time neighboring dependent ESL events occurring in the December-January period in different years in the northern hemisphere, potentially affecting statistics of ESL. Hence a different yearly division would be much more correct, such as from April to following March end of next year or from October to following next  year September end. Personally I use the April to March yearly meteo-marine division.

5.  On page 12, in Figure 2, on the Israel Mediterranean coast are shown 3 stations two with blue dots representing recording periods of up to 25 years and in the middle, one green dot, hardly visible as covered by the blue dots, of record length over 25 years and less than 50 years. In fact, the green dot should be for the Hadera GLOSS station 80, with a high frequency record of over 30 years. Another green station should have been for Tel Aviv Yaffo, also with high frequency SL over 25 years.

6.  On page 29, Figure 10 shows Median relative sea level regional projections (*medium confidence*) from the IPCC AR6 report around Europe. I would like to suggest that the abscissa title be more explicit, stating the following text: Mean sea level by 2100 w.r.t 1995-2014 (m), based on *medium confidence* projections, i.e. excluding ice sheet processes associated with deep uncertainty.

7.  On page 57, section 6.4.6 on wave climate, the statement on line 1222 in my opinion may be misleading, claiming "In winter, mean and extreme waves are highest in the western Mediterranean,". The statement is true if one refers to European coasts only, but since in the paper it is stated that the report covers European seas, and since in many places the whole Mediterranean is shown, it would be appropriate to include the Levantine basin too, and then the correct statement would be that "in winter, mean and extreme waves are highest in the eastern Mediterranean followed by those in the western Mediterranean." Similarly, the statements regarding fetches and correspondingly the return periods for the 100 year return level would increase to over 8 m deep water significant wave height offshore the Israeli coast.

REFERENCES:

Rosen, S.D., 2014. From Tsunami Early Detection and Warning to Multi Hazards Early Detection and Warning, CIESM International Conference on East - West Cooperation in Marine Science, Sochi, Russia, 1-3 December 2014, Abstracts, http://www.ciesm.org/marine/sochi/abstracts/index.php

Eid F.M., et al., 1997. Sea-level Variation Along the Suez Canal, Estuarine, Coastal and Shelf Science (1997) 44, 613–619.

Alam El-Din K.A., et al., 1999. The effect of the Suez Canal development on the tide and tidal current- model study, Conference Paper, November 1999. https://www.researchgate.net/publication/332396520

Elzeir M.A., and T. Hibino, 1999. Hydrodynamic Simulation Of The Suez Canal; A Water Body Connecting Two Open Seas. Annual Journal of Hydraulic Engineering. JSCE. VOL.43, 1999. February.

Suez Canal Authority, Canal Characteristics, https://www.suezcanal.gov.eg/English/About/SuezCanal/Pages/CanalCharacteristics.aspx

ICCI, 2023. State of the Cryosphere 2023 – Two Degrees is Too High. International Cryosphere Climate Initiative (ICCI), Stockholm, Sweden ; https://www.iccinet.org/statecryo23.

USGCRP, 2017. Climate Science Special Report: Fourth National Climate Assessment, Chapter 12, Volume I [Wuebbles, D.J., D.W. Fahey, K.A. Hibbard, D.J. Dokken, B.C. Stewart, and T.K. Maycock (eds.)]. U.S. Global Change Research Program, Washington, DC, USA, 470 pp., https://repository.library.noaa.gov/view/noaa/19486

---

## Author Comment (AC1)

**REVIEWER #1**

This review on sea level rise in Europe will be a very good reference for sea level scientists and users of sea level information. It is generally clearly written and covers all the important information. It is very long, to be completely honest I didn't get the stamina to read it all. I did not read sections 6.3, 6.4 and Box 2 so I hope that the other reviewer will cover these regions which I am less familiar with. I have a suggestion to make the review a little shorter (see detailed comments below) but almost all the material covered is relevant. I have no major comments on the review but I have many small comments to help improve it.

We warmly thank the reviewer for the time spent reviewing the manuscript, for her/his careful reading and constructive comments. We reply to each comment below in blue.

**Comments on figures 14-18:**

Please note that Figures 14-18 are now corresponding to Figures 15-19 in the revised manuscript.

- A "reconstructed" sea level time series is provided but no explanation for what it means is provided. Only a reference. I think this should be explained otherwise readers need to go to Dangendorf et al. 2019 to understand.

Thank you for the suggestion. As the second reviewer also requests more information on the reconstruction, we follow these recommendations and introduced a description of the global reconstruction of global mean sea level changes at the beginning of section 6.

- Which reference period is used to compare the time series? For the projections it is probably 1995-2014 which is the reference period of AR6 but it does not seem the be the case for the reconstructed sea level.

The vertical reference of the reconstructed sea level has been adjusted to match projected mean sea level records, as it is arbitrary (in the same way as tide gauges vertical datum). This is now specified in the introduction of section 6. Captions of Figures 15-19 (formerly Fig 14-18) have been updated.

- I don't think it is a good choice to add one tide gauge per basin in the same plot as the basin means. One can't help but compare the past and the projections and see some clear mismatches for some tide gauges. I suggest that if individual tide gauges are shown, they are compared with projection for this particular location, not basin average.

Fig 15-19 were redone without tide gauge records.

- It would be useful to have mean sea level changes for the coast of this region in Figure 14-18 in the same format as panel b and c.

We thank the reviewer for this suggestion. However, mean sea level changes for Europe are shown in Fig 6 and we chose to not replicate the information with an extra panel in Fig 15-19.

**Abstract:**

"Absolute sea-level change" -> I think geocentric sea-level change is now the standard terminology

Indeed, following Gregory et al. 2019. "Absolute sea level" has been replaced by "geocentric sea level" throughout the paper. To ease understanding of readers, it has also been defined in the introduction (first occurrence of "geocentric sea level" in the main text):

*"In many coastal megacities, including European ones, VLM can induce relative SL trends similar to or larger than trends induced by oceanic and climate factors causing geocentric SL changes (Gregory et al. 2019, aka absolute sea level changes) (e.g., Nicholls et al., 2021; Wu et al., 2022)."*

**Introduction:**

l.61-62: References are needed to support those numbers.

Indeed, we added a reference to Neumann et al. 2015. Numbers are given in their Table 5.

Neumann, B., Vafeidis, A. T., Zimmermann, J., and Nicholls, R. J.: Future Coastal Population Growth and Exposure to Sea-Level Rise and Coastal Flooding - A Global Assessment, PLOS ONE, 10, e0118571, https://doi.org/10.1371/journal.pone.0118571, 2015.

l.90: "contemporary solid Earth deformation due to land ice mass loss" would fit better in the anthropogenic category than the natural one.

As stated in IPCC AR6 (Fox-Kemper et al., 2021) "the SROCC reported limited evidence and medium agreement for anthropogenic forcing of the observed AIS mass balance changes. As stated in Section 3.4.3.2, there remains low confidence in attributing the causes of the observed mass of loss from the AIS since 1993, in spite of some additional process-based evidence to support attribution to anthropogenic forcing.".

Regarding the Greenland icesheet, IPCC AR6 states that: "The AR5 assessed that it is likely that anthropogenic forcing has contributed to the surface melting of Greenland since 1993 (Bindoff et al., 2013). Section 3.4.3.2 assesses that it is very likely that human influence has contributed to the observed surface melting of the Greenland Ice Sheet over the past two decades. There is medium confidence of an anthropogenic contribution to recent mass loss from Greenland.".
To avoid categorizing VLM due to the contemporary mass loss of ice sheets and mountain glaciers into either a natural or an anthropogenic origin, the sentence *"At more coastal scales, relative SL changes can be due to VLM of natural (e.g., sediment compaction in deltas, Earth tectonics, GIA) and anthropogenic origins (e.g., pumping of groundwater, weight of the built environment, solid Earth deformation due to contemporary land ice mass loss)." has been rephrased to:*

*"At more coastal scales, relative SL changes can be due to VLM of natural and anthropogenic origins (e.g., sediment compaction in deltas, Earth tectonics, GIA and solid*

*Earth deformation due to contemporary land ice mass loss, pumping of groundwater, weight of the built environment).”*

l.98 and l.106: What are those references? Sea Level Rise in Europe: knowledge gaps identified through a participatory approach, Sea Level Rise in Europe: A Knowledge Hub at the ocean-climate nexus.

These two references are two companion papers from the Knowledge Hub on Sea Level Rise Assessment Report (Special Issue of State of the Planet: https://sp.copernicus.org/articles/special_issue1286.html )

Jiménez, J. A., Bonaduce, A., Depuydt, M., Galluccio, G., Van Den Hirk, B., Meier, H. E. M., Pinardi, N., Pomarico, L. G., Vazquez Riveiros, N., and Winter, G.: Sea Level Rise in Europe: Knowledge gaps identified through a participatory approach, https://doi.org/10.5194/sp-2023-34, 2023.

The second paper: “Sea Level Rise in Europe: A Knowledge Hub at the ocean-climate nexus” is expected to be submitted. As this is not yet the case, the reference to this paper has been removed from our manuscript.

l.99: I don't think the concept of European Union (EU) is relevant to this review since the focus on Europe as a geographic continent rather political union.

Indeed. The text has been reformulated to:
*“European regional seas (see Jimenez et al., 2023) and their bordering coasts along Europe are presenting contrasting environment (…)”*

**3 Regional observations**

Figure 5: Was the data adjusted for GIA and TOPEX-A instrumental drift? This information is given in the caption of figure 6. I think it would be useful here as well.

The caption of Figure 5 has been updated notably to include this information.

Figure 6: Consider using a map projection that deforms less the area

The projection has been changed to match other maps in the manuscript.

l.318: “In Europe, sea level trends are slightly above the global mean rate, on average (Figure 6).”
I think this claim needs more foundations. Especially since it is also in the abstract. Which area is chosen here to represent Europe? What is the rate?
Given the large spacial difference between the Baltic Sea (~5mm/yr) and the Mediterranean Sea (~2mm/yr) I wonder if it is very useful to make conclusions about the European average.

That is right. The sentence is changed to *“In Europe, since 1993, geocentric sea level trends have been contrasted with high sea level rise in the Baltic Sea (see section 6.5 and Fig 7 for relative sea level rise in the Baltic), low sea level rise in the Mediterranean Sea and a sea level rise close to the global mean rate, in the Atlantic sector (Figure 6).”*

l.341: What is meant by "general circulation" here? Is it the ocean circulation in the Mediterranean Sea?

*Yes, we reformulated this sentence:*

*"Strong differences in SL trends at the sub-basin scale are also recognized in the Mediterranean (…), in which variability and complexity arise from changes in ocean circulations."*

l.356: "The annual cycle amplitude ranges from 40 cm to 100 cm"
I think this is a typo, it should be mm instead of cm.

Additionally, Dangendorf et al. 2013 found mean amplitude of the seasonal cycle of 14 to 20 cm along the German Bight which indicates that the range provided here is too narrow.

Dangendorf, Sönke, Thomas Wahl, Christoph Mudersbach, and Jürgen Jensen. "The Seasonal Mean Sea Level Cycle in the Southeastern North Sea." Journal of Coastal Research, no. 65 (10065) (January 1, 2013): 1915–20. https://doi.org/10.2112/SI65-324.1.
*Thanks, this has been corrected:*

*"The annual cycle amplitude ranges from around 5 cm to 12 cm with the largest amplitude found in the North Sea, Baltic Sea, along the Arctic coast of Norway and in the western Mediterranean Sea (Fernández-Montblanc et al., 2020; Ray et al., 2021), going up to 20 cm in the German Bight (Dangendorf et al. 2013)."*

**4 Drivers of sea level rise and extremes**

l.426-437: This paragraph is important but it is too vague at the moment. It would benefit from some numbers. How much does a melting to Greenland resulting in 1cm GMSLR influences the European coast? Same for Antarctica. For Greenland the influence is so different between North-West and South-East Europe that general sentences like "If the contribution from Greenland increases in the future, this has not very large consequences for European coasts, …" can be misleading.

*We have further specified the difference between Southern and Northern Europe with respect to the Greenland contribution.*

Additionally, only gravitational effects are mentioned here, why not discuss also solid earth deformation, which is very important for the influence of Greenland mass loss, and rotational effects?

*We have added the sentence "This contemporary GRD effect is sometimes referred to as a sea level fingerprint" to make it clear than this is a GRD effect. For most of Europe it is the long wavelength signal of the gravity change that is important – which is why that is the focus of this section.*

*For some areas of Europe, notably the higher latitudes, the elastic Earth response to Greenland ice loss is important. This is discussed in the Section which covers the European Arctic (section 6.3), where we note the elastic uplift signal can be significant.*

The next paragraphs of 4.1 are rather detailed but do not provide regional information and do not have a high added value compared to IPCC AR6 except that they are updated. So I would consider removing them.

The first of the two paragraphs explains which processes are important for the possibility of enhanced mass loss from Greenland. We don't reckon it is discussed as such in the AR6 report and decided to keep it because it provides a physical reasoning as to why the Greenland contribution can increase by a factor of two in the future.

The second paragraph explains that ice shelves play a critical role for the contribution of Antarctica to sea level rise. This is not described explicitly in AR6 and deserves highlighting in the context of future SLR in Europe, which is an essential part of the paper.

l.481: "will be dominate" -> will dominate

Thanks, corrected to: *"For some shelves, atmospheric process will dominate while for other shelves, oceanographic controlled processes will dominate."*

**5 Projections of sea level rise and extremes**

l.498: "the amount of outgoing radiation amount"

Thanks, typo corrected.

Figure 11 is for the whole world but in the caption it is written Europe. I think it would be better to zoom in on Europe.

Indeed. Fig 11 was redone with a zoom on Europe only.

l.681: "with consequences for enhanced SLR in the early 21st century" should be 22nd century

Thanks, corrected.

**6 Key developments per region**

**6.1 Atlantic Ocean**

L.810-816: This whole paragraph is missing references. In which paper was the slope current described? The upwelling?

Thank you for noting it. References were added:
*"A slope current flows northward along the continental slope separating the deep ocean from the continental shelf (Huthnance and Gould, 1989; Clark et al. 2022). Strong summer upwellings of deeper, colder water occur along the coasts of Portugal (Fiúza, 1983)."*

Huthnance, J. M. and Gould, W. J.: On the Northeast Atlantic Slope Current, in: Poleward Flows Along Eastern Ocean Boundaries, edited by: Neshyba, S. J., Mooers, Ch. N. K., Smith,

R. L., and Barber, R. T., Springer New York, New York, NY, 76–81, https://doi.org/10.1007/978-1-4613-8963-7_7, 1989.

Clark, M., Marsh, R., and Harle, J.: Weakening and warming of the European Slope Current since the late 1990s attributed to basin-scale density changes, Ocean Sci., 18, 549–564, https://doi.org/10.5194/os-18-549-2022, 2022.

Fiúza, A. F. G.: Upwelling Patterns off Portugal, in: Coastal Upwelling Its Sediment Record, edited by: Suess, E. and Thiede, J., Springer US, Boston, MA, 85–98, https://doi.org/10.1007/978-1-4615-6651-9_5, 1983.

This quote is also ambiguous without references: "On the continental shelf ocean dynamics are characterized by shorter timescales and spatial scales," Especially the spatial scale. Since ocean meso-scale eddies can't get on the shelf.

This sentence was notably referring to the fact that the dominant spatial scale of baroclinic ocean mesoscale eddies (characterized by the first baroclinic deformation radius, e.g., Stammer, 1997; Chelton et al., 2011; Klocker and Marshall, 2014) is smaller on the shelf than in the deep ocean as it scales with the buoyancy frequency integrated over depth (Chelton et al., 1998; Hallberg et al., 2013; Lacasce and Groeskamp, 2020).

However, we modified this sentence to rather highlight the importance of coastal processes: *"On the continental shelf, higher frequency processes have a more leading role on sea level variability (e.g., Woodworth et al. 2019) and can lead to sea level variability of larger amplitude (due e.g., to tides, storm surges). Although spatial scales of ocean mesoscale dynamics are smaller on continental shelves than in the deep ocean (e.g., Chelton et al., 1998; Hallberg et al., 2013; Lacasce and Groeskamp, 2020), sea level along north-eastern Atlantic European coasts northward of 25°N can also be coherent over thousands of kilometers at decadal timescales (e.g., Calafat et al., 2014), related to coastally trapped waves (Hughes et al., 2019). Along-shore wind forcing is a major contributor to such coastal sea level variability (Calafat et al., 2012)."*

Stammer, D., 1997: Global characteristics of ocean variability estimated from regional TOPEX/POSEIDON altimeter measurements. J. Phys. Oceanogr., 27, 1743–1769, https://doi.org/10.1175/1520-0485(1997)027,1743:GCOOVE.2.0.CO;2.

Chelton D., R. Deszoeke, M. Schlax, K. El Naggar, and N. Siwertz, 1998: Geographical variability of the first baroclinic Rossby radius of deformation. J. Phys. Oceanogr., 28, 433–460, https://doi.org/10.1175/1520-0485(1998)028<0433:GVOTFB>2.0.CO;2

Hallberg, R., 2013: Using a resolution function to regulate parameterizations of oceanic mesoscale eddy effects. Ocean Modell., 72, 92-103, https://doi.org/10.1016/j.ocemod.2013.08.007.

LaCasce, J. H. and Groeskamp, S.: Baroclinic Modes over Rough Bathymetry and the Surface Deformation Radius, Journal of Physical Oceanography, 50, 2835–2847, https://doi.org/10.1175/JPO-D-20-0055.1, 2020.

Hughes, C. W., Fukumori, I., Griffies, S. M., Huthnance, J. M., Minobe, S., Spence, P., Thompson, K. R., and Wise, A.: Sea Level and the Role of Coastal Trapped Waves in

Mediating the Influence of the Open Ocean on the Coast, Surv Geophys, 40, 1467–1492, https://doi.org/10.1007/s10712-019-09535-x, 2019.

l.850: "skew surge" is not defined in the paper. I don't think many users of sea level information know what it is.

We added a definition of skew surge when it is first introduced:
*"A skew surge is the difference between the maximum observed sea level and the maximum predicted tide regardless of their timing during the tidal cycle – there is one skew surge value per tidal cycle (Pugh and Woodworth, 2014)"*.

Pugh, D. and Woodworth, P.: Sea-level science: understanding tides, surges, tsunamis and mean sea-level changes, 2nd ed., Cambridge university press, Cambridge, 2014.

l.854: MSL is not defined. Actually in the rest of the text mean sea level is written instead of the acronym.

MSL has been expanded to mean sea level.

l.860: decreased amplitude over which period? The previous sentence says that the trend depends on the period.

This sentence has been rephrased to specify the period:

*"An analysis of tide gauges with at least 25-yr of data since 1960 indicates that the amplitude of extreme skew surges tends to have decreased along the north-eastern Atlantic coast (Marcos and Woodworth, 2017)"*.

l.880-883: Since this review is about sea level, I would suggest to either mention the relation between those ocean circulation changes and sea level or not mention them as all. I also wonder if it is useful to catalogue papers that did downscaling without mentioning important conclusions. If there was important information for sea level then write it down explicitly otherwise do not cite the papers.

The text has been modified: *"Sterodynamic SLR, which includes global mean thermal expansion of the warming ocean, steric and dynamic SL changes induced by ocean circulations (Gregory et al., 2019) remains the dominant contributor to total SLR along the European Atlantic coast. Regionally downscaled projections of SL changes over parts of the north-eastern Atlantic have been produced (Gomis et al., 2016, Hermans et al., 2022, Chaigneau et al., 2022). Hermans et al. (2020) and Chaigneau et al., (2022) have demonstrated the influence of dynamical downscaling on projections of dynamic SL over the 21st century for the northwestern European region. Hermans et al. (2020) have found that projected changes in dynamic SL in the downscaled simulations are up to 15 cm lower than in the GCM simulations for the RCP8.5 scenario. These differences are notably observed in the Celtic Sea, which is poorly resolved in the coarse resolution GCMs. In Chaigneau et al., (2022), the impact of the regionalization on ocean dynamic SL projections is weaker due to forcings from a higher resolution GCM, including more spatial details. In the same study, the impact of bias correcting the GCM ocean*

*and atmospheric forcings on the regionally downscaled ocean dynamic SL projections is also highlighted."*

**6.2 North Sea**

l.999-1001: It is not clear which contributions this refers to and how this paragraph is linked to the previous one.

This paragraph has been rewritten to:

*"Changes in water depth and other non-astronomical factors, such as changes in stratification and large construction measures, affect tides along the North Sea coast (e.g., Jänicke et al., 2021; A. Jensen, 1984; J. Jensen and Mudersbach, 2007; Mudersbach et al., 2013; P. Woodworth et al., 2017), including in estuaries (e.g., Amin, 1983; Keller, 1901; Jiang et al., 2020) and harbors (e.g., Doodsen, 1924; Marmer, 1935; Schureman, 1934; Vellinga et al., 2014). However, a comprehensive and generalized analysis is still missing."*

**6.5 Baltic Sea**

l.1282-1283: What does "energetically insignificant" mean? The question to answer here is: what is the impact of seiches on sea level variability?

We have expanded the text: *"Seiches are not detectable as a peak in the spectrum. Combined with storm surges, seiches can lead to extreme compound events (Weisse et al., 2021)."*

l.1296-1306: In this paragraph there is no reference to Figure 6 with the trend in geocentric sea level from satellite altimetry while the Baltic Sea really stands out in that figure as the region with the largest rate of sea level rise, with most places rising faster than 5mm/yr. I think this should be discussed in this paragraph.

We added the reference to Figure 6. The following sentence explains why sea level rise in the Baltic Sea is probably larger than in the North Atlantic and elsewhere: *"Over the last 2-3 decades, global mean sea level rose at rates of 3-4 mm yr-1 (Oppenheimer et al., 2019; Weisse et al., 2021; Section 2.2; Figure 6). However, such rates are spatially non-uniform and include impacts of multidecadal variations in wind fields (Passaro et al., 2021)."*

l.1307: "no long-term rising trend was found for ESLs in the Baltic Sea compared to mean changes" I don't understand the logic of this sentence. Why "compared to mean changes"?

We revised the sentence: *"For the 20th century, ESLs in the Baltic Sea will probably not rise more than global mean SL."*

l.1320-1321: "Therefore, projections for this basin require high-resolution regional climate models". I think it would be useful to separate projection of MSL and projections of ESL. In lines 1285-1286 it is written "On time scales longer than 1 month, the mean sea level in the Baltic Sea approximately follows the sea level in Kattegat, outside the Baltic Sea" which would imply that there is no need for models to resolve the complex local physical processes to get the long term MSL projections right. It would be useful to clarify this contradiction.

*We modified the sentence "Therefore, projections of ESLs for this basin require high-resolution regional climate models."*

l.1329: "will approximately rise by about 87% of the global mean rate". This is a good one. If it is "approximately" and "about" then the very precise number is not appropriate.

*Thanks, we modified the sentence to: "will rise by about 90% of the global mean rate".*

**Conclusion**

l.1484: Two "Finally"

*Thanks, corrected.*

*« A major uncertainty for SLR remains attached to ice sheets instabilities and overall contributions, and more robust projections beyond 2100 are needed. Finally, the interpretation of regional SLR variations for local perceptions and decision making is also an area needing improvement. »*

---

## Author Comment (AC2)

**Reviewer #2**

In this manuscript, the authors present a comprehensive assessment of sea-level literature along European coasts. They use literature and data from the literature to summarize the state of the art. This is, of course, a huge effort, that should be very helpful to both scientists, planners and decision makers. However, in its current form, the manuscript is, in my opinion, not ready for publication. The use of acronyms is rather wild, the structure at times unclear and heterogenous among subsections and some important literature is missing. It seems to me that authors have worked individually on sections, but not enough effort has been undertaken to bring the different sections together. Therefore, I recommend major revisions. I'll further clarify these points below. Please also note that I could only read this massive manuscript once, so the authors should really make sure that everything is in shape for resubmission.

We warmly thank the reviewer for this review and for the time spent reading our manuscript and providing constructive feedback. We accounted for the comments and address them below.

**General comments**

- Acronyms are a mess (to be honest). The number of acronyms is huge and not homogeneous throughout the manuscript (way too many to specify them here). I feel that the readability would be way better if the authors would decide on a few really important acronyms that are then properly introduced and used consistently used throughout the manuscript.
  Acronyms were introduced in the abstract if used several times there (SLR, RSLR, GMSLR, ESL, GIA). They were then re-introduced in the main text at their $1^{st}$ occurrence. We chose not to use acronyms in titles/ subtitles. The number of acronyms has been reduced (from 52 to 33). The list of acronyms and their expansion is now provided in Annex 1.

- The subsections in chapter 6 are very heterogeneous, both in terms of structure and content. For instance, some sections summarize VLM under past sea-level changes, while others use a full subsection (differently positioned among chapters). All chapters present the same type of figure, which is useful, but the figures are only discussed for the MedSea and the concepts presented are not really introduced. I also could not find references to these figures in some chapters. I would suggest introducing basic concepts, such as the use of the hybrid sea level reconstruction (Dangendorf et al., 2019), at the very beginning and report the associated numbers (also in reference to the figures) homogeneously for each section; also, in context to numbers presented in other literature. The same applies to the use of VLM information, which seems to follow Oelsmann et al. (2024; https://www.nature.com/articles/s41561-023-01357-2; not yet referenced)?
  Subsections in chapter 6 are dedicated to European regional seas. As main processes and topics to be discussed can differ across European regional seas, we do not necessarily aimed at having the same structure / content in each subsection, rather emphasizing key processes per European sea. However, we thank the reviewer for their comment and aimed at better homogenizing the structure of each subsection.

  Sections 6.1 to 6.5 are now organized more similarly with 1. General context, 2. Past sea level changes, 3. $21^{st}$ century projections. The section on the Mediterranean Sea is an exception, as we kept the sub-sections on Medicanes and Meteotsumanis, as these are more Mediterranean Sea specific, and both the past, present and future conditions are discussed.

*In addition, we now introduce the European regional seas discussed in Section 6 in the introduction of the section, with a figure (Figure 14 in the revised version) showing their geographical coverage. Basic concepts used in section 6 (such as the use of the hybrid sea level reconstruction by Dangendorf et al., 2019 and VLM by Oelsmann et al., 2024 –not yet published at the time of the 1st submission of this manuscript-) are also provided at the beginning of the section. Table 3 has be moved at the beginning of Section 6.*
*Figures 15-19 (formerly Figures 14-18) are now referenced in the different subsections of Section 6, and figures on VLM and 50-yr return levels of the recent past are provided in each subsection.*

- Somewhat related to points 1 and 2, the authors make use of extreme value statistics very heterogeneously throughout the manuscript. While the abstract and parts of the text frequency mention "historical centennial events", all figures in section 6 use water levels corresponding to a 50-yr return period. I would suggest homogenizing that. I would also suggest to properly introduce jargon, e.g., amplification factors, very clearly. It would also not hurt to explain the meaning of the amplification factors in the caption of Figure 12 in support of readability again.
  *Amplification factors were defined at the beginning of section 5.3.1.*
  *"Projections of future changes in ESLs due to SLR are often reported as factors by which the probability of a certain ESL will increase (called amplification factors; Buchanan et al., 2016; Fox-Kemper et al., 2021; Frederikse, Landerer, et al., 2020; Hermans et al., 2023; Jevrejeva et al., 2023; Lambert et al., 2020; Oppenheimer et al., 2019; Rasmussen et al., 2018; Tebaldi et al., 2021; Wahl et al., 2017) or as the height by which coastal defenses need to be raised to restore the historical flood probability (called allowances; Hunter, 2012; Hunter et al., 2013; Slangen et al., 2017; Woodworth et al., 2021)."*

  *Return periods of 50-yr were kept in section 6 for practical reasons and to cover another return period since 100-yr return levels of the historical periods are also discussed in other sections. 50-yr return levels are used throughout section 6.*

- The concept of relative and absolute sea level changes is not properly used in terms of the hybrid sea level reconstruction presented in section 6. The reconstruction contains Gravitation, Rotational and Deformational effects related to present day barystatic mass changes. Thus, even after the removal of GIA, it still contains solid earth components. These can make an important contribution to relative sea level, particularly in the Nordic Seas (as mentioned for the Baltic Sea). Therefore, I would suggest using the term: Relative sea level after the removal of GIA.
  *Thanks, done in the captions of Fig 15-19 and also specified in the introduction of Section 6.*

- The role of the NAO is in my opinion overstated in this manuscript. First, it is the dominant atmospheric mode only during winter. Second, many publications have shown that it only explains a fraction of the variability (e.g., Jevrejeva et al., 2006: https://onlinelibrary.wiley.com/doi/abs/10.1111/j.1600-0870.2005.00090.x), which is either due to the presence of other important teleconnection patterns (e.g. Chafik et al., 2017: https://www.mdpi.com/2077-1312/5/3/43) or more complex oceanic processes that are communication through ocean circulation and therefore integrated over time.

- There are some important references missing for some of the subsections that are provided in the specific comments below.
  *We address them in the comments below.*

**Specific Comments:**

Line 75: To be consistent with the concepts in Gregory et al. (2020), I would suggest introducing the term sterodynamic at this point, which is also used in later parts of the manuscript.

Done. We added the sentence: *"When combined together, the global mean steric SL change and ocean dynamic SL changes are called sterodynamic SL change (Gregory et al., 2019)."*

Line 98: I am not sure what the text in the parenthesis is referring to

This was referring to a companion paper (same special issue), that was only submitted at this stage. The paper is now referred to Jimenez et al., 2023.

Jiménez, J. A., Bonaduce, A., Depuydt, M., Galluccio, G., Van Den Hirk, B., Meier, H. E. M., Pinardi, N., Pomarico, L. G., Vazquez Riveiros, N., and Winter, G.: Sea Level Rise in Europe: Knowledge gaps identified through a participatory approach, https://doi.org/10.5194/sp-2023-34, 2023.

Line 103: I would suggest: "These contrasting atmospheric and oceanic environments lead to…"

Agreed, thanks.

Line 105: I guess you mean SL changes; SLR changes mean acceleration

This has been reformulated:
*"Here, the state of knowledge of observed and 21$^{st}$ century projected changes in mean and extreme SL is documented for European basins as part of the Knowledge Hub on Sea Level Rise Assessment Report".*

Section 2.1.: Although summarizing the last IPCC report, I think it is important to give proper references to the original sources in the text. Just as an example, the statement in line 127 should cite Kopp et al. (2016, https://www.pnas.org/doi/full/10.1073/pnas.1517056113)

While we are sympathetic to the reviewer's comment we believe that is it neither practical nor desirable to include comprehensive line-of-sight from the IPCC summary presented here to the underlying supporting literature. For our purposes, the IPCC assessments are taken as a "point-of-departure" for the European-focused report we present here, building on those findings. We have included the following additional sentence in the opening paragraph of section 2.1 to make our stance clear in this regard:
*"The IPCC reports synthesize a huge body of literature, and we refer the reader to the above assessment reports and references therein for further discussion on the topics summarized in this section."*

Line 132: Recent literature suggests that the acceleration started in the 1960s/1970s and was initiated by steric expansion, while mass input from the ice-sheets came into play over the past few decades.

This section summarizes IPCC SROCC and AR6 main findings, focusing here on the acceleration of GMSLR over the past few decades. We added the sentence *"SLR has accelerated since the late 1960s."*.
In section 4.2 - Sea Level Budget, we discuss further recent literature on acceleration of GMSLR, with the following sentence:
*"Recent studies show that GMSLR started to accelerate in the 1960s/1970s, initiated by an acceleration of thermosteric SLR due to an intensification and a basin-scale equatorward shift*

*of Southern Hemispheric westerlies and induced increased ocean heat uptake (Dangendorf et al., 2019). Since the 1990s, accelerated ice mass loss, mostly from the Greenland icesheet, has also contributed to the GMSLR acceleration (Dangendorf et al., 2019, Frederikse et al. 2020b)".*

Line 135-139: I am not sure what this statement means for Europe. Tropical cyclones primarily affect the other Western Atlantic region.
True. We removed this sentence.

Line 175: GIA is more than only VLM, geoid contributions are significant particularly in northern Europe
True. We have changed the sentence to also include gravity and rotational changes:
*"A key driver of these spatial variations is GIA: the ongoing GRD response to past ice mass changes"*

Line 223: Forecasted (i.e., initialized from observations) or projected sea levels?
The web tool "Se havnivå i kart" allows to visualize both present-day, forecasted sea levels (provided by the Norwegian Meteorological Institute-MetNO) and projected sea levels.
An example is provided here by clicking on the Tromsø tide gauge:

https://www.kartverket.no/en/at-sea/se-havniva/result?id=1082341

Information in the "Tides and Water Level" tab is related to observations, predicted tides and forecasted water levels, while information in the "Sea Level" tab is related to sea level projections.

Line 262: What are low-frequency global mean sea level records? What is low frequency, what is global mean?
The sentence :
*"Low frequency global mean SL records at tide gauges at monthly and yearly frequencies are obtained by national providers and compiled and distributed by the GLOSS Permanent Service for Mean Sea Level (www.psmsl.org) (Holgate et al., 2013)."* has been reformulated to:
*"Monthly and annual mean SL records from tide gauges are obtained by national providers worldwide and compiled and distributed by the GLOSS Permanent Service for Mean Sea Level (www.psmsl.org) (Holgate et al., 2013)."*

Figure 3: Might make sense to add the longest period covered by those tide gauge records.
Figure 2 already shows the length of PSMSL tide gauge records. Doing it for GESLA in Fig 3b would provide a very similar figure.

Figure 4: I suggest to add uncertainty bands to these plots.
Done. Fig 4 caption has been updated accordingly.

Line 301: sea level dynamics are highly….
Corrected.

Figure 5: What's the meaning of the two periods; why only since 2001?

Two 10-yr periods were selected to highlight the changing decadal trend of global mean sea level.

Line 322: Cloud and rain patterns lead to changes in thermal expansion?
Refers to the following sentence of the submitted manuscript: *"On interannual time scales the global mean SL record shows significant variations which are mostly generated by El Nino Southern Oscillation events during which the cloud pattern and the rain pattern are changed leading to changes in thermal expansion and land water storage respectively (e.g., Cazenave et al., 2014; Hamlington et al., 2020)."*

During El Nino events, the global hydrological cycle and atmospheric circulation are altered, with modifications of cloud cover and rain patterns. During El Nino, more precipitation occurs over the ocean (mostly in the tropics), resulting in a tendency for land water deficit. As a result, global land water storage is reduced during El Nino, and GMSL is temporarily increased.
It has been suggested that steric and barystatic effects both have sizeable, or even comparable, contributions to the GMSL budget during ENSO (Piecuch and Quinn, 2016; Hamlington et al., 2020), with a temporarily increased steric contribution to GMSL during the warm phase of ENSO and accumulated heat in ocean (mostly upper tropical Pacific Ocean).

The sentence has been reformulated to:
*"On interannual time scales, the global mean SL record shows significant variations which are mostly generated by El Nino Southern Oscillation events and its influence on the ocean heat content and global hydrological cycle. During El Nino events, the global mean SL is temporarily increased due to both an increase in ocean mass and in ocean thermal expansion (e.g., Cazenave et al., 2014; Piecuch and Quinn, 2016; Hamlington et al., 2020). Indeed, during El Nino events, more precipitation occurs over the ocean (mostly in the tropics), resulting in a temporary increase in the barystatic component of global mean SL. In addition, the ocean heat content temporarily increases during El Nino, with a dominance of the Tropical Pacific Ocean, leading to sizeable increases in global mean steric SL."*
Piecuch, C. G. and Quinn, K. J.: El Niño, La Niña, and the global sea level budget, Ocean Sci., 12, 1165–1177, https://doi.org/10.5194/os-12-1165-2016, 2016.

Line 330: What is an "effective temporal resolution", and why is it 34 days? Isn't every along-track point tracked every 10 days?
Although gridded altimetric products are provided as daily means over a 1/4° grid from a combination of satellite missions (amongst which the Topex/Poseidon, Jason1-3 and Sentinel-6 with a 10-d revisit time), the dynamical content of these maps does not have full 1⁄4° spatial and 1 d temporal resolutions due to the filtering properties of the optimal interpolation. The effective resolution corresponds to the spatiotemporal scales of the features that can be properly resolved in the maps. This is fully explained in Ballarotta et al., (2019) as cited in the text.
The text has been reformulated to:
*"To analyze sea level changes at regional scales, gridded altimetric product can be used. Although such products are provided as daily maps on a 1/4°x 1/4° grid, the dynamical content of these maps does not have full 1/4 ° spatial and 1-d temporal resolutions due to the filtering properties of the optimal interpolation. The effective resolution corresponds to the spatiotemporal scales of the features that can be properly resolved in the maps. The temporal effective temporal resolution has been estimated to around 34 days (spatially varying), and the*

*effective spatial resolution has been estimated to range from 100 to 200 km in the north-eastern Atlantic, and from 90 to 160 km in the Mediterranean and Black Seas (Ballarotta et al., 2019)."*

Line 349: estimates
Thanks, done.

Line 384: InSAR can image the spatial pattern of VLM…
Done.

Line 413: I think this statement refers to extreme sea levels rather than storm surges. Jänicke et al (2021) assessed changes in the mean tidal range, so those do not correspond to any changes in storm surge. An additional reference here is Dangendorf et al., 2013: https://link.springer.com/article/10.1007/s10236-013-0614-4
Thank you. We have substituted the reference as suggested.

Line 467: What ranges do these numbers refer to?
We modified the sentence to:

*"Fox-Kemper et al. (2021) assess the Antarctic contribution to global mean SLR in 2300 (without MICI) to range between –0.14m and +0.78m (17-83 percentile) for a low emission scenario (SSP1-2.6, and to range between –0.28m and +3.13 m for a very high emission scenario (SSP5-8.5)."*

Line 513: I am not sure whether that statement is correct. This is period dependent and does not hold for each individual contributions, right? Maybe state: Since the late 1960s, the sum of individual contributions have led to a persistent acceleration.
The sentence *"Since the late 1960s, all contributions to SLR accelerated (Dangendorf et al., 2019)"* has been reformulated to:
*"Recent studies show that GMSLR started to accelerate in the 1960s/1970s, initiated by an acceleration of thermosteric SLR due to an intensification and a basin-scale equatorward shift of Southern Hemispheric westerlies and induced increased ocean heat uptake (Dangendorf et al., 2019). Since the 1990s, accelerated ice mass loss, mostly from the Greenland icesheet, has also contributed to the GLMSLR acceleration (Dangendorf et al., 2019, Frederikse et al. 2020b)".*

Line 548: To be fair, I would add that uncertainty exists in this region with respect to VLM estimates. For instance, the reported numbers in Frederikse et al. (2016) for a slightly different period are smaller than in Dangendorf et al. (2021).
This has been added, thank you.

Line 589 following: Treu et al. (2024; https://essd.copernicus.org/articles/16/1121/2024/) produced the first full sea level reconstruction based on a combination of different reconstruction and modelling approaches.
Treu et al (2024) uses hydrodynamic model outputs to account for extreme sea levels. We therefore believe that this approach lies in the former group discussed in the paragraph above. The source data for high-frequency changes is Muis et al (2020), already cited in there.

Section 5.2.: I feel that MISI and MICI might be explained more thoroughly in terms of the associated physics.
We have added a short explanation on these mechanisms in section 5.2

Section 5.3.2.: The literature summary is rather recent. I would suggest incorporating early attempts such as Woth (2005): https://agupubs.onlinelibrary.wiley.com/doi/10.1029/2005GL023762 , Woth et al. (2005): https://link.springer.com/article/10.1007/s10236-005-0024-3; WASA (1998): https://www.webofscience.com/wos/woscc/full-record/WOS:000073809200001?SID=USW2EC0FB7RmlUpG8iI3Cn6Fxd3jY; Sterl et al. (2009): https://os.copernicus.org/articles/5/369/2009/

L780 (in section 5.3.2) has been modified to include early attempts. *"In the southern North Sea, Jevrejeva et al., (2023) showed an increase of +50 cm in extreme storm surges and waves under a low-probability high-impact scenario (Figure 13), in line with early attempts to account for dynamic changes in storm surges (Woth 2005; Woth et al., 2005)."* The Woth et al. (2005) reference has also been added in Sect., 6.2. WASA (1998) and Sterl et al., 2009 were discarded as no significant changes in storm surges (and waves) were found for the past period for WASA (1998) and 21st century projections for Sterl et al., (2009).

Line 782: Here and throughout the manuscript, SLR is handled as the only driving factor behind changes in tides and corresponding extremes. However, recent literature (including the cited Jänicke paper) suggests significant contributions by density changes. Thus, it should be noted that the reported numbers in projections are lacking those mechanisms.
The sentence has been modified, as the mentioned studies have considered the spatial variations of MSL and not just SLR:
*"For example, tidal ranges may change by several tens of centimeters in Europe depending on the spatial variability of SLR, considered SL drivers and the inclusion of flooding of low-lying topography (Haigh et al., 2020; Idier et al., 2017; Pickering et al., 2017)."*
The reference Jänicke et al., 2021 has also been added L789 reporting changes in tides over the historical period.
Note that another sentence addresses changes in tides in section 6.2.2:
*"Changes in water depth and other non-astronomical factors, such as changes in stratification and large construction measures, affect tides along the North Sea coast (e.g., Jänicke et al., 2021; A. Jensen, 1984; J. Jensen and Mudersbach, 2007; Mudersbach et al., 2013; P. Woodworth et al., 2017), including in estuaries (e.g., Amin, 1983; Keller, 1901; Jiang et al., 2020) and harbors (e.g., Doodsen, 1924; Marmer, 1935; Schureman, 1934; Vellinga et al., 2014). However, a comprehensive and generalized analysis is still missing."*

Line 830: Where are these numbers coming from? From the trend assessment tool? Why not plot them for those periods? Also, the reference PSMSL is not in the list
This sentence has been deleted.

Line 834: Is this true? Isn't GIA, or VLM in general, the dominant driver of regional sea level trends in the region?
This sentence discussed the driver of the spatial heterogeneity of RSLR over this region, not - contributions to basin-mean RSLR (cf Fig 9.26 IPCC AR6). GRD effects are mentioned in the sentence.

Line 885: What's the historical centennial climate extreme event?
This is defined in Section 5.3.1. A reference to that section was added to clarify.

Section 6.2.2.: An important reference here is Wahl et al. (2013): https://www.sciencedirect.com/science/article/abs/pii/S0012825213000937 that has not been included yet. I am also missing all the Woodworth (e.g., https://academic.oup.com/gji/article/213/1/222/4757068 and some more earlier assessments) and Hogarth (e.g., https://www.sciencedirect.com/science/article/pii/S0079661120300720) UK sea level assessments that the authors should be very familiar with.

Thanks for your suggestion, we have added the following two sentences and referenced these and other relevant papers:

*"Observed trends vary by one to three tenths of a mm/yr between different parts of the North Sea region (Wahl et al., 2013). Several assessments of sea level trends around the British Isle' (e.g., Woodworth et al., 1999, 2009; Haigh et al 2009; Woodworth, 2017; Hogarth et al. 2020, 2021), include tide gauge sites in the North Sea, and again observed trends typically range between one to three tenths of a mm yr⁻¹, over the last century".*

Hogarth, P., Pugh, D.T., Hughes, C.W, Williams, S.D.P.: Changes in mean sea level around Great Britain over the past 200 years, Progress in Oceanography, 192, 2021.

Hogarth, P., Hughes, C.W, Williams, S.D.P., Wilson, C: Improved and extended tide gauge records for the British Isles leading to more consistent estimates of sea level rise and acceleration since 1958, Progress in Oceanography, 184, 2020.

Haigh, I., Nicholls, R.J., Wells, N.: Mean sea level trends around the English Channel over the 20th century and their wider context, Continental Shelf Research, 29 (17), 2083-2098m 2009.

P.L. Woodworth, M.N. Tsimplis, R.A. Flather, I. Shennan: A review of the trends observed in British Isles mean sea level data measured by tide gauges. Geophysical Journal International, 136 (3), 651-670, 1999.

P.L. Woodworth, R.M. Teferle, R.M. Bingley, I. Shennan, S.D.P. Williams: Trends in UK mean sea level revisited. Geophysical Journal International, 176 (1) (2009), pp. 19-30

Philip L Woodworth, Sea level change in Great Britain between 1859 and the present, Geophysical Journal International, Volume 213, Issue 1, April 2018, Pages 222–236, https://doi.org/10.1093/gji/ggx538

Line 1015: Please quantify "large effect"
The order of magnitude was added : "*up to 30% of the total sterodynamic SLR simulated for the 21st century"*

Line 1070: I think there is a link missing?
The reference to NKG has been cut so a link is not needed anymore.

Line 1099: Isn't GIA the biggest factor in those areas?
We agree this is poorly formulated so have change to:

*"Projections for the European Arctic indicate the region will experience a SL change somewhat below the global average rise (e.g. Simpson et al., 2017; Table 3). Apart from GIA, several components of projected SL changes are relevant for the European Arctic...."*

Line 1171: Black Sea sea level

Done, thanks.

Line 1180: This information should go at the beginning of section 6 as it is used for the figures in all regions. I would also suggest to pick the results up in each section in the context of the corresponding literature. Furthermore, the approach requires a bit more introduction in general. Is it based on Oelsmann et al. (2024; https://www.nature.com/articles/s41561-023-01357-2)?

VLM linear trends shown in Figures 15-19 are indeed based on Oelsmann et al. 2024. At the time of submission of our manuscript, the study by Oelsmann et al. Was not yet published. As it is now, we introduce the study at the beginning of Section 6.

Line 1287: Again, shouldn't GIA be the dominant factor, leading to far lower than global projections? Shouldn't that factor first be emphasized given its importance?

We agree and rephrased the paragraph *"It is expected that SLR in the southern Baltic Sea approximately follows the projected GMSLR (or slightly less) due to the melting of ice sheets and glaciers and the expansion of the warming water (Hieronymus and Kalén, 2020; Meier, et al., 2022; Pellikka et al., 2020; Weisse et al., 2021). However, in the northern sub-basins of the Baltic Sea, GIA (Section 3.3) is the dominant driver (Ekman, 1996)."*

Section 6.5.2.: This section is lacking some literature: e.g., Gräwe et al. (2019): https://journals.ametsoc.org/view/journals/clim/32/11/jcli-d-18-0174.1.xml; Donner et al. (2012): https://npg.copernicus.org/articles/19/95/2012/; some of the Ekman studies (https://psmsl.org/products/author_archive/ekman_2003.pdf and references therein)

A discussion of all related articles is out-of-scope of this review as we rely on the most recent Baltic Earth assessment report by Weisse et al. (2021). Following the suggestions by the reviewer, we added the references Donner et al. (2012) and Gräwe et al. (2019).

---

## Author Comment (AC3)

**Community Comment - CC4 - Hartmut Hein**

The text is a very comprehensive and certainly almost complete review on the topic of sea level rise in Europe. I enjoyed reading the text.

We thank you for your feedback and for providing comments on our manuscript.

However, I note that the chapter "3.3 Vertical land motion including human induced subsidence." the "human induced subsidence" is only mentioned in the title and in the introductory sentence without further references. However, since the time scale of "human induced subsidence" is considerably shorter than that of natural subsidence, the combination can result in non-linear subsidence behavior over time. This topic is crucial for interpreting observed sea level data. Numerous publications address human-induced subsidence. For instance, Candela and Koster (2022) argue that the complexity arises from the overlaying of different sources of human-induced subsidence. Smith et al. (2019) demonstrate the time-dependent changes in subsidence caused by deep gas production at the Groningen field.

We have removed "human induced subsidence" from the subtitle, since VLM can have a number of causes.

In the text we have added the following for background:
*"As discussed, subsidence can be natural or human induced. Gas production at the Groningen field, for example, situated in the northeast Netherlands, has caused measurable subsidence since the 1960s (Smith et al., 2019). Understanding the processes causing subsidence and their respective timescales is crucial for sea-level studies. This can be particularly challenging in areas where subsidence has multiple causes and requires to disentangle the individual contributions to VLM (Candela and Koster, 2022)."*

It is not within the scope of this paper to look in detail at the local causes of VLM, but rather we give a regional picture of what's going on across Europe.

Candela, T., & Koster, K. (2022). The many faces of anthropogenic subsidence. Science, 376(6600), 1381-1382.

Smith, J. D., Avouac, J. P., White, R. S., Copley, A., Gualandi, A., & Bourne, S. (2019). Reconciling the long-term relationship between reservoir pore pressure depletion and compaction in the Groningen region. Journal of Geophysical Research: Solid Earth, 124(6), 6165-6178.

Row 318 states that the average sea level rise in Europe is slightly higher than the global average. This may contradict the study by Frederikse et al, 2020b cited two lines earlier. Due to the proximity of Greenland, lower rates of rise are expected, especially in northern Europe. In their interpretation, Frederikse et al, 2020b point out the effects of gravity, rotation and deformation on the trends in the North Atlantic subpolar region. Your statement that sea level rise in Europe is slightly above the global average refers to a period that includes only the satellite domain. It should be noted that a period of 30 years on a high-resolution scale is not suitable for making meaningful trend statements. Instead, it mainly reflects multi-decadal variability.

Sea level rise is due to both density changes of the ocean (steric sea level) and the mass addition from ice sheet and glaciers melt. In the future, when ice sheet melt will dominate sea level rise, indeed we expect sea level to rise slower than the global mean rate in Europe because of the gravity field adjustment around Greenland. But for the time being, the steric contribution is still the dominant contribution to regional sea level rise and this contribution is particularly high around Europe which makes current sea level to rise faster in Europe than on average. This steric contribution is due both to the internal variability and the forced signal (due to anthropogenic forcings). As we discuss geocentric sea level based on altimetry in this section, the statement here on European sea level refers indeed only to the period 1993-2021.

To clarify this point, we changed the statement to *"In Europe, geocentric SL trends since 1993 have been contrasted with high SLR in the Baltic Sea (see section 6.5 and Fig 7 for RSLR in the Baltic), low sea level rise in the Mediterranean Sea and a SLR close to the global mean rate in the Atlantic sector on average (Figure 6)"*

---

## Author Comment (AC4)

**Comments on sp-2023-36 by Sergiu Dov Rosen***

* Retired Senior Scientist, Haifa, Israel. Previously MedGLOSS Coordinator, member of GLOSS Group of Experts, chair/co-chair of Working Group 3 of ICG/NEAMTWS on Sea Level Data Collection and Exchange, incl. Offshore Tsunami Detection and Instruments.

Estimeed authors,

First I wish to congratulate the authors for the excellent paper produced, covering the full range of processes leading to SLR and its future assessment for European seas.

We thank you for providing feedback on our paper.

My comments are mainly related to the processes contributing to SLR and ESL in the Levantine basin of the Mediterranean Sea, particularly in the Nile littoral cell, covering the South Eastern Mediterranean coasts from Egypt Nile delta to Israel. They are listed below and then a short list of references.

1. The paper mentioned that one of the factors affecting the SL is the ocean circulation and that regional models of higher resolution are needed to improve the assessments (page 2, lines 46-47). However, in the South Eastern Mediterranean, an important contribution seems that it has been overlooked as it was not mentioned in the paper. I refer to the discharge of Red Sea water via the Suez Canal in the Mediterranean Sea, which, since the beginning of the 1990's increased significantly, due to the Canal deepening and widening a few times, as already indicated by Rosen (2014). The Red Sea waters flowing to the Mediterranean, pass on their way through the Bitter Lakes region, where they collect salt and heat, entering the Mediterranean much saltier and warm than both the Red Sea and the Mediterranean Sea, sinking toward the bottom of the Mediterranean. My rough estimate, based on published cross sections and flow speeds ( Eid F.M., et al., 1997; Alam El-Din K.A., et al., 1999; Elzeir M.A., and T. Hibino, 1999; Suez Canal Authority) is of a discharge of about 100 to 120 $Km^3$/year. Thus, a significant amount of salt has been added to the Mediterranean water volume. Published studies indeed confirm the presence of increased salinity waters off the SE Levantine basin. Furthermore, the warmer waters add to the heating of the lower waters of the region thus contributing to the thermosteric contribution of the Mediterranean. Unfortunately, so far model studies of the SLR in the Mediterranean did not include this contribution which lacks their assessments. This however has been observed from the analyses at the GLOSS station 80 Hadera which indicated a faster SLR of 5.9 mm/year for the period 04/1992-003/2014 (and about 5 mm/year for the period 04/1992-03/2023) relative the global average SLR rate for the same period of 3.5 mm/year. Furthermore, another flow change apparently has not been incorporated in the models. I refer to the Nile river discharge to the Mediterranean, which until 1965, when the High Aswan dam was finished, was of millions of $m^3$/year, reduced to about 5,000 $m^3$/year ever since. However, some part of the Nile water of unknown volume, used for agricultural use in the Nile delta coast, has been seeping to the Mediterranean, as indicated by satellite monitoring.

We thank the reviewer for the detailed explanation of the Suez Canal. We have added a sentence in section 6.4.1 about the impact of the warm and salty waters on the steric signal of the Southeastern Mediterranean basin. Regarding the impact on sea level changes of these water inputs, it is not clear from figures 3 (tide gauges) and 6 (altimetry). We prefer not to quote the relative sea level trend from a tide gauge, as it does not differ from many other tide gauges

around the basin. Altimetric sea level trends do not support either a clear signal on geocentric sea level from the water flowing from the Red Sea or the Nile.

2. In the introduction of the paper (page 3, lines 61-62) flood coastal plains are mentioned. In my opinion a clarification would be appropriate, indicating that in the paper the flood plain is defined as that due only to the penetration of sea level. However, the true flood plain is of higher elevation, induced by reduced river flow discharge into the sea, caused by the reduced flow gradient due to elevated sea water levels at the river mouths during river floods.

We modified the sentence to:

*"Sea level rise (SLR) is a major concern for Europe, where more than 50 million people live in low-elevation (≤ 10m) coastal zones and 30 million in the 100-year event marine coastal flood plains (Neumann et al., 2015)."*

Coastal flooding due to sea level rise as well as pluvial/fluvial and marine compound flooding are addressed in a companion paper (van de Wal et al., 2023).

van De Wal, R., Melet, A., Bellafiore, D., Vousdoukas, M. I., Camus, P., Ferrarin, C., Essink, G. H. P. O., Haigh, I. D., Lionello, P., Luijendijk, A. P., Toimil, A., and Staneva, J.: Sea Level Rise in Europe: impacts and consequences, State of the Planet, submitted, 2023.

3. In section 2.1 (page 6, lines 124-125) the article refers as stated, mainly to the two IPCC latest reports, which were published a few years ago. However, it seems to have disregarded important new information published in November 2023, indicating potential much higher and faster sea level rise from melting ice caps (ICCI, 2023).

Indeed, section 2.1 focuses on IPCC SROCC and AR6.

The ICCI report of 2023 is not a peer reviewed report and as such not as rigorous as IPCC reports so it does not seem to be warranted to use it in the introduction here. At the same time, it is true that the latest IPCC report of Working Group 1 is already a few years old by now and science has made progress since then. The higher values mentioned in the ICCI report for Greenland based on the paper by Beckmann and Winkelmann (2023) are therefore discussed in section 4:

*"Along a similar line of reasoning, Beckmann and Winkelmann et al. 2023 argued a substantial increase of mass loss in Greenland if extreme warm summers are added to the projections."*

Thanks for pointing this out.

Beckmann, J. and Winkelmann, R.: Effects of extreme melt events on ice flow and sea level rise of the Greenland Ice Sheet, The Cryosphere, 17, 3083–3099, https://doi.org/10.5194/tc-17-3083-2023, 2023.

4. a. Both in the IPCC AR6 report and in your article (page 6, line 150), future SLR are referred relative to the average SL during the decade 1995-2014. The reasoning to refer sea level rise to a decade (1995-2014) has not been clearly motivated. Indeed one possible explanation I found is due to similar global average temperatures as during the Last Interglacial stage, about 125,000 years ago (global average sea surface temperature was about 0.5° ± 0.3°C (0.9° ± 0.5°F) above

the preindustrial level [that is, comparable to the average over 1995–2014, when global mean temperature was about 0.8°C (1.4°F) above the preindustrial levels] (USGCRP, 2017). Personally I am not convinced that such reasoning is adequate for selecting the average of that period for reference since the accuracy of the temperatures during the historic times is somewhat questionable. I would rather prefer reference to a certain year or series of calm years without strong NO and El Ninio/La Ninia activities.

On page 6, line 150 of the submitted version, we report IPCC AR6 projections of global mean sea level rise.

In IPCC AR6, the choice of the 1995-2014 baseline to inform on future changes is motivated by the fact that:

- Data are averaged over a 20-yr period to define a climate state. Indeed, 20 year-mean enables to remove most of the large internal variability which operates on interannual time scales and to focus on the climate state. This way we can compare a current climate state with projected climate states in the future without a large uncertainty induced by the internal variability
- The last two decades of the CMIP6 historical simulations are therefore selected. At the time of production of CMIP6 simulations, historical forcings were available until 2014. Projections therefore started in 2015. The last two decades of historical runs, corresponding to the 'recent past baseline period' used as a reference to evaluate future projected changes. Therefore, the 1995-2014 period was used for IPCC AR6.

This is standard in IPCC reports and climate change community.

4b. Furthermore, the use of Gregorian calendary years is wrong in reviewer's opinnion, as division by calendary years (January to December) from climatic cycle point of view, has the potential to place time neighboring dependent ESL events occurring in the December-January period in different years in the northern hemisphere, potentially affecting statistics of ESL. Hence a different yearly division would be much more correct, such as from April to following March end of next year or from October to following next year September end. Personally I use the April to March yearly meteo-marine division.

We follow standard community reporting methodology with calendar years.

5. On page 12, in Figure 2, on the Israel Mediterranean coast are shown 3 stations two with blue dots representing recording periods of up to 25 years and in the middle, one green dot, hardly visible as covered by the blue dots, of record length over 25 years and less than 50 years. In fact, the green dot should be for the Hadera GLOSS station 80, with a high frequency record of over 30 years. Another green station should have been for Tel Aviv Yaffo, also with high frequency SL over 25 years.

Figure 2 is based on PSMSL tide gauge records. In this database, Tel Aviv Yaffo data record covers 2011-2022 (hence less than 25 years of data) and the Hadera data record covers 1992-2022 (hence more than 25 years of data).

https://psmsl.org/data/obtaining/map.html#metadataTab

Regarding the high-frequency records, unfortunately they are not included in GESLA dataset, as they are not publicly available in any of the repositories that are part of GESLA.

6. On page 29, Figure 10 shows Median relative sea level regional projections (medium confidence) from the IPCC AR6 report around Europe. I would like to suggest that the abscissa title be more explicit, stating the following text: Mean sea level by 2100 w.r.t 1995-2014 (m), based on medium confidence projections, i.e. excluding ice sheet processes associated with deep uncertainty.

We further specified the legend.

7. On page 57, section 6.4.6 on wave climate, the statement on line 1222 in my opinion may be misleading, claiming "In winter, mean and extreme waves are highest in the western Mediterranean,". The statement is true if one refers to European coasts only, but since in the paper it is stated that the report covers European seas, and since in many places the whole Mediterranean is shown, it would be appropriate to include the Levantine basin too, and then the correct statement would be that "in winter, mean and extreme waves are highest in the eastern Mediterranean followed by those in the western Mediterranean." Similarly, the statements regarding fetches and correspondingly the return periods for the 100-year return level would increase to over 8 m deep water significant wave height offshore the Israeli coast.

Figures 4 and 5 of the recent wave reanalysis of Barbariol et al. (2021, cited in our manuscript) show that mean and extreme waves in winter are higher in the western than in the eastern Mediterranean, though a local maximum of the mean wave height is present also in the Ionian Sea. Values along the coast of Israel are remarkably high, but figure 10 of Toomey et al. (2022, cited in our manuscript) shows that the highest 100-y return values are located along the North African coast of the western Mediterranean basin.

REFERENCES:

Rosen, S.D., 2014. From Tsunami Early Detection and Warning to Multi Hazards Early Detection and Warning, CIESM International Conference on East - West Cooperation in Marine Science, Sochi, Russia, 1-3 December 2014, Abstracts, http://www.ciesm.org/marine/sochi/abstracts/index.php

Eid F.M., et al., 1997. Sea-level Variation Along the Suez Canal, Estuarine, Coastal and Shelf Science (1997) 44, 613– 619.

Alam El-Din K.A., et al., 1999. The effect of the Suez Canal development on the tide and tidal current- model study, Conference Paper, November 1999. https://www.researchgate.net/publication/332396520

Elzeir M.A., and T. Hibino, 1999. Hydrodynamic Simulation Of The Suez Canal; A Water Body Connecting Two Open Seas. Annual Journal of Hydraulic Engineering. JSCE. VOL.43, 1999. February.

Suez Canal Authority, Canal Characteristics,

https://www.suezcanal.gov.eg/English/About/SuezCanal/Pages/CanalCharacteristics.aspx

ICCI, 2023. State of the Cryosphere 2023 – Two Degrees is Too High. International Cryosphere Climate Initiative (ICCI), Stockholm, Sweden ; https://www.iccinet.org/statecryo23.

USGCRP, 2017. Climate Science Special Report: Fourth National Climate Assessment, Chapter 12, Volume I [Wuebbles, D.J., D.W. Fahey, K.A. Hibbard, D.J. Dokken, B.C. Stewart, and T.K. Maycock (eds.)]. U.S. Global Change Research Program, Washington, DC, USA, 470 pp., https://repository.library.noaa.gov/view/noaa/19486

---

## Author Comment (AC5)

**Comment #2 Sergiu D. Rosen**

This is my second comment to the article preview discussion.

1st, I found a misspelling in my previous, whereas in place of decade it should be decades.

Thanks, noted.

2nd Another point forgot to mention is that due to SLR induced flooding. Due to SLR and particularly during extreme sea levels, the flow gradients of the rivers will decrease, potentially leading to increased floods. I believe this should be mentioned in the article.

Compound sea level and fluvial flooding is addressed in a companion paper:

van de Wal, R. S. W., Melet, A., Bellafiore, D., Vousdoukas, M., Camus, P., Ferrarin, C., Oude Essink, G., Haigh, I. D., Lionello, P., Luijendijk, A., Toimil, A., and Staneva, J.: Sea Level Rise in Europe: Impacts and consequences, State Planet Discuss. [preprint], https://doi.org/10.5194/sp-2023-38, in review, 2023.

---

## Author Comment (AC6)

**COMMUNITY COMMENT**

Dear Authors,

I extend my heartfelt congratulations to all of you for your outstanding collaborative work on the comprehensive analysis of sea level rise in Europe. Your research delves into crucial aspects of sea level rise due to climate change in future projections, and it is indeed an excellent contribution to the field.

I would like to bring to your attention that we recently published a paper specifically focusing on sea level rise in the Thrace Peninsula area, utilizing data from the IPCC report. While I noticed your mention of Marmara Sea level rise in your preprint, I believe there is still more to explore and emphasize regarding Marmara SLR. This region is home to over 20 million people, and their livelihoods depend on various aspects of the sea. More than half of the population resides in the vicinity of coastal areas, making it a critical area for in-depth investigation.

For further insights into our recent work, you can find our freshly published paper in Remote Sensing here:
https://www.researchgate.net/publication/376110912_Coastal_Vulnerability_Assessment_of_Thrace_Peninsula_Implications_for_Climate_Change_and_Sea_Level_Rise

Once again, congratulations on your remarkable work, and thank you for your significant contributions to the field. I wish you continued success in your research endeavors.

Best regards,

Mehmet Ozdes

Many thanks for your feedback.
The current paper on sea level rise in Europe does not go into as many details as your study does. To be consistent in the level of information provided across the paper, we therefore just added one sentence (before section "6.4.3. Vertical land motion"):

*"In their study on the Thrace Peninsula in Turkey, a vulnerable area to SLR bordered by the Marmara, Aegean and Black Seas, Ozsahin et al. (2023) recommend using local mean SL measurements. As highlighted by Kopp et al. (2014), this reflects the need for specific SLR information to generate more accurate projections of SLR."*

Kopp, R.E.; Horton, R.M.; Little, C.M.; Mitrovica, J.X.; Oppenheimer, M.; Rasmussen, D.J.; Strauss, B.H.; Tebaldi, C. Probabilistic 21st and 22nd Century Sea-Level Projections at a Global Network of Tide-Gauge Sites. Earth's Future, 2, 383-406, https://doi.org/10.1002/2014EF000239, 2014.

Ozsahin, E.; Ozdes, M.; Ozturk, M.; Yang, D. Coastal Vulnerability Assessment of Thrace Peninsula: Implications for Climate Change and Sea Level Rise. Remote Sens., 15, 5592. https://doi.org/10.3390/rs15235592, 2023.

---

## Author Response (AR1)

The handling editor thanks the authors for their submission of Chapter 2 "Sea Level Rise in Europe: observations and projections" and for their revisions of the paper based on feedback from two anonymous reviewers and three community comments. The authors have addressed the reviewers comments satisfactorily either by following the reviewers suggestions or convincingly arguing why they decided not to.

Based on my review of the revised manuscript, I recommend that this manuscript requires minor corrections before finalization. The corrections I suggest are mainly related to improving clarity and detailed below.

**We thank the handling editor for the extra corrections suggested to improve the paper.**

**1 Introduction**

L81: icesheets -> ice sheets
Done

**2 Summary of previous assessments**

L133: I think the two references can be in the same parentheses.
Indeed, done.

**3 Regional observations**

Figure 3, caption: the caption is referring to panel a and b but as far as I can see, the panels are not labelled. I suggest changing to left and right. Also, there is no "top right".
Thanks, this has been corrected.

Figure 4, caption: you refer to Figure 3b but there is no "b" in Figure 3.
Thanks, this has been corrected.

L301: sea level dynamics ARE highly...
Corrected, thanks.

Figure 5: Referring to a comment by Rev2: You do not refer to the two periods shown in Figure 5. I would ask you to explain them in the caption and add a sentence in the text to highlight the changing decadal trends.
The caption of Fig 5 has been updated to:

**Figure 5: Global mean SL measured by satellite altimetry since 1993 (red curve), shaded area represents the uncertainty and the dotted line shows a trend line with an acceleration. The annual and semi-annual periodic signals are removed and the timeseries is low-pass filtered (175 days cut-off). The timeseries is corrected for GIA using the ICE5G-VM2 GIA model (Peltier, 2004) to consider the ongoing movement of land. Over 1993-1998, global mean sea level is corrected for the TOPEX-A instrumental drift, based on comparisons between altimeter and tide gauges measurements (Ablain et al., 2017; Legeais et al., 2020). Over 1993-2022, the GMSLR trend is 3.29±0.33 mm yr$^{-1}$ (uncertainty at 90% confidence level) and the GMSLR acceleration is 0.11±0.06 mm yr$^{-2}$. Trends are also reported for the period 2001-2011 and 2011-2021 to highlight the changing decadal trend of global mean sea level. The shaded envelope indicates uncertainties (17$^{th}$-83$^{rd}$ percentiles). Data source: EU Copernicus Marine Service product (2019): Ocean Monitoring Indicator based on the C3S altimetric SL product. Credit: C3S/ECMWF/Copernicus Marine.**

L378: time -> temporal

Done.

L380: I think it would be useful to explain what you mean by "relative measure of VLM". Relative to what?

Repeat levelling determines changes in elevation across a network of points. In that way you can say that these measurements are made relative to themselves. By making repeat measurements, VLM can be determined across the levelling network. This is different to GNSS which is an "absolute" measure of VLM, as measurements are made in a geocentric reference frame (ITRF).

To improve its clarity, the sentence "*Historically, repeat levelling has been the main technique and gives a relative measure of VLM*" has been rewritten to:
"*Historically, repeat levelling has been the main technique. It determines changes in elevation across a network of points and gives a measure of VLM across the levelling network. The repetition of levelling also provides VLM measurements relative to past ones »*.

L433: Calafat et al. (2022) determined ... (remove "has")
Done.

**Section 4 Drivers of sea level rise and extremes**

I would suggest distinguishing more between past and present drivers and projected drivers. This section does a little bit of both. Section 4.1 covers the past and projections (ice sheets only), section 4.2 mostly the observational period and section 4.3 seems to be period-independent.

Section 4.1 could be shortened by moving the part on projections to section 5, particularly the last paragraph on ice shelf collapse.

We decided to dedicate Section 4 to processes and drivers of SLR and extremes, while Section 5 rather assesses projected SLR and extremes, although we acknowledge some overlap. As such, in Section 4, both the past and future periods can be tackled, depending on processes (such as ice sheets instabilities).

L463: Fig. 2.8b → Figure 8b
Done.

L464: make about → account for
Done.

L469-470: suggest rewriting to: The role of atmospheric dynamics is also uncertain.
Done.

L482: Figure 8c
Done.

L487: MICI has not yet been defined.
The sentence has been updated to : "Fox-Kemper et al. (2021) assess the Antarctic contribution to global mean SLR in 2300 (without **Marine Ice Cliff Instability possible**

**contribution, see section 5.2**) to range between –0.14m and +0.78m (17-83[th] percentiles) for a low emission scenario (SSP1-2.6, and to range between –0.28m and +3.13 m for a very high emission scenario (SSP5-8.5)."

L521: I think you can remove "which are the changes in the amount of water stored on land"
Done.

L529: replace "is" with "are"
Thanks, corrected.

L535: icesheets -> ice sheets
Done.

**Figure 9, legend: what is GMOM?**

The caption of Figure 9 has been updated to :

**Figure 9: Global mean SL (GMSL) budget from 2006 to 2021. Global mean SL is estimated by satellite altimetry (black curve, data from the Copernicus Marine Environment Monitoring Service). Global mean ocean mass (GMOM) change (sum of ice sheet mass loss, glaciers ice melt and land water storage changes) is estimated from GRACE and GRACE-FO (blue curve, data taken from the JPL, CSR, GSFC mascon solutions). Global mean thermosteric sea level (GMTSSL) change is estimated from Argo (Green curve, data taken from an ensemble of the NOAA, EN4, SCRIPPS and JAMSTEC Argo product). From Barnoud et al., 2021.**

**Section 5 Projections of sea level rise...**

L656-657: "Figure 11 regional SLR projections..." the first part of this sentence reads strange.

This sentence was updated to: "Figure 11 indicates the first decade in which the median projected regional SL change over European Seas has crossed a certain threshold (0.5, 0.75, 1.0 m above the 1995-2014 baseline) under two emissions scenarios."

Figure 11, caption: you refer to a, b, ... e, f but I don't see the numbering in the panels.

Labels a, b, …e, f were added in panels of Figure 11.

L702: Is there an "and" missing between the references?

Added.

L704: ABUMIP? It is in the list of acronyms but only mentioned once. Consider writing it out.

Done.

L738: replace "certain" with "selected return heights"? (If this is what you mean here) – I think the concept of amplification factor could benefit from an example.

This sentence has been reformulated to:

"Projections of future changes in ESLs due to SLR are often reported through so-called amplification factors, which correspond to the change in the expected frequency of a given contemporary ESL height under climate change scenarios (…)"

The following sentence, which provided an example for the amplification factor of the ESL with a return period of 100-yr, has been expanded to provide a more direct link to the amplification factor concept:

"For instance, the IPCC AR6 (Fox-Kemper et al., 2021) projected that the SL associated with the historical centennial event, which is the event that historically had a 1% chance of occurring each year (once per century on average), will be exceeded at least annually (i.e. corresponding to an amplification factor of 100) at 19-31% of 634 tide gauges worldwide in 2050, and at 60-82% in 2100."

In L745-746 you mention the "amplification factor of the frequency of ESLs" as opposed to "probability of ESLs" in L738. While I think I understand, it might be confusing to some readers.

Line 738 has been rewritten. The new text avoids confusion with lines 745-746 as it does not mention "probability of ESLs" anymore.

Figure 12 caption: can you give a brief description of what the amplification factor means? I agree with Rev2 that this could be useful.

We updated the caption to:

**Figure 12: Amplification factors showing the expected change of frequency of the historical centennial SL event in 2100 projected by the IPCC AR6, for Europe, under the SSP2-4.5 middle-of-the-road emission scenario (obtained from Fig. 9.32 of Fox-Kemper et al., 2021). In this figure, an amplification factor of 10 means that the historical centennial SL event will become a decennial event in 2100, while an amplification factor of 100 means that the historical centennial SL event will become an annual event in 2100.**

**Figure 13: Add information about which scenario is used (low probability?). And I guess the changes are associated with storm surges, waves AND MSL changes?**

The caption of Figure 13 has been updated to:

**Figure 13: Projected changes in the height of ESLs associated with storm surges and waves only under a worst case scenario (95[th] percentile of the centennial event, corresponding to a return period of 0.01 yr[-1]) by 2100 relative to 1980-2014 along the European coastline (adapted from Fig. 3 of Jevrejeva et al., 2023, using data from Vousdoukas et al., 2018).**

We confirm that the changes are only associated with storm surges and waves in this figure (corresponding to Fig 3 of Jevrejeva et al. 2023, as stated in the caption).

**Section 6 Key developments per region**

The newly added introduction to this section is a great improvement and serves as a central thread to the subsections. Well done.

Thank you.

I only have a few suggestions that hopefully improve clarity.

- The basin-averaged sea level for reconstructions and projections is formed over the coloured areas shown in Figure 14, correct? That is, they are not averages over coastal sea level as shown in Figure 15-19b and c?
  *Indeed, as indicated in the caption of Fig 15-19a, the reconstructed* basin average *(Figure 14) mean relative SL is shown in these panels. The captions of Fig 15-19 have been updated to account for your other comments (see below).*

- Following a comment by Rev2, I think it is worth mentioning explicitly that the reconstruction still contains GRD effects due to present day barystatic mass changes, thus justifying calling the RSLR instead of geocentric SLR.
  *This is now clarified in the text, see the answer to your next comment.*

- Is it correct that GIA is removed in the reconstruction but not in the projections shown in Figures 15-19a? If so, I would ask the authors to consider removing GIA from the projections, too, for consistency or, alternatively, show the reconstruction with GIA. Either way, please clearly state for both, reconstructions and projections, whether they are shown with or without GIA.
  *Thanks for pointing this out. We corrected the text to improve the clarity regarding the account of GIA in Figures 15-19a.*
  *In the introduction of Section 6, the sentence:*

  *"In addition, Figures 15 to 19 provide, for each regional sea, basin-averaged relative SL (but GIA effects are not included) over 1900-2014, basin averaged projected multi-model ensemble mean relative SL until 2100"* has been updated to: *"In addition, Figures 15 to 19 provide, for each regional sea, basin-averaged relative SL (**with GIA and GRD effects being included**) over 1900-2014, basin averaged projected multi-model ensemble mean relative SL until 2100".*

  *The sentences "Note that in Figures 15-19, relative SL shown is after removal of GIA effects. In addition, the vertical reference of the reconstructed relative SL timeseries has been adjusted to match projected mean sea level records, as it is arbitrary" have been updated to:*
  *" **In Figures 15-19, reconstructed relative SL with the effect of GIA (and GRD from contemporary mass loss of land-based ice) are shown.** In addition, the vertical reference of the reconstructed relative SL timeseries has been adjusted to match projected mean sea level records, as it is arbitrary."*

  *Finally, the captions of Fig 15-19 have been updated to: "Figure 1[5-9]: (a) Yearly reconstructed basin average (Figure 14) mean relative SL over 1900-2014 from Dangendorf et al. (2019) **with the effect of GIA and GRD from contemporary mass loss of land-based ice,** together with basin average projected multi-model ensemble mean relative SL until 2100 and relative to 1995-2014 under SSP1-2.6, SSP2-4.5 and SSP5-8.5. Shadings indicate the 17th-83rd percentile uncertainties under SSP2-4.5 and SSP5-8.5. **Projections were obtained from AR6 IPCC accounting for VLM (including GIA) effects.** (b) Linear trends of VLM over 1995-2020 (Oelsmann et al., 2024). (c) 50- year return levels of extreme still water levels representative of the recent past from (…)."*

- Table 3: What about the North Sea? GIA has a relatively large impact in its northeastern part along the Norwegian coast.
  We decided to show the basin-averaged rates for the European Seas that are the most impacted by GIA, i.e., the European Arctic and Baltic Sea.

Figure 15-19, captions: since the subpanels are labelled a, b and c you can use that in the caption as well (instead of top, lower left, lower right). As a consequence, I would also ask the authors to refer to e.g. Figure 15a,b,c when appropriate.

Indeed. Captions of Figures 15-19 were updated (see above) and panels are referred to (a), (b) and (c), throughout the manuscript.

**6.1 Atlantic Ocean**

In general: north-eastern -> northeastern

Done (as well for south-eastern, north-western, north-east, etc.)

L911: Figure 15b.

Done.

L928: Refer to Figure 15c somewhere in this paragraph? Figure 15 is only mentioned very generally in the end. I suggest you move this last sentence of the paragraph higher up!

The last sentence of the paragraph "Extreme still water levels over the recent past range from 1-2 m for the coast of Portugal to 7-8 m in the macrotidal Bay of Mont Saint Michel (France) (Figure 15)" has been moved higher up and modified to the following one, with an explicit reference to Figure 15c:

"The 50-yr return period extreme still water levels over the recent past range from 1-2 m for the coast of Portugal to 7-8 m in the macrotidal Bay of Mont Saint Michel (France) (Figure 15c)."

L966: One parenthesis too many in the reference.
Done.

L999: "ones of the world regions" → two regions
This sentence has been rewritten to :

"The English Channel and the Irish Sea are amongst the world regions where tides would change the most substantially in response to SLR"

L1004: e.g,, -> e.g.,
Thanks, corrected.

**6.2 North Sea**

L1046: should this be "relative" SL?

Yes, added.

L1049: Several...

Corrected, thanks.

L1054: Refer to Figure 16b somewhere in this sentence.
Added at the end of the sentence.

L1084: raise → change or affect? Variability or negative trends may also lower the baseline.
The sentence has been rewritten using "influence".

L1142: Remove comma after Lobeto et al. for consistency with subsequent references.
Done.

**6.4 Mediterranean Sea and Black Sea**

L1313: Figure 18, top panel → Figure 18a
Done.

**6.5 Baltic Sea**

Figure 19a: are you sure you are plotting the reconstructed sea level WITHOUT GIA? To me, it seems that the trend is negative, certainly for the entire period but also for 1950-2014 when it should be positive according to Table 3 (Baltic no-GIA)
This has been corrected, thanks.

**7 Conclusion**

L1566: south-eastern -> southeastern (for consistency with northeastern used higher up in this section)
Done.

---

## Editor Decision (ED1)

The handling editor thanks the authors for their submission of Chapter 2 "Sea Level Rise in Europe: observations and projections" and for their revisions of the paper based on feedback from two anonymous reviewers and three community comments. The authors have addressed the reviewers comments satidfactorily either by following the reviewers suggestions or convincingly arguing why they decided not to.

Based on my review of the revised manuscript, I recommend that this manuscript requires minor corrections before finalization. The corrections I suggest are mainly related to improving clarity and detailed below.

**1 Introduction**
L81: icesheets -> ice sheets

**2 Summary of previous assessments**
L133: I think the two references can be in the same parentheses.

**3 Regional observations**

Figure 3, caption: the caption is referring to panel a and b but as far as I can see, the panels are not labelled. I suggest changing to left and right. Also, there is no "top right".

Figure 4, caption: you refer to Figure 3b but there is no "b" in Figure 3.

L301: sea level dynamics ARE highly…

Figure 5: Referring to a comment by Rev2: You do not refer to the two periods shown in Figure 5. I would ask you to explain them in the caption and add a sentence in the text to highlight the changing decadal trends.

L378: time -> temporal

L380: I think it would be useful to explain what you mean by "relative measure of VLM". Relative to what?

L433: Calafat et al. (2022) determined … (remove "has")

**Section 4 Drivers of sea level rise and extremes**
I would suggest distinguishing more between past and present drivers and projected drivers. This section does a little bit of both. Section 4.1 covers the past and projections (ice sheets only), section 4.2 mostly the observational period and section 4.3 seems to be period-independent.

Section 4.1 could be shortened by moving the part on projections to section 5, particularly the last paragraph on ice shelf collapse.

L463: Fig. 2.8b → Figure 8b

L464: make about → account for

L469-470: suggest rewriting to: The role of atmospheric dynamics is also uncertain.

L482: Figure 8c

L487: MICI has not yet been defined.

L521: I think you can remove "which are the changes in the amount of water stored on land"

L529: replace "is" with "are"

L535: icesheets -> ice sheets

Figure 9, legend: what is GMOM?

**Section 5 Projections of sea level rise…**

L656-657: "Figure 11 regional SLR projections…" the first part of this sentence reads strange.

Figure 11, caption: you refer to a, b, … e, f but I don't see the numbering in the panels.

L702: Is there an "and" missing between the references?

L704: ABUMIP? It is in the list of acronyms but only mentioned once. Consider writing it out.
L738: replace "certain" with "selected return heights"? (If this is what you mean here) – I think the concept of amplification factor could benefit from an example. In L745-746 you mention the "amplification factor of the frequency of ESLs" as opposed to "probability of ESLs" in L738. While I think I understand, it might be confusing to some readers.
Figure 12 caption: can you give a brief description of what the amplification factor means? I agree with Rev2 that this could be useful.
Figure 13: Add information about which scenario is used (low probability?). And I guess the changes are associated with storm surges, waves AND MSL changes?

**Section 6 Key developments per region**
The newly added introduction to this section is a great improvement and serves as a central thread to the subsections. Well done. I only have a few suggestions that hopefully improve clarity.
- The basin-averaged sea level for reconstructions and projections is formed over the coloured areas shown in Figure 14, correct? That is, they are not averages over coastal sea level as shown in Figure 15-19b and c?
- Following a comment by Rev2, I think it is worth mentioning explicitly that the reconstruction still contains GRD effects due to present day barystatic mass changes, thus justifying calling the RSLR instead of geocentric SLR.
- Is it correct that GIA is removed in the reconstruction but not in the projections shown in Figures 15-19a? If so, I would ask the authors to consider removing GIA from the projections, too, for consistency or, alternatively, show the reconstruction with GIA. Either way, please clearly state for both, reconstructions and projections, whether they are shown with or without GIA.
- Table 3: What about the North Sea? GIA has a relatively large impact in its northeastern part along the Norwegian coast.

Figure 15-19, captions: since the subpanels are labelled a, b and c you can use that in the caption as well (instead of top, lower left, lower right). As a consequence, I would also ask the authors to refer to e.g. Figure 15a,b,c when appropriate.

**6.1 Atlantic Ocean**
In general: north-eastern -> northeastern
L911: Figure 15b.
L928: Refer to Figure 15c somewhere in this paragraph? Figure 15 is only mentioned very generally in the end. I suggest you move this last sentence of the paragraph higher up!
L966: One parenthesis too many in the reference.
L999: "ones of the world regions" → two regions
L1004: e.g,, -> e.g.,

**6.2 North Sea**
L1046: should this be "relative" SL?
L1049: Several…
L1054: Refer to Figure 16b somewhere in this sentence.
L1084: raise → change or affect? Variability or negative trends may also lower the baseline.
L1142: Remove comma after Lobeto et al. for consistency with subsequent references.

**6.4 Mediterranean Sea and Black Sea**
L1313: Figure 18, top panel → Figure 18a

**6.5 Baltic Sea**

Figure 19a: are you sure you are plotting the reconstructed sea level WITHOUT GIA? To me, it seems that the trend is negative, certainly for the entire period but also for 1950-2014 when it should be positive according to Table 3 (Baltic no-GIA)

**7 Conclusion**
L1566: south-eastern -> southeastern (for consistency with northeastern used higher up in this section)